# Practical Differentially Private Top-$k$ Selection with Pay-what-you-get Composition

David Durfee[1] and Ryan Rogers[1]

[1]Data Science Applied Research, LinkedIn

## Abstract

We study the problem of top-$k$ selection over a large domain universe subject to user-level differential privacy. Typically, the exponential mechanism or report noisy max are the algorithms used to solve this problem. However, these algorithms require querying the database for the count of each domain element. We focus on the setting where the data domain is unknown, which is different than the setting of frequent itemsets where an *apriori* type algorithm can help prune the space of domain elements to query. We design algorithms that ensures (approximate) $(\varepsilon, \delta > 0)$-differential privacy and only needs access to the true top-$\bar{k}$ elements from the data for any chosen $\bar{k} \geq k$. We consider both the setting where a user's data can modify an arbitrary number of counts by at most 1, i.e. unrestricted sensitivity, and the setting where a user's data can modify at most some small, fixed number of counts by at most 1, i.e. restricted sensitivity. Additionally, we provide a *pay-what-you-get* privacy composition bound for our algorithms. That is, our algorithms might return fewer than $k$ elements when the top-$k$ elements are queried, but the overall privacy budget only decreases by the size of the outcome.

## 1 Introduction

Determining the top-$k$ most frequent items from a massive dataset in an efficient way is one of the most fundamental problems in data science, see Ilyas et al. [17] for a survey of top-$k$ processing techniques. However, it is important to consider users' privacy in the dataset, since results from data mining approaches can reveal sensitive information about a user's data [20]. Simple thresholding techniques, e.g. $k$-anonymity, do not provide formal privacy guarantees, since adversary background knowledge or linking other datasets may cause someone's data in a protected dataset to be revealed [24]. Our aim is to provide rigorous privacy techniques for determining the top-$k$ so that it can be built on top of highly distributed, real-time systems that might already be in place.

Differential privacy has become the gold standard for rigorous privacy guarantees in data analytics. One of the primary benefits of differential privacy is that the privacy loss of a computation on a dataset can be quantified. Many companies have adopted differential privacy, including Google [15], Apple [1], Uber [18], Microsoft [9], and LinkedIn [21], as well as government agencies, like the U.S. Census Bureau [8]. In this work, we hope to extend the use of differential privacy in practical systems to allow analysts to compute the $k$ most frequent elements in a given dataset. We are certainly not the first to explore this topic, yet the previous works require querying the count of every domain element, e.g. report noisy max [10] and the exponential mechanism [25], or require some structure on the large domain universe, e.g. frequent item sets (see Related Work). We aim to design practical, (approximate) differentially private algorithms that do not require any structure on the data domain, which is typically the case in exploratory data analysis. Our algorithms work in the setting where data is preprocessed prior to running our algorithms, so that the differentially private computation only accesses a subset of the data while still providing user level privacy in the full underlying dataset.

We design $(\varepsilon, \delta > 0)$-differentially private algorithms that can return the top-$k$ results by querying the counts of elements that only exist in the dataset. To ensure user level privacy, where we want to protect the privacy of a user's entire local dataset that might consist of many data records, we consider two different settings. In the *restricted sensitivity setting*, we assume that a user can modify the counts by at most 1 across at most a fixed number $\Delta$ of elements in a data domain, which is assumed to be known. An example of such a setting would be computing the top-$k$ countries where users have a certain skill set. Assuming a user can only be in one country, we have $\Delta = 1$. In the more general setting, we consider *unrestricted sensitivity*, where a user can modify the counts by at most 1 across an arbitrary number of elements. An example of the unrestricted setting would be if we wanted to compute the top-$k$ articles with distinct user engagement (liked, commented, shared, etc.). We design different algorithms for either setting so that the privacy parameter $\varepsilon$ needs to scale with either $\approx \Delta$ in the restricted sensitivity setting or $\approx \sqrt{k}$ in the unrestricted setting. Thus, our differentially private algorithms will ensure user level privacy despite a user being able to modify the counts of an arbitrary number of elements.

The reason that our algorithms are approximate differentially private is because we want to allow our algorithms to not have to know the data domain, or any structure on it. For exploratory analyses, one would like to not have to provide the algorithm the full data domain beforehand. The mere presence of a domain element in the exploratory analysis might be the result of a single user's data. Hence, if we remove a user's data in a neighboring dataset, there are some outcomes that cannot occur. We design algorithms such that these events occur with very small $\delta$ probability. Simultaneously, we ensure that the private algorithms do not compromise the efficiency of existing systems.

As a byproduct of our analysis, we also include some results of independent interest. In particular, we give a composition theorem that essentially allows for *pay-what-you-get* privacy loss. Since our algorithms can potentially output fewer than $k$ elements when asked for the top-$k$, we allow the analyst to ask more queries if the algorithms return fewer than $k$ outcomes, up to some fixed bound. Further, we define a condition on differentially private algorithms that allows for better composition bounds than the general optimal composition bounds [19, 26]. Lastly, we show how we can achieve a one-shot differentially private algorithm that provides a ranked top-$k$ result and has privacy parameter that scales with $\sqrt{k}$, which uses a different noise distribution than work from Dwork et al. [14].

We see this work as bringing together multiple theoretical results in differential privacy to arrive at a practical privacy system that can be used on top of existing, real-time data analytics platforms for massive datasets distributed across multiple servers. Essentially, the algorithms allow for solving the top-$\bar{k}$ problem first with the existing infrastructure for any chosen $\bar{k} \geq k$, and then incorporate noise and a threshold to output the top-$k$, or fewer outcomes. In our approach, we can think of the existing system, such as online analytical processing (OLAP) systems, as a blackbox top-$k$ solver and without adjusting the input dataset or opening up the blackbox, we can still implement private algorithms.

## 1.1 Related Work

There are several works in differential privacy for discovering the most frequent elements in a dataset, e.g. top-$k$ selection and heavy hitters. In the *local* privacy setting, there has been both academic work [3, 4] as well as industry solutions [1, 16] to identifying the heavy hitters. Note that these algorithms require some additional structure on the data domain, such as fixed length words, where the data can be represented as a sequence of some known length and each element of the sequence belongs to some known set. We will be working in the *trusted curator* model. There has been several works in this model that estimate frequent itemsets subject to differential privacy, including [5, 23, 28, 22, 29]. Similar to our work, Bhaskar et al. [5] first solve the top-$\bar{k}$ problem nonprivately (but $\bar{k} \geq k$ can be the full domain for certain databases) and then use the exponential mechanism to return an estimate for the top-$k$. The primary difference between these works and ours is that the domain universe in our setting is unknown and not assumed to have any structure. For itemsets, one can iteratively build up the domain from smaller itemsets, as in the locally private algorithms.

We assume no structure on the domain, as one would assume without considering privacy restrictions. This is a highly desirable feature for making differential privacy practical, since the algorithms can work over arbitrary domains. Chaudhuri et al. [7] considers the problem of returning the argmax subject to differential privacy, where their algorithm works in the *range independent* setting. That is, their algorithms can return domain elements that are unknown to the analyst querying the dataset. However, their large margin mechanism can run over the entire domain universe in the worst case.

The algorithms in [7] and [5] share a similar approach in that both use the exponential mechanism on elements above a threshold. In order to obtain *pure*-differential privacy ($\delta = 0$), Bhaskar et al. [5] samples uniformly from elements below the threshold, whereas Chaudhuri et al. [7] never sample anything from this remaining set and thus satisfy *approximate*-differential privacy ($\delta > 0$). Our approach will also follow this high-level idea, but we set the threshold based on an input parameter to ensure computational efficiency. To our knowledge, there are no top-$k$ differentially private algorithms for the unknown domain setting that never require iterating over the entire domain.

There have been several works bounding the total privacy loss of an (adaptive) sequence of differentially private mechanisms, including *basic composition* [12, 10], *advanced composition* (with improvements) [13, 11, 6], and *optimal composition* [19, 26]. There has also been work in bounding the privacy loss when the privacy parameters themselves can be chosen adaptively — where the previous composition theorems cannot be applied — with *pay-as-you-go* composition [27]. In this work, we provide a *pay-what-you-get* composition theorem for our algorithms which allows the analyst to only pay for the number of elements that were returned by our algorithms in the overall privacy budget. Because our algorithms can return fewer than $k$ elements when asked for the top-$k$, we want to ensure the analyst can ask many more queries if fewer than $k$ elements have been given.

## 2    Preliminaries

We will represent the domain as $[d] := \{1, \cdots, d\}$ and a user $i$'s data as $x_i \in 2^{[d]} =: \mathcal{X}$. We then write a dataset of $n$ users as $\mathbf{x} = \{x_1, \cdots, x_n\}$. We say that $\mathbf{x}, \mathbf{x}'$ are neighbors if they differ in the addition or deletion of one user's data, e.g. $\mathbf{x} = \mathbf{x}' \cup \{x_i\}$. We now define differential privacy [12].

**Definition 2.1** (Differential Privacy). *An algorithm $\mathcal{M}$ that takes a collection of records in $\mathcal{X}$ to some arbitrary outcome set $\mathcal{Y}$ is $(\varepsilon, \delta)$-differentially private (DP), or $\varepsilon$-DP if $\delta = 0$, if for all neighbors $\mathbf{x}, \mathbf{x}'$ and for all outcome sets $S \subseteq \mathcal{Y}$, we have*

$$\Pr[\mathcal{M}(\mathbf{x}) \in S] \leq e^\varepsilon \Pr[\mathcal{M}(\mathbf{x}') \in S] + \delta$$

In this work, we want to select the top-$k$ most frequent elements in a dataset $\mathbf{x}$. Let $h_j(\mathbf{x}) \in \mathbb{N}$ denote the number of users that have element $j \in [d]$, i.e. $h_j(\mathbf{x}) = \sum_{i=1}^{n} \mathbb{1}\{j \in x_i\}$. We then sort the counts and denote the ordering as $h_{i_{(1)}}(\mathbf{x}) \geq \cdots \geq h_{i_{(d)}}(\mathbf{x})$ with corresponding elements $i_{(1)}, \cdots, i_{(d)} \in [d]$. Hence, from dataset $\mathbf{x}$, we seek to output $i_{(1)}, \cdots, i_{(k)}$ where we break ties in some arbitrary, data independent way.

Note that for neighboring datasets $\mathbf{x}$ and $\mathbf{x}'$, the corresponding neighboring histograms $\mathbf{h} = \mathbf{h}(\mathbf{x})$ and $\mathbf{h}' = \mathbf{h}(\mathbf{x}')$ can differ in all $d$ positions by at most 1, i.e. $||\mathbf{h} - \mathbf{h}'||_\infty \leq 1$. In some instances, one user can only impact the count on at most a fixed number of coordinates. We then say that $\mathbf{h}, \mathbf{h}'$ are $\Delta$-restricted sensitivity neighbors if $||\mathbf{h} - \mathbf{h}'||_\infty \leq 1$ and $||\mathbf{h} - \mathbf{h}'||_0 \leq \Delta$.

The algorithms we describe will only need access to a histogram $\mathbf{h}(\mathbf{x}) = (h_1(\mathbf{x}), \cdots, h_d(\mathbf{x})) \in \mathbb{N}^d$, where we drop $\mathbf{x}$ when it is clear from context. We will be analyzing the privacy loss of an individual user over many different top-$k_1$, top-$k_2$, $\cdots$ queries on a larger, overall dataset. Consider the example where we want to know the top-$k_1$ articles that distinct users engaged with, then we want to know the top-$k_2$ articles that distinct users engaged with in Germany, and so on. A user's data can be part of each input histogram, so we want to compose the privacy loss across many different queries.

In our algorithms, we will add noise to the histogram counts. The noise distributions we consider are from a Gumbel random variable or a Laplace random variable where $\texttt{Gumbel}(b)$ has density function $p_{\texttt{Gumbel}}(z; b)$ and $\texttt{Lap}(b)$ has density function $p_{\texttt{Lap}}(z; b)$, with

$$p_{\texttt{Gumbel}}(z; b) = \frac{1}{b} \cdot \exp\left(-(z/b + e^{-z/b})\right) \qquad \text{and} \qquad p_{\texttt{Lap}}(z; b) = \frac{1}{2b} \cdot \exp\left(-|z|/b\right). \quad (1)$$

## 3    Main Algorithm and Results

We now present our main algorithm for reporting the top-$k$ domain elements. The *limited domain* procedure $\texttt{LimitDom}^{k,\bar{k}}$ given in Algorithm 1 takes as input a histogram $\mathbf{h} \in \mathbb{N}^d$, parameter $k$, some cutoff $\bar{k} \geq k$ for the number of elements to consider, and privacy parameters $(\varepsilon, \delta)$. It then returns at most $k$ indices in relative rank order. Our algorithm can be thought of as solving the top-$\bar{k}$ problem

with access to the true data, then from this set of histogram counts, adds noise to each count and sorts them to return at most $k$ indices with counts that are above some data dependent, noisy threshold. The noise that we add will be from a Gumbel distribution, given in (1), which has a nice connection with the exponential mechanism [25] (see Section 4). In later sections we will present a sketch of the analysis with some extensions. We present the formal analysis in the supplementary file.

---

**Algorithm 1** $\texttt{LimitDom}^{k,\bar{k}}$; Top-$k$ from the $\bar{k} \geq k$ limited domain

---

**Input:** Histogram $\mathbf{h}$; privacy parameters $\varepsilon, \delta$.
**Output:** Ordered set of indices.
Sort $h_{(1)} \geq h_{(2)} \geq \cdots$.
Set $h_\perp = h_{(\bar{k}+1)} + 1 + \ln(\min\{\Delta, \bar{k}, d - \bar{k}\}/\delta)/\varepsilon$.
Set $v_\perp = h_\perp + \texttt{Gumbel}(1/\varepsilon)$.
**for** $j \leq \bar{k}$ **do**
    Set $v_{(j)} = h_{(j)} + \texttt{Gumbel}(1/\varepsilon)$.
Sort $\{v_{(j)}\} \cup v_\perp$ and let $v_{i_{(1)}}, ...., v_{i_{(j)}}, v_\perp$ be the sorted list up until $v_\perp$.
Return $\{i_{(1)}, ..., i_{(j)}, \perp\}$ if $j < k$, otherwise return $\{i_{(1)}, ..., i_{(k)}\}$.

---

We now state its privacy guarantee.

**Theorem 1.** *Algorithm 1 is $(\varepsilon', \delta + \delta')$-DP for any $\delta' \geq 0$ where*

$$\varepsilon' = \min\left\{k\varepsilon, k\varepsilon \cdot \left(\frac{e^\varepsilon - 1}{e^\varepsilon + 1}\right) + \varepsilon\sqrt{2k\ln(1/\delta')}, \frac{k\varepsilon^2}{2} + \varepsilon\sqrt{\frac{1}{2}k\ln(1/\delta')}\right\}. \qquad (2)$$

Note that our algorithm is not guaranteed to output $k$ indices, and this is key to obtaining our privacy guarantees. The primary difficulty here is that the indices within the true top-$\bar{k}$ can change by adding or removing one person's data. The purpose of the threshold, $\perp$, is then to ensure that the probability of outputting any index in the top-$\bar{k}$ for histogram $\mathbf{h}$ but not in the top-$\bar{k}$ for a neighboring histogram $\mathbf{h}'$ is bounded by $\frac{\delta}{\min\{\Delta,\bar{k}\}}$. We give more high-level intuition in Section 3.2.

In order to maximize the probability of outputting $k$ indices, we want to minimize our threshold value. In the unrestricted sensitivity setting, it becomes natural to consider how to set $\bar{k}$, as there becomes a tradeoff where $h_{(\bar{k}+1)}$ is decreasing in $\bar{k}$ whereas $\ln(\bar{k}/\delta)/\varepsilon$ is increasing in $\bar{k}$. Ideally, we would set $\bar{k}$ to be a point within the histogram in which we see a sudden drop, but setting it in such a data dependent manner would violate privacy. Instead, we will simply consider the optimization problem of finding index $\bar{k}$ that minimizes $h_{(\bar{k}+1)} + 1 + \ln(\bar{k}/\delta)/\varepsilon$ (and is computationally feasible), and we will solve this problem with standard DP techniques. We can then find a noisy estimate of the optimal parameter $\bar{k}$ for a given histogram $\mathbf{h}$ and the resulting procedure increases the privacy loss by substituting $k + 1$ for $k$ in the guarantees in Theorem 1. See the supplementary file for more details.

**Pay-what-you-get Composition**

---

**Algorithm 2** $\texttt{multiLimitDom}^{k^\star, \ell^\star}$; Multiple queries to limited domain

---

**Input:** Adaptive stream $\mathbf{h}_1, \mathbf{h}_2, ....,$ integers $k^\star$ and $\ell^\star$, and per iterate privacy parameters $\varepsilon, \delta$.
**Output:** Sequence of outputs $(o_1, \cdots, o_\ell)$ for $\ell \leq \ell^\star$.
**while** $k^\star > 0$ and $\ell^\star > 0$ **do**
    Based on previous outcomes, select adaptive histogram $\mathbf{h}_i$ and parameters $k_i, \bar{k}_i$
    **if** $k_i \leq k^\star$ **then**
        Let $o_i = \texttt{LimitDom}^{k_i, \bar{k}_i}(\mathbf{h}_i)$ with privacy parameters $\varepsilon$ and $\delta$
        $k^\star \leftarrow k^\star - |o_i|$ and $\ell^\star \leftarrow \ell^\star - 1$
Return $o = (o_1, o_2, \cdots)$

---

While the privacy loss for Algorithm 1 will be a function of $k$ regardless of whether it outputs far fewer than $k$ indices, we can actually show that in making multiple calls to this algorithm, we can instead bound the privacy loss in terms of the number of indices that are output. More specifically, we will take the length of the output for each call to Algorithm 1, which is not deterministic, and

ensure that the sum of these lengths does not exceed some $k^\star$. Additionally, we need to restrict how many top-$k$ queries can be asked of our system, which we denote as $\ell^\star$. Accordingly, the privacy loss will then be in terms of $k^\star$ and $\ell^\star$. We detail the procedure $\mathtt{multiLimitDom}^{k^\star, \ell^\star}$ in Algorithm 2.

From a practical perspective, this means that if we allowed a client to make multiple top-$k$ queries with a total budget of $k^\star$, whenever a top-$k$ query was made their total budget would only decrease with the size of the output, as opposed to $k$. We will further discuss in Section 3.1 how this property in some ways can actually provide higher utility than standard approaches that have access to the full histogram and must output $k$ indices. We then have the following privacy statement.

**Theorem 2.** *For any $\delta' \geq 0$, $\mathtt{multiLimitDom}^{k^\star, \ell^\star}$ in Algorithm 2 is $(\varepsilon^\star, 2\ell^\star\delta + \delta')$-DP where*

$$\varepsilon^\star = \min\left\{ k^\star\varepsilon, k^\star\varepsilon \cdot \left(\frac{e^\varepsilon - 1}{e^\varepsilon + 1}\right) + \varepsilon\sqrt{2k^\star \ln(1/\delta')}, \frac{k^\star\varepsilon^2}{2} + \varepsilon\sqrt{\frac{1}{2}k^\star \ln(1/\delta')} \right\}. \tag{3}$$

In the supplementary file, we present other variants of our main algorithm that are better to use in specific settings. In particular, we consider alternatively adding Laplace noise to Algorithm 1 which allows our $\varepsilon$-privacy loss parameter to scale with $\Delta$ (rather than $\sqrt{k}$) in the restricted sensitivity setting, but unlike using Gumbel noise, it does not benefit from the pay-what-you-get composition.

**Improved Advanced Composition**

We also provide a result that may be of independent interest. In Section 4, we consider a slightly tighter characterization of pure $(\delta = 0)$ differential privacy, which we refer to as *range-bounded*, and show that it can improve the total privacy loss over a sequence of adaptively chosen private algorithms. In particular, we consider the exponential mechanism, which is known to be $\varepsilon$-DP, and show that it has even stronger properties that allow us to show it is $\varepsilon$-*range-bounded* under the same parameters. Accordingly, we can then give improved advanced composition bounds for the exponential mechanism compared to the optimal composition bounds for general $\varepsilon$-DP mechanisms given in [19, 26] (we show a comparison of these bounds in Section 4.3).

## 3.1 Accuracy Comparisons

In contrast to previous work in top-$k$ selection subject to DP, our algorithms can return fewer than $k$ indices. Typically, accuracy in this setting is to return a set of exactly $k$ indices such that each returned index has a count that is at least the $k$-th ranked value minus some small amount. We then relax the utility statement to allow returning fewer than $k$ indices

**Definition 3.1.** *Given histogram $\mathbf{h}$ along with non-negative integers $k$ and $\alpha$, we say that a subset of indices $\mathbf{d} \subseteq [d] \cup \{\perp\}$ is an $(\alpha, k)$-accurate if for any $i \in \mathbf{d}$ such that $i \neq \perp$, we have $h_i \geq h_{(k)} - \alpha$.*

For this definition, we can give asymptotically better accuracy guarantees than the exponential mechanism achieves, which are known to be tight [2], but it is important to mention that our definition does not require $k$ indices to be output. Accordingly, we add a sufficient condition under which our algorithm will return $k$ indices with a given probability. We defer the proof to the supplementary file.

**Lemma 3.1.** *For any histogram $\mathbf{h}$, with probability at least $1 - \beta$ the output from Algorithm 1 with parameters $k, \bar{k}, \varepsilon, \delta$ is $(\alpha, k)$-accurate where $\alpha = \ln(k\bar{k}/\beta)/\varepsilon$. Additionally, we have that Algorithm 1 will return $k$ indices with probability at least $1 - \beta$ if*

$$h_{(k)} \geq h_{(\bar{k}+1)} + 1 + \ln(\min\{\Delta, \bar{k}\}/\delta)/\varepsilon + \ln(k/\beta)/\varepsilon$$

The first statement is essentially equivalent to Theorem 6 in [2] which would have $\alpha = \ln(kd/\beta)/\varepsilon$ in this setting because we incorporate advanced composition at the end of the analysis and we consider the absolute counts (not normalized by the total number of users). Accordingly, our $\alpha$ parameter swaps $d$ with $\bar{k}$ as expected, and will improve the accuracy statement for the output indices.

Even for histograms in which we are unlikely to return $k$ indices, we see this as the primary advantage of our *pay-what-you-get* composition. The indices that are returned are normally the clear winners, i.e. indices with counts substantially above the $(\bar{k} + 1)$th value, and then the $\perp$ value is informative that the remaining values are approximately equal where the analyst only has to pay for this single output as opposed to paying for the remaining outputs that are close to a random permutation.

## 3.2 Our Techniques

The main challenge with ensuring differential privacy in our setting is that preprocessing the data to the true top-$\bar{k}$ indices will lead to different domains for neighboring histograms. More explicitly, the indices within the top-$\bar{k}$ can change by adding or removing one user's data, and this makes ensuring pure differential privacy without knowing the domain impossible. However, the key insight will be that only indices whose value is within 1 of $h_{(\bar{k}+1)}$, i.e. the count of the $(\bar{k}+1)$th ranked index, can go in or out of the top-$\bar{k}$ by adding or removing one user's data. Accordingly, the noisy threshold that we add will be explicitly set such that for indices with value within 1 of $h_{(\bar{k}+1)}$, the noisy estimate exceeding the noisy threshold will be a low probability event. By restricting our output indices to those whose noisy estimate are in the top-$k$ **and** exceed a noisy threshold, we ensure that indices in the top-$\bar{k}$ for one histogram but not in a neighboring histogram will output with probability at most $\frac{\delta}{\min\{\Delta,\bar{k}\}}$. A union bound over the possible indices that can change will then give our desired bound on these bad events. We now present the high level reasoning behind the proof of privacy in Theorem 1 and defer the formal analysis to the supplementary material.

1. Adding `Gumbel` noise and taking the top-$k$ at once is equivalent to selecting the $\mathrm{argmax}$ using the exponential mechanism then removing that index and iterating, see Lemma 4.2.[1]

2. To get around the fact that the domains can change in neighboring datasets, we define a variant of Algorithm 1 that takes a histogram and a domain as input. We then prove that this variant is DP for any input domain, and for a choice of domain that depends on the input histogram, it is the same as Algorithm 1

3. Due to the choice of the count for element $\bot$, we show that for any given neighboring datasets $\mathbf{h}, \mathbf{h}'$, the probability that Algorithm 1 evaluated on $\mathbf{h}$ can return any element that is not part of the domain with $\mathbf{h}'$ occurs with probability $\delta$.

We now provide a sketch of the analysis for proving the pay-what-you-get composition bound in Theorem 2 and defer the formal proof to the supplementary material.

1. Because Algorithm 1 can be expressed as multiple iterations of the exponential mechanism, we can string together many calls to Algorithm 1 as an adaptive sequence of DP mechanisms.

2. With multiple calls to Algorithm 1, if we ever get a $\bot$ outcome, we can simply start a new top-$k$ query and hence a new sequence of exponential mechanism calls. Hence, we do not need to get $k$ outcomes before we switch to a new query.

3. To get the improved constants in (3), compared to advanced composition given in Theorem 3 [13], we introduce a tigher *range-bounded* characterization, which the exponential mechanism satisfies, that enjoys better composition, see Lemma 4.4.

## 4  Existing DP Algorithms and Extensions

We now cover some existing differentially private algorithms and extensions to them. We start with the exponential mechanism [25], and show how it is equivalent to adding noise from a particular distribution and taking the $\mathrm{argmax}$ outcome. Next, we will present a stronger privacy condition than differential privacy which will in fact lead to an improved composition bound than the optimal composition bounds [19, 26] for general DP mechanisms. Throughout, we will make use of the following composition theorem in differential privacy.

**Theorem 3** (Composition [10, 13] with improvements by [19, 26]). *Let* $\mathcal{M}_1, \mathcal{M}_2, \cdots, \mathcal{M}_t$ *be each* $(\varepsilon_i, \delta_i)$-*DP, where* $\mathcal{M}_i$ *may depend on the previous outcomes of* $\mathcal{M}_1, \cdots, \mathcal{M}_{i-1}$, *then the composed algorithm* $\mathcal{M}(\mathbf{x}) = (\mathcal{M}_1(\mathbf{x}), \mathcal{M}_2(\mathbf{x}), \cdots, \mathcal{M}_t(\mathbf{x}))$ *is* $(\varepsilon', \sum_{i=1}^{t} \delta_i + \delta')$-*DP for any* $\delta' \geq 0$ *where*

$$\varepsilon' = \min \left\{ \sum_{i=1}^{t} \varepsilon_i, \sum_{i=1}^{t} \varepsilon_i \cdot \left( \frac{e^{\varepsilon_i} - 1}{e^{\varepsilon_i} + 1} \right) + \sqrt{2 \sum_{i=1}^{t} \varepsilon_i^2 \ln(1/\delta')} \right\}.$$

## 4.1 Exponential Mechanism and Gumbel Noise

The exponential mechanism takes a quality score $q$ that maps a dataset in $\mathcal{D}$ and outcome to $\mathbb{R}$ and can be thought of as evaluating how good $q(\mathbf{x}, y)$ is for an outcome $y \in \mathcal{Y}$ on dataset $\mathbf{x} \in \mathcal{D}$. For our setting, we will be using the following quality score $q(\mathbf{h}, i) = h_i$ in the exponential mechanism.

**Definition 4.1** (Exponential Mechanism). *Let $EM_q : \mathcal{D} \to \mathcal{Y}$ be a randomized mapping where for all outputs $y \in \mathcal{Y}$ we have*

$$\Pr[EM_q(\mathbf{x}) = y] \propto \exp\left(\varepsilon q(\mathbf{x}, y)/\Delta(q)\right) \qquad \text{where} \qquad \Delta(q) := \sup_{y \in \mathcal{Y}} |q(\mathbf{x}, y) - q(\mathbf{x}', y)|.$$

We say that a quality score $q(\cdot, \cdot)$ is *monotonic* in the dataset if the addition of a data record can either increase (decrease) or remain the same with any outcome, e.g. $q(\mathbf{x}, y) \le q(\mathbf{x} \cup \{x_i\}, y)$ for any input and outcome $y$. Note that $q(\mathbf{h}, i) = h_i$ is monotonic. We then have the following privacy guarantee.

**Lemma 4.1.** *The exponential mechanism $EM_q$ is $2\varepsilon$-DP. Further, if $q$ is monotonic, then $EM_q$ is $\varepsilon$-DP.*

We point out that the exponential mechanism can be simulated by adding noise from $\texttt{Gumbel}(\Delta(q)/\varepsilon)$ to each quality score value and then reporting the outcome with the largest noisy count. This is similar to the report noisy max mechanism [10] except Gumbel noise is added rather than Laplace. We define $\texttt{pEM}_q^k$ to be the iterative peeling algorithm that first samples the outcome with the largest quality score then repeats on the remaining outcomes and continues $k$ times. We further define $\mathcal{M}_{\texttt{Gumbel}}^k(\mathbf{q}(\mathbf{x}))$ to be the algorithm that adds $\texttt{Gumbel}(\Delta(q)/\varepsilon)$ to each $q(\mathbf{x}, y)$ for $y \in \mathcal{Y}$ and takes the $k$ indices with the largest noisy counts. We then make the following connection between $\texttt{pEM}^k$ and $\mathcal{M}_{\texttt{Gumbel}}^k$, so that we can compute the top-$k$ outcomes in one-shot. We defer the proof to the supplementary file.

**Lemma 4.2.** *For any input $\mathbf{x} \in \mathcal{X}$ the peeling exponential mechanism $pEM_q^k(\mathbf{x})$ is equal in distribution to $\mathcal{M}_{Gumbel}^k(\mathbf{q}(\mathbf{x}))$. That is for any outcome vector $(o_1, \cdots, o_k) \in [d]^k$ we have*

$$\Pr[pEM_q^k(\mathbf{x}) = (o_1, \cdots, o_k)] = \Pr[\mathcal{M}_{Gumbel}^k(\mathbf{q}(\mathbf{x})) = (o_1, \cdots, o_k)]$$

We next show that the one-shot noise addition is $(\approx \sqrt{k}\varepsilon, \delta)$-DP using Theorem 3. Dwork et al. [14] also considered a one-shot approach with Laplace noise addition and in order to get the $\sqrt{k}\varepsilon$ factor on the privacy loss, their algorithm could not return the ranked list of indices. Using Gumbel noise allows us to return the ranked list of indices in one-shot with the same privacy loss.

**Corollary 4.1.** *For any $\delta \ge 0$, the one-shot $\mathcal{M}_{Gumbel}^k(\mathbf{q}(\cdot))$ is $(\varepsilon', \delta)$-DP and if $q$ is monotonic in the dataset then $\mathcal{M}_{Gumbel}^k(\mathbf{q}(\cdot))$ is $(\varepsilon'', \delta)$-DP where*

$$\varepsilon' = 2 \cdot \min\left\{ k\varepsilon, k\varepsilon(\frac{e^{2\varepsilon} - 1}{e^{2\varepsilon} + 1}) + \varepsilon\sqrt{2k \ln(\frac{1}{\delta})} \right\}, \ \varepsilon'' = \min\left\{ k\varepsilon, k\varepsilon(\frac{e^{\varepsilon} - 1}{e^{\varepsilon} + 1}) + \varepsilon\sqrt{2k \ln(\frac{1}{\delta})} \right\}.$$

## 4.2 Bounded Range Composition

It turns out that we can actually improve on the total privacy loss for this algorithm and for a wider class of algorithms in general. We first define a slightly stronger condition than (pure) differential privacy that can give a tighter characterization of the privacy loss for certain DP mechanisms.

**Definition 4.2** (Range-Bounded). *Given a mechanism $\mathcal{M}$ that takes a collection of records in $\mathcal{X}$ to outcome set $\mathcal{Y}$, we say that $\mathcal{M}$ is $\varepsilon$-range-bounded if for any neighboring databases $\mathbf{x}, \mathbf{x}'$ we have*

$$\sup_{y \in \mathcal{Y}} \ln\left( \frac{\Pr[\mathcal{M}(\mathbf{x}) = y]}{\Pr[\mathcal{M}(\mathbf{x}') = y]} \right) - \inf_{y' \in \mathcal{Y}} \ln\left( \frac{\Pr[\mathcal{M}(\mathbf{x}) = y']}{\Pr[\mathcal{M}(\mathbf{x}') = y']} \right) \le \varepsilon$$

In the supplementary file, we show that exponential mechanism achieves the same privacy parameters as in Lemma 4.1 for our stronger charaterization.

**Lemma 4.3.** *The exponential mechanism $EM_q$ is $2\varepsilon$-range-bounded. Further if $q$ is monotonic then $EM_q$ is $\varepsilon$-range bounded.*

We now show that we can achieve a better composition bound when we compose $\varepsilon$-range-bounded algorithms as opposed to using Theorem 3, which applies to the composition of general DP algorithms.

In fact, our composition bound for range-bounded algorithms improves on the *optimal* composition theorem for general DP algorithms [19, 26]. See the supplementary file for a comparison of the different bounds. We defer the proof, which largely follows a similar argument to [13].

**Lemma 4.4.** *Let* $\mathcal{M}_1, \mathcal{M}_2, \cdots, \mathcal{M}_t$ *each be* $\varepsilon_i$*-bounded range where the choice of* $\mathcal{M}_i$ *may depend on the previous outcomes of* $\mathcal{M}_1, \cdots, \mathcal{M}_{i-1}$, *then the composed algorithm* $\mathcal{M}(\mathbf{x})$ *of each of the algorithms* $\mathcal{M}_1(\mathbf{x}), \mathcal{M}_2(\mathbf{x}), \cdots, \mathcal{M}_t(\mathbf{x})$ *is* $(\varepsilon', \delta)$*-DP for any* $\delta \geq 0$ *where*

$$\varepsilon' = \min\left\{\sum_{i=1}^t \varepsilon_i, \sum_{i=1}^t \varepsilon_i \cdot \left(\frac{e^{\varepsilon_i} - 1}{e^{\varepsilon_i} + 1}\right) + \sqrt{2\sum_{i=1}^t \varepsilon_i^2 \ln(\frac{1}{\delta})}, \sum_{i=1}^t \frac{\varepsilon_i^2}{2} + \sqrt{\frac{1}{2}\sum_{i=1}^t \varepsilon_i^2 \ln(\frac{1}{\delta})}\right\}. \quad (4)$$

Note that in order to see an improvement in the advanced composition bound, we do not necessarily require that an $\varepsilon$-DP mechanism is also $\varepsilon$-*range-bounded*, but could be relaxed. More specifically, the significant term that is normally considered in advanced composition is $\sqrt{2k\ln(1/\delta')}\varepsilon$, which can be replaced with $\sqrt{\frac{\alpha^2}{2}k\ln(1/\delta')}\varepsilon$ for composing $\alpha\varepsilon$-*range-bounded* mechanisms with $\alpha \leq 2$. Consequently, we believe that this could be useful for mechanisms beyond the exponential mechanism.

### 4.3 Comparison between Bounded Range DP Composition and Optimal DP Composition

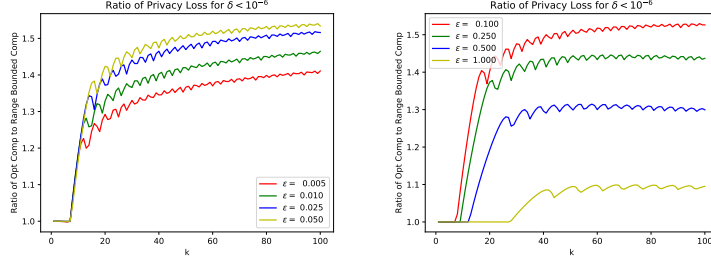

Figure 1: Plotting the ratio of the optimal DP composition bound from [19] given in Lemma 4.5 to the bounded range DP composition bound from Lemma 4.4 for various $\varepsilon$ values and $\delta < 10^{-6}$.

Here we compare the composition bound given in Lemma 4.4 and show that it can actually improve on the optimal bound for generally DP, which we state here for the homogeneous case.

**Lemma 4.5** (Optimal DP Composition [19]). *For any* $\varepsilon \geq 0$ *and* $\delta \in [0, 1]$, *the composed mechanism of* $k$ *adaptively chosen* $\varepsilon$*-DP is* $((k - 2i)\varepsilon, \delta_i)$*-DP for all* $i \in \{0, 1, \cdots, \lfloor k/2 \rfloor\}$ *where* $\delta_i = \sum_{\ell=0}^{i-1} \binom{k}{\ell}\left(e^{(k-\ell)\varepsilon} - e^{(k-2i+\ell)\varepsilon}\right)/(1 + e^\varepsilon)^k$.

In Figure 1, we plot, for various $k$ and $\varepsilon$, the ratio between the composition bound for range bounded DP algorithms and the general DP optimal composition bound, where a ratio larger than 1 means that our bound is smaller. Due to the discrete formula for $\delta_i$ in the optimal composition formula, we select the index $i$ that produces the smallest $(k - 2i)\varepsilon$ while $\delta_i \leq 10^{-6}$. Frequently, this $\delta_i$ that is selected is much smaller than the threshold $10^{-6}$, so we use this same $\delta_i$ when we compare our bounds to the optimal composition bound. Note that the jaggedness in the plot is because the optimal composition bound might be $((k - 2i)\varepsilon, \delta \ll 10^{-6})$-DP at round $k$ but $((k + 1 - 2(i + 1))\varepsilon, \delta \approx 10^{-6})$-DP at round $k + 1$. Hence, plotting only the first privacy parameter might be non-monotonic.

## 5 Conclusion

We have presented a way to efficiently report the top-$k$ elements in a dataset subject to differential privacy. Our approach does not require adjusting the input data to an existing system, nor does it require altering the non-private data analytics. Our algorithms can be seen as being an additional layer on top of existing systems so that we can leverage highly efficient, scalable data analytics platforms in our private systems. Our algorithms can balance utility in terms of both privacy with $\varepsilon$ as well as efficiency with $\bar{k}$. Further, we have improved on the general composition bounds in differential privacy that can be applied in our setting to extract more utility under the same privacy budget and have provided a pay-what-you-get composition bound.

## Footnotes

[1] Note that we could have alternatively written our algorithm in terms of iteratively applying exponential mechanism for easier analysis, but instead adding `Gumbel` noise once is computationally more efficient.

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
