[Supplementary Material]

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

. There are different approaches to solving this problem depending on whether you are in the *local* privacy model, which assumes that each data record is privatized prior to aggregation on the server, or in the *trusted curator* privacy model, which assumes that the data is stored centrally and then private algorithms can be run on top of it. In the local setting, there has been academic work [3, 4] as well as industry solutions [1, 16] to identifying the heavy hitters. Note that these algorithms require some additional structure on the data domain, such as fixed length words, where the data can be represented as a sequence of some known length $\ell$ and each element of the sequence belongs to some known set. One can then prune the space of potential heavy hitters by eliminating subsequences that are not heavy, since a subsequence is frequent only if it is contained in a frequent sequence.

We will be working in the trusted curator model. There has been several works in this model that estimate frequent itemsets subject to differential privacy, including [5, 23, 28, 22, 29]. Similar to our work, Bhaskar et al. [5] first solve the top-$\bar{k}$ problem nonprivately (but with restrictions on the choice of $\bar{k} \geq k$ which can be $d$ for certain databases) and then use the exponential mechanism to return an estimate for the top-$k$. The primary difference between these works and ours is that the domain universe in our setting is unknown and not assumed to have any structure. For itemsets, one can iteratively build up the domain from smaller itemsets, as in the locally private algorithms.

We assume no structure on the domain, as one would assume without considering privacy restrictions. This is a highly desirable feature for making differential privacy practical, since the algorithms can work over arbitrary domains. Chaudhuri et al. [7] considers the problem of returning the argmax subject to differential privacy, where their algorithm works in the *range independent* setting. That is, their algorithms can return domain elements that are unknown to the analyst querying the dataset. However, their large margin mechanism can run over the entire domain universe in the worst case. The algorithms in [7] and [5] share a similar approach in that both

use the exponential mechanism on elements above a threshold (completeness). In order to obtain *pure*-differential privacy ($\delta = 0$), [5] samples uniformly from elements below the threshold, whereas [7] never sample anything from this remaining set and thus satisfy *approximate*-differential privacy ($\delta > 0$). Our approach will also follow this high-level idea, but set the threshold in a different manner to ensure computational efficiency. To our knowledge, there are no top-$k$ differentially private algorithms for the unknown domain setting that never require iterating over the entire domain.

When the data domain is known and we want to compute the top-$k$ most frequent elements, then the usual approach is to first either use report noisy max [10], which adds Laplace noise to each count and reports the index of the largest noisy count, or use the exponential mechanism [25]. Then we can use a *peeling technique*, which removes the top element's count and then uses report noisy max or the exponential mechanism again. There has also been work in achieving a one-shot version that adds Laplace noise to the counts once and can return a set of $k$ indices, which would be computationally more efficient, [14].

There have been several works bounding the total privacy loss of an (adaptive) sequence of differentially private mechanisms, including *basic composition* [12, 10], *advanced composition* (with improvements) [13, 11, 6], and *optimal composition* [19, 26]. There has also been work in bounding the privacy loss when the privacy parameters themselves can be chosen adaptively — where the previous composition theorems cannot be applied — with *pay-as-you-go* composition [27]. In this work, we provide a *pay-what-you-get* composition theorem for our algorithms which allows the analyst to only pay for the number of elements that were returned by our algorithms in the overall privacy budget. Because our algorithms can return fewer than $k$ elements when asked for the top-$k$, we want to ensure the analyst can ask many more queries if fewer than $k$ elements have been given.

## 2 Preliminaries

We will represent the domain as $[d] := \{1, \cdots, d\}$ and a user $i$'s data as $x_i \in 2^{[d]} =: \mathcal{X}$. We then write a dataset of $n$ users as $\mathbf{x} = \{x_1, \cdots, x_n\}$. We say that $\mathbf{x}, \mathbf{x}'$ are neighbors if they differ in the addition or deletion of one user's data, e.g. $\mathbf{x} = \mathbf{x}' \cup \{x_i\}$. We now define differential privacy [12].

**Definition 2.1** (Differential Privacy). *An algorithm $\mathcal{M}$ that takes a collection of records in $\mathcal{X}$ to some arbitrary outcome set $\mathcal{Y}$ is $(\varepsilon, \delta)$-differentially private (DP) if for all neighbors $\mathbf{x}, \mathbf{x}'$ and for all outcome sets $S \subseteq \mathcal{Y}$, we have*

$$\Pr[\mathcal{M}(\mathbf{x}) \in S] \leq e^{\varepsilon} \Pr[\mathcal{M}(\mathbf{x}') \in S] + \delta$$

*If $\delta = 0$, then we simply write $\varepsilon$-DP.*

In this work, we want to select the top-$k$ most frequent elements in a dataset $\mathbf{x}$. Let $h_j(\mathbf{x}) \in \mathbb{N}$ denote the number of users that have element $j \in [d]$, i.e. $h_j(\mathbf{x}) = \sum_{i=1}^{n} \mathbb{1}\{j \in x_i\}$. We then sort the counts and denote the ordering as $h_{i_{(1)}}(\mathbf{x}) \geq \cdots \geq h_{i_{(d)}}(\mathbf{x})$ with corresponding elements $i_{(1)}, \cdots, i_{(d)} \in [d]$. Hence, from dataset $\mathbf{x}$, we seek to output $i_{(1)}, \cdots, i_{(k)}$ where we break ties in some arbitrary, data independent way.

Note that for neighboring datasets $\mathbf{x}$ and $\mathbf{x}'$, the corresponding neighboring histograms $\mathbf{h} = \mathbf{h}(\mathbf{x})$ and $\mathbf{h}' = \mathbf{h}(\mathbf{x}')$ can differ in all $d$ positions by at most 1, i.e. $||\mathbf{h} - \mathbf{h}'||_\infty \leq 1$. In some instances, one user can only impact the count on at most a fixed number of coordinates. We then say that $\mathbf{h}, \mathbf{h}'$ are $\Delta$-restricted sensitivity neighbors if $||\mathbf{h} - \mathbf{h}'||_\infty \leq 1$ and $||\mathbf{h} - \mathbf{h}'||_0 \leq \Delta$.

The algorithms we describe will only need access to a histogram $\mathbf{h}(\mathbf{x}) = (h_1(\mathbf{x}), \cdots, h_d(\mathbf{x})) \in \mathbb{N}^d$, where we drop $\mathbf{x}$ when it is clear from context. We will be analyzing the privacy loss of an individual user over many different top-$k_1$, top-$k_2$, $\cdots$ queries on a larger, overall dataset. Consider the example where we want to know the top-$k_1$ articles that distinct users engaged with, then we want to know the top-$k_2$ articles that distinct users engaged with in Germany, and so on. A user's data can be part of each input histogram, so we want to compose the privacy loss across many different queries.

In our algorithms, we will add noise to the histogram counts. The noise distributions we consider are from a Gumbel random variable or a Laplace random variable where $\texttt{Gumbel}(b)$ has PDF $p_{\texttt{Gumbel}}(z;b)$, $\texttt{Lap}(b)$ has PDF $p_{\texttt{Lap}}(z;b)$, and

$$p_{\texttt{Gumbel}}(z;b) = \frac{1}{b} \cdot \exp\left(-(z/b + e^{-z/b})\right) \qquad \text{and} \qquad p_{\texttt{Lap}}(z;b) = \frac{1}{2b} \cdot \exp\left(-|z|/b\right). \qquad (1)$$

## 3 Main Algorithm and Results

We now present our main algorithm for reporting the top-$k$ domain elements and state its privacy guarantee. The *limited domain* procedure $\texttt{LimitDom}^{k,\bar{k}}$ is given in Algorithm 1 and takes as input a histogram $\mathbf{h} \in \mathbb{N}^d$, parameter $k$, some cutoff $\bar{k} \geq k$ for the number of domain elements to consider, and privacy parameters $(\varepsilon, \delta)$. It then returns at most $k$ indices in relative rank order. At a high level, our algorithm can be thought of as solving the top-$\bar{k}$ problem with access to the true data, then from this set of histogram counts, adds noise to each count to determine the noisy top-$k$ and include each index in the output only if its respective noisy count is larger than some noisy threshold. The noise that we add will be from a Gumbel random variable, given in (1), which has a nice connection with the exponential mechanism [25] (see Section 4). In later sections we will present its formal analysis and some extensions.

---

**Algorithm 1** $\texttt{LimitDom}^{k,\bar{k}}$; Top-$k$ from the $\bar{k} \geq k$ limited domain

---

**Input:** Histogram $\mathbf{h}$; privacy parameters $\varepsilon, \delta$.
**Output:** Ordered set of indices.
Sort $h_{(1)} \geq h_{(2)} \geq \cdots$.
Set $h_\perp = h_{(\bar{k}+1)} + 1 + \ln(\min\{\Delta, \bar{k}\}/\delta)/\varepsilon$.[1]
Set $v_\perp = h_\perp + \texttt{Gumbel}(1/\varepsilon)$.
**for** $j \leq \bar{k}$ **do**
    Set $v_{(j)} = h_{(j)} + \texttt{Gumbel}(1/\varepsilon)$.
Sort $\{v_{(j)}\} \cup v_\perp$.
Let $v_{i_{(1)}}, \ldots, v_{i_{(j)}}, v_\perp$ be the sorted list up until $v_\perp$.
Return $\{i_{(1)}, \ldots, i_{(j)}, \perp\}$ if $j < k$, otherwise return $\{i_{(1)}, \ldots, i_{(k)}\}$.

---

We now state its privacy guarantee.

**Theorem 1.** *Algorithm 1 is $(\varepsilon', \delta + \delta')$-DP for any $\delta' \geq 0$ where*

$$\varepsilon' = \min\left\{ k\varepsilon, k\varepsilon \cdot \left(\frac{e^\varepsilon - 1}{e^\varepsilon + 1}\right) + \varepsilon\sqrt{2k\ln(1/\delta')}, \frac{k\varepsilon^2}{2} + \varepsilon\sqrt{\frac{1}{2}k\ln(1/\delta')} \right\}. \tag{2}$$

Note that our algorithm is not guaranteed to output $k$ indices, and this is key to obtaining our privacy guarantees. The primary difficulty here is that the indices within the true top-$\bar{k}$ can change by adding or removing one person's data. The purpose of the threshold, $\perp$, is then to ensure that the probability of outputting any index in the top-$\bar{k}$ for histogram $\mathbf{h}$ but not in the top-$\bar{k}$ for a neighboring histogram $\mathbf{h}'$ is bounded by $\frac{\delta}{\min\{\Delta, k\}}$. We give more high-level intuition on this in Section 3.2.

In order to maximize the probability of outputting $k$ indices, we want to minimize our threshold value. Accordingly, whenever we have restricted sensitivity such that $\min\{\Delta, \bar{k}\} = \Delta$, we can simply choose $\bar{k}$ to be as large as is computationally feasible because that will minimize our threshold $h_{(\bar{k}+1)} + 1 + \ln(\min\{\Delta, \bar{k}\}/\delta)/\varepsilon$. However, if the sensitivity is unrestricted or quite large, it becomes natural to consider how to set $\bar{k}$, as there becomes a tradeoff where $h_{(\bar{k}+1)}$ is decreasing in $\bar{k}$ whereas $\ln(\bar{k}/\delta)/\varepsilon$ is increasing in $\bar{k}$. Ideally, we would set $\bar{k}$ to be a point within the histogram in which we see a sudden drop, but setting it in such a data dependent manner would violate privacy. Instead, we will simply consider the optimization problem of finding index $\bar{k}$ that minimizes $h_{(\bar{k}+1)} + 1 + \ln(\bar{k}/\delta)/\varepsilon$ (and is computationally feasible), and we will solve this problem with standard DP techniques.

**Lemma 3.1** (Informal). *We can find a noisy estimate of the optimal parameter $\bar{k}$ for a given histogram $\mathbf{h}$, and this will only increase our privacy loss by substituting $k+1$ for $k$ in the guarantees in Theorem 1.*

**Pay-what-you-get Composition**

While the privacy loss for Algorithm 1 will be a function of $k$ regardless of whether it outputs far fewer than $k$ indices, we can actually show that in making multiple calls to this algorithm, we can instead bound the privacy loss in terms of the number of indices that are output. More specifically, we will instead take the length of the output for each call to Algorithm 1, which is not deterministic, and ensure that the sum of these lengths does not exceed some $k^\star$. Additionally, we need to restrict how many individual top-$k$ queries can be asked of our system, which we denote as $\ell^\star$. Accordingly, the privacy loss will then be in terms of $k^\star$ and $\ell^\star$. We detail the multiple calls procedure $\mathtt{multiLimitDom}^{k^\star, \ell^\star}$ in Algorithm 2.

From a practical perspective, this means that if we allowed a client to make multiple top-$k$ queries with a total budget of $k^\star$, whenever a top-$k$ query was made their total budget would only decrease in the size of the output, as opposed to $k$. We will further discuss in Section 3.1 how this property in some ways can actually provide higher utility than standard approaches that have access to the full histogram and must output $k$ indices. We then have the following privacy statement.

**Theorem 2.** *For any $\delta' \geq 0$, $\mathtt{multiLimitDom}^{k^\star, \ell^\star}$ in Algorithm 2 is $(\varepsilon^\star, 2\ell^\star\delta + \delta')$-DP where*

$$\varepsilon^\star = \min\left\{ k^\star\varepsilon, k^\star\varepsilon \cdot \left(\frac{e^\varepsilon - 1}{e^\varepsilon + 1}\right) + \varepsilon\sqrt{2k^\star\ln(1/\delta')}, \frac{k^\star\varepsilon^2}{2} + \varepsilon\sqrt{\frac{1}{2}k^\star\ln(1/\delta')} \right\}. \tag{3}$$

---

**Algorithm 2** $\texttt{multiLimitDom}^{k^\star, \ell^\star}$; Multiple queries to random threshold

---

**Input:** An adaptive stream of histograms $\mathbf{h}_1, \mathbf{h}_2, ....,$ fixed integers $k^\star$ and $\ell^\star$, along with per iterate privacy parameters $\varepsilon, \delta$.
**Output:** Sequence of outputs $(o_1, \cdots, o_\ell)$ for $\ell \le \ell^\star$.
**while** $k^\star > 0$ and $\ell^\star > 0$ **do**
     Based on previous outcomes, select adaptive histogram $\mathbf{h}_i$ and parameters $k_i, \bar{k}_i$
     **if** $k_i \le k^\star$ **then**
         Let $o_i = \texttt{LimitDom}^{k_i, \bar{k}_i}(\mathbf{h}_i)$ with privacy parameters $\varepsilon$ and $\delta$
         $k^\star \leftarrow k^\star - |o_i|$ and $\ell^\star \leftarrow \ell^\star - 1$
   Return $o = (o_1, o_2, \cdots)$

---

### Extensions

We further consider the restricted sensitivity setting, where any individual can change at most $\Delta$ counts. Algorithm 1 allowed for a smaller additive factor of $\ln(\Delta/\delta)/\varepsilon$ on the threshold for this setting, but the privacy loss for $\varepsilon$ was still in terms of $k$. The primary reason for this is that, unlike $\texttt{Lap}$ noise, adding $\texttt{Gumbel}$ noise to a value and releasing this estimate is not differentially private. Accordingly, if we instead run Algorithm 1 with $\texttt{Lap}$ noise, then we can achieve a $\Delta$ dependency on the $\varepsilon$. We note that adding $\texttt{Lap}$ noise instead will not allow us to provably achieve the same guarantees as Theorem 1, and we discuss some of the intuition for this later.

**Lemma 3.2** (Informal). *If we instead add $\texttt{Lap}$ noise to Algorithm 1, and we have $\Delta$-restricted sensitivity where $\Delta < k$, then we can obtain $(\Delta\varepsilon, (e^{\varepsilon\Delta} + 1)\bar{\delta})$-DP where $\bar{\delta} = \frac{\delta}{4} \cdot (3 + \ln(\Delta/\delta))$*

In addition, we give a slight variant of Algorithm 1 in Section 6.2 that will achieve the same privacy guarantees at the cost of some generality, but will be even more practical for implementation.

### Improved Advanced Composition

We also provide a result that may be of independent interest. In Section 4, we consider a slightly tighter characterization of pure ($\delta = 0$) differential privacy, which we refer to as *range-bounded*, and show that it can improve upon the total privacy loss over a sequence of adaptively chosen private algorithms. In particular, we consider the exponential mechanism, which is known to be $\varepsilon$-DP, and show that it has even stronger properties that allow us to show it is $\varepsilon$-*range-bounded* under the same parameters. Accordingly, we can then give improved advanced composition bounds for exponential mechanism compared to the optimal composition bounds for general $\varepsilon$-DP mechanisms given in [19, 26] (we show a comparison of these bounds in Appendix A).

## 3.1 Accuracy Comparisons

In contrast to previous work in top-$k$ selection subject to DP, our algorithms can return fewer than $k$ indices. Typically, accuracy in this setting is to return a set of exactly $k$ indices such that each returned index has a count that is at least the $k$-th ranked value minus some small amount. There are known lower bounds for this definition of accuracy [2] that are tight for the exponential mechanism. Relaxing the utility statement to allow returning fewer than $k$ indices, we can show that our algorithm will achieve asymptotically better accuracy where $d$ is replaced with $\bar{k}$ because

our algorithm is essentially privately determining the top-$k$ on the true top-$\bar{k}$ instead of top-$d$. In fact, if we set $\bar{k} = k$, then we will only output indices in the top-$k$ and achieve perfect accuracy, but it is critically important to note that we are unlikely to output all $k$ indices in this parameter setting. We then provide additional conditions under which we output $k$ indices with probability at least $1 - \beta$ (these formal accuracy statements are encompassed in Lemma 8.1). This condition requires a certain distance between $h_{(k)}$ and $h_{(\bar{k}+1)}$, which is comparable to the requirement for determining $\bar{k}$ for privately outputting top-$k$ itemsets in [5], and we achieve similar accuracy guarantees under this condition. The key difference becomes that for some histograms $\bar{k}$ can be as large as $d$ and hence less efficient for the algorithm in [5], but it will always return $k$ indices. Conversely, for those same histograms we maintain computational efficiency because our $\bar{k}$ is a fixed parameter, but our routine will most likely output fewer than $k$ indices.

Even for those histograms in which we are unlikely to return $k$ indices, we see this as the primary advantage of our *pay-what-you-get* composition. If there are a lot of counts that are similar to the $\bar{k}$-th ranked value, our algorithm will simply return a single $\perp$ rather than a random permutation of these indices, and the analyst need only pay for a single $\perp$ outcome rather than for up to $k$ indices in this random permutation. Essentially, the indices that are returned are normally the clear winners, i.e. indices with counts substantially above the $(\bar{k}+1)$th value, and then the $\perp$ value is informative that the remaining values are approximately equal where the analyst only has to pay for this single output as opposed to paying for the remaining outputs that are close to a random permutation. We see this as an added benefit to allowing the algorithm to return fewer than $k$ indices.

## 3.2   Our Techniques

The primary difficulty with ensuring differential privacy in our setting is that initially taking the true top-$\bar{k}$ indices will lead to different domains for neighboring histograms. More explicitly, the indices within the top-$\bar{k}$ can change by adding or removing one user's data, and this makes ensuring pure differential privacy impossible. However, the key insight will be that only indices whose value is within 1 of $h_{(\bar{k}+1)}$, the value of the $(\bar{k}+1)$th index, can go in or out of the top-$\bar{k}$ by adding or removing one user's data. Accordingly, the noisy threshold that we add will be explicitly set such that for indices with value within 1 of $h_{(\bar{k}+1)}$, the noisy estimate exceeding the noisy threshold will be a low probability event. By restricting our output set of indices to those whose noisy estimate are in the top-$k$ **and** exceed the noisy threshold, we ensure that indices in the top-$\bar{k}$ for one histogram but not in a neighboring histogram will output with probability at most $\frac{\delta}{\min\{\Delta, \bar{k}\}}$. A union bound over the total possible indices that can change will then give our desired bound on these bad events.

We now present the high level reasoning behind the proof of privacy in Theorem 1.

1. Adding `Gumbel` noise and taking the top-$k$ in one-shot is equivalent to iteratively choosing the subsequent index using the exponential mechanism with peeling, see Lemma 4.2.[2]

2. To get around the fact that the domains can change in neighboring datasets, we define a variant of Algorithm 1 that takes a histogram and a domain as input. We then prove that

this variant is DP for any input domain, see Corollary 5.1, and for a choice of domain that depends on the input histogram, it is the same as Algorithm 1, see Lemma 5.4

3. Due to the choice of the count for element $\perp$, we show that for any given neighboring datasets $\mathbf{h}, \mathbf{h}'$, the probability that Algorithm 1 evaluated on $\mathbf{h}$ can return any element that is not part of the domain with $\mathbf{h}'$ occurs with probability $\delta$, see Lemma 5.5.

We now present an overview of the analysis for the pay-what-you-get composition bound in Theorem 2.

1. Because Algorithm 1 can be expressed as multiple iterations of the exponential mechanism, we can string together many calls to Algorithm 1 as an adaptive sequence of DP mechanisms.

2. With multiple calls to Algorithm 1, if we ever get a $\perp$ outcome, we can simply start a new top-$k$ query and hence a new sequence of exponential mechanism calls. Hence, we do not need to get $k$ outcomes before we switch to a new query.

3. To get the improved constants in (3), compared to advanced composition given in Theorem 3 [13], we introduce a tigher *range-bounded* characterization, which the exponential mechanism satisfies, that enjoys better composition, see Lemma 4.4.

# 4   Existing DP Algorithms and Extensions

We now cover some existing differentially private algorithms and extensions to them. We start with the exponential mechanism [25], and show how it is equivalent to adding noise from a particular distribution and taking the argmax outcome. Next, we will present a stronger privacy condition than differential privacy which will then lead to improved composition theorems than the optimal composition theorems [19, 26] for general DP.

Throughout, we will make use of the following composition theorem in differential privacy.

**Theorem 3** (Composition [10, 13] with improvements by [19, 26]). *Let* $\mathcal{M}_1, \mathcal{M}_2, \cdots, \mathcal{M}_t$ *be each* $(\varepsilon_i, \delta_i)$-*DP, where the choice of* $\mathcal{M}_i$ *may depend on the previous outcomes of* $\mathcal{M}_1, \cdots, \mathcal{M}_{i-1}$, *then the composed algorithm* $\mathcal{M}(\mathbf{x}) = (\mathcal{M}_1(\mathbf{x}), \mathcal{M}_2(\mathbf{x}), \cdots, \mathcal{M}_t(\mathbf{x}))$ *is* $(\varepsilon'(\delta'), \sum_{i=1}^{t} \delta_i + \delta')$-*DP for any* $\delta' \geq 0$ *where*

$$\varepsilon'(\delta') = \min \left\{ \sum_{i=1}^{t} \varepsilon_i, \sum_{i=1}^{t} \varepsilon_i \cdot \left( \frac{e^{\varepsilon_i} - 1}{e^{\varepsilon_i} + 1} \right) + \sqrt{2 \sum_{i=1}^{t} \varepsilon_i^2 \ln(1/\delta')} \right\}.$$

## 4.1   Exponential Mechanism and Gumbel Noise

The exponential mechanism takes a quality score $q : \mathcal{X} \times \mathcal{Y} \to \mathbb{R}$ and can be thought of as evaluating how good $q(\mathbf{x}, y)$ is for an outcome $y \in \mathcal{Y}$ on dataset $\mathbf{x}$. For our setting, we will be using the following quality score $q(\mathbf{h}, i) = h_i$ in the exponential mechanism.

**Definition 4.1** (Exponential Mechanism). *Let* $EM_q : \mathcal{X} \to \mathcal{Y}$ *be a randomized mapping where for all outputs* $y \in \mathcal{Y}$ *we have*

$$\Pr[EM_q(\mathbf{x}) = y] \propto \exp(\tfrac{\varepsilon}{\Delta(q)} q(\mathbf{x}, y))$$

where $\Delta(q)$ is the sensitivity of the quality score, i.e. for all neighboring inputs $\mathbf{x}, \mathbf{x}'$ we have $\sup_{y \in \mathcal{Y}} |q(\mathbf{x}, y) - q(\mathbf{x}', y)| \leq \Delta(q)$

We say that a quality score $q(\cdot, \cdot)$ is monotonic in the dataset if the addition of a data record can either increase (decrease) or remain the same with any outcome, e.g. $q(\mathbf{x}, y) \leq q(\mathbf{x} \cup \{x_i\}, y)$ for any input and outcome $y$. Note that $q(\mathbf{h}, i) = h_i$ is monotonic in the dataset. We then have the following privacy guarantee.

**Lemma 4.1.** *The exponential mechanism $\mathtt{EM}_q$ is $2\varepsilon$-DP. Further, if $q$ is monotonic in the dataset, then $\mathtt{EM}_q$ is $\varepsilon$-DP.*

We point out that the exponential mechanism can be simulated by adding Gumbel noise $\mathtt{Gumbel}(\Delta(q)/\varepsilon)$ to each quality score value and then reporting the outcome with the largest noisy count. This is similar to the report noisy max mechanism [10] except Gumbel noise is added rather than Laplace. We define $\mathtt{pEM}_q^k$ to be the iterative peeling algorithm that first samples the outcome with the largest quality score then repeats on the remaining outcomes and continues $k$ times. We further define $\mathcal{M}_{\mathtt{Gumbel}}^k(\mathbf{q}(\mathbf{x}))$ to be the algorithm that adds $\mathtt{Gumbel}(\Delta(q)/\varepsilon)$ to each $q(\mathbf{x}, y)$ for $y \in \mathcal{Y}$ and takes the $k$ indices with the largest noisy counts. We then make the following connection between $\mathtt{pEM}^k$ and $\mathcal{M}_{\mathtt{Gumbel}}^k$, so that we can compute the top-$k$ outcomes in one-shot. We defer the proof to Appendix B.1

**Lemma 4.2.** *For any input $\mathbf{x} \in \mathcal{X}$ the peeling exponential mechanism $\mathtt{pEM}_q^k(\mathbf{x})$ is equal in distribution to $\mathcal{M}_{\mathtt{Gumbel}}^k(\mathbf{q}(\mathbf{x}))$. That is for any outcome vector $(o_1, \cdots, o_k) \in [d]^k$ we have*

$$\Pr[\mathtt{pEM}_q^k(\mathbf{x}) = (o_1, \cdots, o_k)] = \Pr[\mathcal{M}_{\mathtt{Gumbel}}^k(\mathbf{q}(\mathbf{x})) = (o_1, \cdots, o_k)]$$

We next show that the one-shot noise addition is ($\approx \sqrt{k}\varepsilon, \delta$)-DP using Theorem 3. Previous work [14] considered a one-shot approach for top-$k$ selection subject to DP with Laplace noise addition and in order to get the $\sqrt{k}\varepsilon$ factor on the privacy loss, their algorithm could not return the ranked list of indices. Using Gumbel noise allows us to return the ranked list of indices in one-shot with the same privacy loss.

**Corollary 4.1.** *The one-shot $\mathcal{M}_{\mathtt{Gumbel}}^k(\mathbf{q}(\cdot))$ is $(\varepsilon', \delta)$-DP for any $\delta \geq 0$ where*

$$\varepsilon' = \min\left\{2k\varepsilon, 2k\varepsilon \cdot \left(\frac{e^{2\varepsilon} - 1}{e^{2\varepsilon} + 1}\right) + 2\varepsilon\sqrt{2k\ln(1/\delta)}\right\}.$$

*Further, if the quality score $q$ is monotonic in the dataset, then $\mathcal{M}_{\mathtt{Gumbel}}^k(\mathbf{q}(\cdot))$ is also $(\varepsilon'', \delta)$-DP for any $\delta \geq 0$ where*

$$\varepsilon'' = \min\left\{k\varepsilon, k\varepsilon \cdot \left(\frac{e^{\varepsilon} - 1}{e^{\varepsilon} + 1}\right) + \varepsilon\sqrt{2k\ln(1/\delta)}\right\}.$$

## 4.2 Bounded Range Composition

It turns out that we can actually improve on the total privacy loss for this algorithm and for a wider class of algorithms in general. We first define a slightly stronger condition than (pure) differential privacy that can give a tighter characterization of the privacy loss for certain DP mechanisms.

**Definition 4.2** (Range-Bounded)**.** *Given a mechanism $\mathcal{M}$ that takes a collection of records in $\mathcal{X}$ to outcome set $\mathcal{Y}$, we say that $\mathcal{M}$ is $\varepsilon$-range-bounded if for any $y, y' \in \mathcal{Y}$ and any neighboring databases $\mathbf{x}, \mathbf{x}'$ we have*

$$\frac{\Pr[\mathcal{M}(\mathbf{x}) = y]}{\Pr[\mathcal{M}(\mathbf{x}') = y]} \leq e^{\varepsilon} \frac{\Pr[\mathcal{M}(\mathbf{x}) = y']}{\Pr[\mathcal{M}(\mathbf{x}') = y']}$$

*where we use the probability density function instead for continuous outcome spaces.* [3]

It is straightforward to see that this definition is within a factor of 2 of standard differential privacy.

**Corollary 4.2.** *If a mechanism $\mathcal{M}$ is $\varepsilon$-range-bounded, then it is also $\varepsilon$-DP and conversely if $\mathcal{M}$ is $\varepsilon$-DP then it is also $2\varepsilon$-range-bounded. Furthermore, if $\mathcal{M}$ is $\varepsilon$-range-bounded, then we have*

$$\sup_{y \in \mathcal{Y}} \ln\left(\frac{\Pr[\mathcal{M}(\mathbf{x}) = y]}{\Pr[\mathcal{M}(\mathbf{x}') = y]}\right) - \inf_{y' \in \mathcal{Y}} \ln\left(\frac{\Pr[\mathcal{M}(\mathbf{x}) = y']}{\Pr[\mathcal{M}(\mathbf{x}') = y']}\right) \leq \varepsilon$$

The final consequence is exactly where our *range-bounded* terminology comes from because this implies that for any $y \in \mathcal{Y}$ there is some fixed $t \in [0, \varepsilon]$ such that

$$\ln\left(\frac{\Pr[\mathcal{M}(\mathbf{x}) = y]}{\Pr[\mathcal{M}(\mathbf{x}') = y]}\right) \in [-t, \varepsilon - t].$$

In contrast, $\varepsilon$-DP only guarantees that for any $y \in \mathcal{Y}$

$$\ln\left(\frac{\Pr[\mathcal{M}(\mathbf{x}) = y]}{\Pr[\mathcal{M}(\mathbf{x}') = y]}\right) \in [-\varepsilon, \varepsilon]$$

where we know that this range is tight for some mechanisms such as randomized response, which was the mechanism used for proving optimal advanced composition bounds [19, 26]. However, for other mechanisms this range is too loose. For the exponential mechanism, constructing worst-case neighboring databases such that some output's probability increases by a factor of about $e^{\varepsilon}$ requires the quality score of that output to increase and all other quality scores to decrease, which implies that their output probability remains about the same. We then show that exponential mechanism achieves the same privacy parameters as in Lemma 4.1 for our stronger charaterization.

**Lemma 4.3.** *The exponential mechanism $\mathtt{EM}_q$ is $2\varepsilon$-range-bounded, furthermore if $q$ is monotonic in its dataset then $\mathtt{EM}_q$ is $\varepsilon$-range bounded.*

*Proof.* Consider any outcomes $y, y' \in \mathcal{Y}$, and take any neighboring inputs $\mathbf{x}$ and $\mathbf{x}'$.

Plugging in the specific forms of these probabilities, it is straightforward to see that the denominators will cancel and we are left with the following with the substitution $\varepsilon_q = \frac{\varepsilon}{\Delta(q)}$

$$\frac{\Pr[\mathtt{EM}_q(\mathbf{x}) = y]}{\Pr[\mathtt{EM}_q(\mathbf{x}') = y]} \frac{\Pr[\mathtt{EM}_q(\mathbf{x}') = y']}{\Pr[\mathtt{EM}_q(\mathbf{x}) = y']} = \frac{\exp(\varepsilon_q q(\mathbf{x}, y))}{\exp(\varepsilon_q q(\mathbf{x}', y))} \frac{\exp(\varepsilon_q q(\mathbf{x}', y'))}{\exp(\varepsilon q(\mathbf{x}, y'))} \leq e^{2\varepsilon}.$$

When $q$ is monotonic in the dataset, we have either the case where $\frac{\exp(\varepsilon_q q(\mathbf{x},y))}{\exp(\varepsilon_q q(\mathbf{x}',y))} \leq e^{\varepsilon}$ and $\frac{\exp(\varepsilon_q q(\mathbf{x}',y'))}{\exp(\varepsilon_q q(\mathbf{x},y'))} \leq 1$ or the case where $\frac{\exp(\varepsilon_q q(\mathbf{x},y))}{\exp(\varepsilon_q q(\mathbf{x}',y))} \leq 1$ and $\frac{\exp(\varepsilon_q q(\mathbf{x}',y'))}{\exp(\varepsilon_q q(\mathbf{x},y'))} \leq e^{\varepsilon}$. Hence the factor of 2 savings in the privacy parameter.

$\square$

We now show that we can achieve a better composition bound when we compose $\varepsilon$-range-bounded algorithms as opposed to using Theorem 3, which applies to the composition of general DP algorithms. Intuitively this composition will save a factor of 2 because the range that will maximize the variance is $[-\frac{\varepsilon}{2}, \frac{\varepsilon}{2}]$ due to the fact that if the range was instead skewed towards $\varepsilon$ (i.e. a range of $[-o(1), \varepsilon - o(1)]$) then almost all of the probability mass has to be on events with log-ratio around $-o(1)$. Rather than using Azuma's inequality on the sum of the privacy losses, as is done in the original advanced composition paper [13], we use the more general Azuma-Hoeffding bound.

**Theorem 4** (Azuma-Hoeffding[4]). *Let $(X_t)$ be a martingale with respect to the filtration $(\mathcal{F}_t)$. Assume that there exist $\mathcal{F}_{t-1}$ measurable variables $A_t, B_t$ and a constant $c_t$ such that*

$$A_t \leq X_t - X_{t-1} \leq B_t \qquad B_t - A_t \leq c_t.$$

*Then for any $\beta > 0$ we have*

$$\Pr[Z_t - Z_0 \geq \beta] \leq \exp\left(\frac{-2\beta^2}{\sum_{i=1}^t c_i^2}\right).$$

In fact, our composition bound for range-bounded algorithms improves on the *optimal* composition theorem for general DP algorithms [19, 26]. See Appendix A for a comparison of the different bounds. We defer the proof, which largely follows a similar argument to [13], to Appendix B.2.

**Lemma 4.4.** *Let $\mathcal{M}_1, \mathcal{M}_2, \cdots, \mathcal{M}_t$ each be $\varepsilon_i$-bounded range where the choice of $\mathcal{M}_i$ may depend on the previous outcomes of $\mathcal{M}_1, \cdots, \mathcal{M}_{i-1}$, then the composed algorithm $\mathcal{M}(\mathbf{x})$ of each of the algorithms $\mathcal{M}_1(\mathbf{x}), \mathcal{M}_2(\mathbf{x}), \cdots, \mathcal{M}_t(\mathbf{x})$ is $(\varepsilon''(\delta), \delta)$-DP for any $\delta \geq 0$ where*

$$\varepsilon''(\delta) = \min\left\{\sum_{i=1}^t \varepsilon_i, \sum_{i=1}^t \varepsilon_i \cdot \left(\frac{e^{\varepsilon_i} - 1}{e^{\varepsilon_i} + 1}\right) + \sqrt{2\sum_{i=1}^t \varepsilon_i^2 \ln(1/\delta)}, \sum_{i=1}^t \frac{\varepsilon_i^2}{2} + \sqrt{\frac{1}{2}\sum_{i=1}^t \varepsilon_i^2 \ln(1/\delta)}\right\}. \quad (4)$$

Note that in order to see an improvement in the advanced composition bound, we do not necessarily require that an $\varepsilon$-DP mechanism is also $\varepsilon$-*range-bounded*, but could be relaxed to showing it is $\alpha\varepsilon$-*range-bounded* for some $\alpha < 2$. In particular, this will still give improvements with respect to the simpler formulation of the advanced composition bound. More specifically, the significant term that is normally considered in advanced composition is $\sqrt{2k\ln(1/\delta')}\varepsilon$, which can be replaced with $\sqrt{\frac{\alpha^2}{2}k\ln(1/\delta')}\varepsilon$ for composing $\alpha\varepsilon$-*range-bounded* mechanisms with $\alpha \leq 2$. Consequently, we believe that this formulation could be useful for mechanisms beyond the exponential mechanism.

# 5 Limited Domain Algorithm

In this section we present the analysis of our main procedure in Algorithm 1. We begin by giving basic properties of histograms when an individual's data is added or removed, and how this can change the domain of the true top-$\bar{k}$. This will be critical for achieving our bounds on the bad events when an index moves in or out of the true top-$\bar{k}$ for a neighboring database. Next, we will give an alternative formulation of our algorithm based upon a peeling exponential mechanism. The general idea will be to show that once we have bounded the probability of outputting indices unique to the true top-$\bar{k}$ of one on the neighboring histograms, then we can just consider the remaining similar outputs according to this peeling exponential mechanism and bound this in terms of pure differential privacy. Finally, we will provide a proof of Theorem 1.

## 5.1 Properties of Data Dependent Thresholds

In this section we will cover basic properties of how the domain of elements above a data dependent threshold can change in neighboring histograms, i.e. $\mathbf{h}$ and $\mathbf{h}'$, where $||\mathbf{h} - \mathbf{h}'||_\infty \leq 1$. In our algorithm, we will use a data dependent threshold, such as the $\bar{k}$-th ordered count $h_{(\bar{k})}$. Our first property that we use often within our analysis is that the count of the $\bar{k}$th largest histogram value will not change by more than one (even though the index itself may change).

**Lemma 5.1.** *For any neighboring histograms $\mathbf{h}, \mathbf{h}'$, where w.l.o.g. $\mathbf{h} \geq \mathbf{h}'$, and for any $\bar{k} \leq d$, we must have either $h_{(\bar{k})} = h'_{(\bar{k})}$ or $h_{(\bar{k})} = h'_{(\bar{k})} + 1$.*

*Proof.* Let $i'_{(j)}$ be the index for $h'_{(j)}$. By assumption we have $\mathbf{h} \geq \mathbf{h}'$, which implies that for each index $i'_{(j)}$ we must have $h_{i'_{(j)}} \geq h'_{i'_{(j)}}$. Therefore, for each $j \leq \bar{k}$, we have $h_{i'_{(j)}} \geq h'_{i'_{(\bar{k})}}$, which implies $h_{(\bar{k})} \geq h'_{(\bar{k})}$.

Similarly, we let $i_{(j)}$ be the index for $h_{(j)}$, and we know that $\mathbf{h}$ and $\mathbf{h}'$ are neighboring so for each index $i_{(j)}$ we must have $h'_{i_{(j)}} + 1 \geq h_{i_{(j)}}$. Therefore, for each $j \leq \bar{k}$, we have $h'_{i_{(j)}} + 1 \geq h_{i_{(\bar{k})}}$, which implies $h'_{(\bar{k})} + 1 \geq h_{(\bar{k})}$. □

Instead of considering the entire domain of size $d$, our algorithms will be limited to a much smaller domain $\mathbf{d}^{\bar{k}}(\mathbf{h})$ for each database and a given value $\bar{k}$, where

$$\mathbf{d}^{\bar{k}}(\mathbf{h}) := \{i_{(j)} \in [d] : j \leq \bar{k} \text{ and } h_{i_{(1)}} \geq h_{i_{(2)}} \geq \cdots \geq h_{i_{(d)}}\}. \tag{5}$$

and assume that there is some arbitrary (data-independent) tie-breaking that occurs for ordering the histograms. We then have the following result, which bounds how much the change in counts between neighboring databases can be on elements that are in the set difference of the two domains.

**Lemma 5.2.** *For any $\Delta$-restricted sensitivity neighboring histograms $\mathbf{h}, \mathbf{h}'$, and some fixed $\bar{k} < d$, if $i \in \mathbf{d}^{\bar{k}}(\mathbf{h}) \setminus \mathbf{d}^{\bar{k}}(\mathbf{h}')$ then $h_i \leq h_{(\bar{k}+1)} + 1$ and if $i \in \mathbf{d}^{\bar{k}}(\mathbf{h}') \setminus \mathbf{d}^{\bar{k}}(\mathbf{h})$ then $h'_i \leq h'_{(\bar{k}+1)} + 1$*

*Proof.* If $i \in \mathbf{d}^{\bar{k}}(\mathbf{h}) \setminus \mathbf{d}^{\bar{k}}(\mathbf{h}')$, then $h'_i \leq h'_{(\bar{k}+1)}$ because $i \notin \mathbf{d}^{\bar{k}}(\mathbf{h}')$. We first consider the case $\mathbf{h} \geq \mathbf{h}'$, which implies $h'_{(\bar{k}+1)} \leq h_{(\bar{k}+1)}$ by Lemma 5.1 and because they are neighbors, we must have $h_i \leq h'_i + 1$. Therefore, $h_i \leq h'_i + 1 \leq h'_{(\bar{k}+1)} + 1 \leq h_{(\bar{k}+1)} + 1$ as desired. If instead $\mathbf{h}' \geq \mathbf{h}$, then again by Lemma 5.1 we have $h'_{(\bar{k}+1)} \leq h_{(\bar{k}+1)} + 1$, and we must also have $h_i \leq h'_i$. Therefore, $h_i \leq h'_i \leq h'_{(\bar{k}+1)} \leq h_{(\bar{k}+1)} + 1$ as desired.

The other claim follows symmetrically.

$\square$

We now show that the set difference between the domain under $\mathbf{h}$ and $\mathbf{h}'$ is no more than $\bar{k}$ and the restricted sensitivity of the neighboring histograms

**Lemma 5.3.** *For any $\Delta$-restricted sensitivity neighboring histograms $\mathbf{h}, \mathbf{h}'$, and some fixed $\bar{k} < d$, then we must have*

$$|\mathbf{d}^{\bar{k}}(\mathbf{h}) \setminus \mathbf{d}^{\bar{k}}(\mathbf{h}')| \leq \min\{\Delta, \bar{k}, d - \bar{k}\}.$$

*Proof.* By definition, we have $|\mathbf{d}^{\bar{k}}(\mathbf{h}) \setminus \mathbf{d}^{\bar{k}}(\mathbf{h}')| \leq \min\{\bar{k}, d - \bar{k}\}$, so we will assume $\Delta < \bar{k}$ and show for $\Delta$. We assume w.l.o.g. that $\mathbf{h} \geq \mathbf{h}'$, and because we know by construction that $|\mathbf{d}^{\bar{k}}(\mathbf{h})| = \bar{k}$ for any $\mathbf{h}$, then proving $|\mathbf{d}^{\bar{k}}(\mathbf{h}) \setminus \mathbf{d}^{\bar{k}}(\mathbf{h}')| \leq \Delta$ will imply $|\mathbf{d}^{\bar{k}}(\mathbf{h}') \setminus \mathbf{d}^{\bar{k}}(\mathbf{h})| \leq \Delta$. It now suffices to show that for any $i \in \mathbf{d}^{\bar{k}}(\mathbf{h}) \setminus \mathbf{d}^{\bar{k}}(\mathbf{h}')$ we must have $h_i > h'_i$. If $h_i = h'_i$ then the position of index $i$ cannot have moved up the ordering from $\mathbf{h}'$ to $\mathbf{h}$ because we assumed $\mathbf{h} \geq \mathbf{h}'$. Therefore, if $i \notin \mathbf{d}^{\bar{k}}(\mathbf{h}')$ and $h_i = h'_i$ we must also have $i \notin \mathbf{d}^{\bar{k}}(\mathbf{h})$.

$\square$

These properties will ultimately be critical in bounding the probability of indices outside of $\mathbf{d}^{\bar{k}}(\mathbf{h}) \cap \mathbf{d}^{\bar{k}}(\mathbf{h}')$ being output. Note that we typically think of $\bar{k} \ll d$, so we will eliminate $d - \bar{k}$ from the minimum statement in Lemma 5.3 throughout the rest of the analysis.

## 5.2   Limited Domain Peeling Exponential Mechanism

Our main procedure $\texttt{LimitDom}^{k,\bar{k}}$ is given in Algorithm 1, which involves adding Gumbel noise to each of the top-$\bar{k}$ terms in the histogram we are given where $\bar{k} \geq k$. Note that from Section 4 we know that our analysis can be done by considering the exponential mechanism instead of noise addition.

We now generalize the exponential mechanism we presented in Section 4.

**Definition 5.1** (Limited Histogram Exponential Mechanism)**.** *We define the Limited Histogram Exponential Mechanism for any $\bar{k} \leq d$ to be $LEM^{\bar{k}} : \mathbb{N}^d \times 2^{[d]} \to [d] \cup \{\perp\}$ such that*

$$\Pr[LEM^{\bar{k}}(\mathbf{h}, \mathbf{d}) = i] = \frac{\exp(\varepsilon h_i)}{\exp(\varepsilon h_\perp) + \sum_{j \in \mathbf{d}} \exp(\varepsilon h_j)}$$

*for all $i \in \mathbf{d} \cup \{\perp\}$ where $\mathbf{d} \subseteq [d]$ and*

$$h_\perp = h_{(\bar{k}+1)} + 1 + \ln(\min\{\bar{k}, \Delta\}/\delta)/\varepsilon. \tag{6}$$

From Lemma 4.3 we then have the following result due to the fact that the exponential mechanism is $\varepsilon$-DP and we are simply adding a new coordinate $\perp$ with count $h_{(\bar{k}+1)} + 1 + \ln(\min\{\bar{k}, \Delta\}/\delta)/\varepsilon$.

**Corollary 5.1.** *For any fixed $\mathbf{d} \subseteq [d]$ then $LEM^{\bar{k}}(\cdot, \mathbf{d})$ is $\varepsilon$-range bounded and $\varepsilon$-DP.*

In order to make our peeling algorithm $\texttt{pLEM}^{k,\bar{k}}$ in Algorithm 3 equivalent to $\texttt{LimitDom}^{k,\bar{k}}$ in Algorithm 1, we will need to iterate over the same set of indices. Recall how we defined the limited domain $\mathbf{d}^k(\mathbf{h})$ in (5).

**Lemma 5.4.** *For any input histogram* $\mathbf{h}$, *LimitDom*$^{k,\bar{k}}(\mathbf{h})$ *and* $pLEM^{k,\bar{k}}(\mathbf{h}, \mathbf{d}^{\bar{k}}(\mathbf{h}))$ *are equal in distribution.*

*Proof.* Note that both $\texttt{LimitDom}^{k,\bar{k}}(\mathbf{h})$ and $\texttt{pLEM}^{k,\bar{k}}(\mathbf{h}, \mathbf{d}^{\bar{k}}(\mathbf{h}))$ will only consider terms in $\mathbf{d}^{\bar{k}}(\mathbf{h})$ to add to the output. We showed in Lemma 4.2 that adding Gumbel noise to all counts in a histogram, in this case $(h_j : j \in \mathbf{d}^{\bar{k}}(\mathbf{h}) \cup \{\bot\})$, and taking the largest $k$ is equivalent to peeling the exponential mechanism to return the largest count $k$ times. Lastly, if we select $\bot$ as one of the indices, then we do not return any other indices with smaller count than $h_\bot$. $\qquad\square$

---

**Algorithm 3** $\texttt{pLEM}^{k,\bar{k}}$; Peeling Exponential Mechanism version of Algorithm 1

---

    **Input:** Histogram $\mathbf{h}$, subset $\mathbf{d} \subseteq [d]$ of indices; privacy parameters $\varepsilon, \delta$.
    **Output:** Ordered set of indices.
    Set $\mathcal{I} = \emptyset$
    **while** $|\mathcal{I}| < k$ **do**
        Set $o = \texttt{LEM}^{\bar{k}}(\mathbf{h}, \mathbf{d} \setminus \mathcal{I})$
        **if** $o = \bot$ **then**
            Let $\mathcal{I} \leftarrow \mathcal{I} \cup \{o\}$ #concatenate $o$ to $\mathcal{I}$ to retain the order
            Return $\mathcal{I}$
        **else**
            Let $\mathcal{I} \leftarrow \mathcal{I} \cup \{o\}$ #concatenate $o$ to $\mathcal{I}$ to retain the order
    Return $\mathcal{I}$

---

**Corollary 5.2.** *For any fixed* $\mathbf{d} \subseteq [d]$ *and neighboring histograms* $\mathbf{h}, \mathbf{h}'$, *we have that* $pLEM^{k,\bar{k}}(\cdot, \mathbf{d})$ *is* $(\varepsilon', \delta')$ *for any* $\delta' \geq 0$ *where* $\varepsilon'$ *is given in* (2).

We will now fix two neighboring histograms $\mathbf{h}, \mathbf{h}'$, and separate out our outcome space into bad events for $\mathbf{h}$ and $\mathbf{h}'$. In particular, these will just be outputs that contain some index in the top-$\bar{k}$ for one, but not in the top-$\bar{k}$ for the other.

**Definition 5.2.** *For any neighboring histograms* $\mathbf{h}, \mathbf{h}'$, *then we define* $\mathcal{S}$ *as the outcome set of* $pLEM^{k,\bar{k}}(\mathbf{h}, \mathbf{d}^{\bar{k}}(\mathbf{h}))$ *and the outcome set of* $pLEM^{k,\bar{k}}(\mathbf{h}', \mathbf{d}^{\bar{k}}(\mathbf{h}'))$ *as* $\mathcal{S}'$.
    *We then define the* bad outcomes *as*

$$\mathcal{S}^\delta := \mathcal{S} \setminus \mathcal{S}' \qquad and \qquad \mathcal{S}'^\delta := \mathcal{S}' \setminus \mathcal{S}$$

The bulk of the heavy lifting will then be done by the following two lemmas that bound the bad events, and also give a simpler way to compare the good events in terms of pure differential privacy. For bounding the bad events, we need to upper bound the probability of outputting an index in $\mathbf{d}^k(\mathbf{h}) \setminus \mathbf{d}^k(\mathbf{h}')$. If we consider one call to the exponential mechanism, then we could obtain an upper bound on the probability of outputting a given index in $\mathbf{d}^k(\mathbf{h}) \setminus \mathbf{d}^k(\mathbf{h}')$, by restricting the possible outputs to just that index and $\bot$. This will then give us the bound of $\delta$. However, applying this over the possible $k$ iterative calls will give a bound of $k\delta$, so we will instead need a slightly more sophisticated argument that accounts for the fact that the iterative process terminates whenever $\bot$ is output.

**Lemma 5.5.** *For any neighboring histograms* $\mathbf{h}, \mathbf{h}'$,

$$\Pr[pLEM^{k,\bar{k}}(\mathbf{h}, \mathbf{d}^{\bar{k}}(\mathbf{h})) \in \mathcal{S}^\delta] \leq \delta$$

We defer the proof to Appendix C. The next lemma will give us a clean way to compare the good events, that will mainly be due to the fact that conditional probabilities are simpler to work with in the exponential mechanism. More specifically, if we consider the rejection sampling scheme of redrawing when we see a bad event, then the resulting probability distribution is actually equivalent to simply restricting our domain to $\mathbf{d}^k(\mathbf{h}) \cap \mathbf{d}^k(\mathbf{h}')$, the set of indices in the top-$\bar{k}$ for both histograms. This will then allow us to compare the probability distributions of both histograms outputting from the same domain.

**Lemma 5.6.** *For any neighboring histograms* $\mathbf{h}, \mathbf{h}'$, *such that* $\mathbf{d}^\varepsilon = \mathbf{d}^{\bar{k}}(\mathbf{h}') \cap \mathbf{d}^{\bar{k}}(\mathbf{h})$, *then we have that for any* $o \in \mathcal{S} \cap \mathcal{S}'$

$$\Pr[pLEM^{k,\bar{k}}(\mathbf{h}, \mathbf{d}^{\bar{k}}(\mathbf{h})) = o] = \Pr[pLEM^{k,\bar{k}}(\mathbf{h}, \mathbf{d}^\varepsilon) = o] \cdot \Pr[pLEM^{k,\bar{k}}(\mathbf{h}, \mathbf{d}^{\bar{k}}(\mathbf{h})) \notin \mathcal{S}^\delta]$$

We defer the proof to Appendix C. This lemma does not immediately give us pure differential privacy on outcomes in $\mathcal{S} \cap \mathcal{S}'$ because while we will be able to compare $\Pr[pLEM^{k,\bar{k}}(\mathbf{h}, \mathbf{d}^\varepsilon) = o]$ and $\Pr[pLEM^{k,\bar{k}}(\mathbf{h}', \mathbf{d}^\varepsilon) = o]$ using Corollary 5.2, we still need to account for $\Pr[pLEM^{k,\bar{k}}(\mathbf{h}, \mathbf{d}^{\bar{k}}(\mathbf{h})) \notin \mathcal{S}^\delta]$ which we know is at least $1 - \delta$. This will give us a reasonably simple way to achieve a bound of $2\delta$ on the total variation distance, but with some additional work we can eliminate the factor of two. In particular, we will use the following general result in the proof of our main result.

**Claim 5.1.** *For any* $\delta_1 \in [0, 1]$ *and* $\delta_2 \in [0, 1)$, *and any non-negative* $x \leq 1 - \delta_2$, *we have that*

$$x\frac{1 - \delta_1}{1 - \delta_2} + \delta_1 \leq x + \max\{\delta_1, \delta_2\}$$

*Proof.* Multiplying each term by $(1 - \delta_2)$ and cancelling like terms gives the equivalent inequality of

$$\delta_2 x + \delta_1(1 - \delta_2) \leq \delta_1 x + \max\{\delta_1, \delta_2\}(1 - \delta_2)$$

If $\delta_1 \geq \delta_2$, then $\delta_2 x \leq \delta_1 x$ and we are done. If $\delta_1 < \delta_2$, then our inequality reduces to

$$\delta_2 x + \delta_1(1 - \delta_2) \leq \delta_1 x + \delta_2(1 - \delta_2)$$

Rearranging terms we get this is equivalent to

$$\delta_1(1 - \delta_2 - x) \leq \delta_2(1 - \delta_2 - x)$$

which holds because we assumed $x \leq 1 - \delta_2$. $\square$

We now combine these lemmas and claim to provide our main result of this section, and we will then show how Theorem 1 immediately follows from this lemma.

**Lemma 5.7.** *For any neighboring histograms* $\mathbf{h}, \mathbf{h}'$ *and for any* $S \subseteq \mathcal{S}$, *we have that*

$$\Pr[pLEM^{k,\bar{k}}(\mathbf{h}, \mathbf{d}^{\bar{k}}(\mathbf{h})) \in S] \leq e^{\varepsilon'} \Pr[pLEM^{k,\bar{k}}(\mathbf{h}', \mathbf{d}^{\bar{k}}(\mathbf{h}')) \in S] + \delta + \delta'$$

*for any* $\delta' \geq 0$, *where* $\varepsilon' = \min\{k\varepsilon, \sqrt{2k\ln(1/\delta')}\varepsilon + k\varepsilon(e^\varepsilon - 1)/(e^\varepsilon + 1), \sqrt{\frac{k}{2}\ln(1/\delta')}\varepsilon + k\varepsilon^2/2\}$.

*Proof.* We will first separate $S$ such that $S^\delta = S \cap \mathcal{S}^\delta$ and $S^\varepsilon = S \setminus S^\delta$. For ease of notation, we will let

$$\Pr[\texttt{pLEM}^{k,\bar{k}}(\mathbf{h}, \mathbf{d}^{\bar{k}}(\mathbf{h})) \in \mathcal{S}^\delta] = \delta_{\mathbf{h}} \qquad \text{and} \qquad \Pr[\texttt{pLEM}^{k,\bar{k}}(\mathbf{h}', \mathbf{d}^{\bar{k}}(\mathbf{h}')) \in \mathcal{S}'^\delta] = \delta_{\mathbf{h}'}$$

This then implies

$$\Pr[\texttt{pLEM}^{k,\bar{k}}(\mathbf{h}, \mathbf{d}^{\bar{k}}(\mathbf{h})) \in S] = \Pr[\texttt{pLEM}^{k,\bar{k}}(\mathbf{h}, \mathbf{d}^{\bar{k}}(\mathbf{h})) \in S^\varepsilon] + \Pr[\texttt{pLEM}^{k,\bar{k}}(\mathbf{h}, \mathbf{d}^{\bar{k}}(\mathbf{h})) \in S^\delta]$$
$$\leq \Pr[\texttt{pLEM}^{k,\bar{k}}(\mathbf{h}, \mathbf{d}^{\bar{k}}(\mathbf{h})) \in S^\varepsilon] + \delta_{\mathbf{h}}$$

Applying Lemma 5.6, with $\mathbf{d}^\varepsilon = \mathbf{d}^{\bar{k}}(\mathbf{h}') \cap \mathbf{d}^{\bar{k}}(\mathbf{h})$, and the fact that $\Pr[\texttt{pLEM}^{k,\bar{k}}(\mathbf{h}, \mathbf{d}^{\bar{k}}(\mathbf{h})) \in \mathcal{S}^\delta] + \Pr[\texttt{pLEM}^{k,\bar{k}}(\mathbf{h}, \mathbf{d}^{\bar{k}}(\mathbf{h})) \notin \mathcal{S}^\delta] = 1$, we have

$$\Pr[\texttt{pLEM}^{k,\bar{k}}(\mathbf{h}, \mathbf{d}^{\bar{k}}(\mathbf{h})) \in S] \leq \Pr[\texttt{pLEM}^{k,\bar{k}}(\mathbf{h}, \mathbf{d}^\varepsilon) \in S^\varepsilon](1 - \delta_{\mathbf{h}}) + \delta_{\mathbf{h}}$$

From Corollary 5.2 we know that for $\delta' \geq 0$ and $\varepsilon'$ given in (2)

$$\Pr[\texttt{pLEM}^{k,\bar{k}}(\mathbf{h}, \mathbf{d}^\varepsilon) \in S^\varepsilon] \leq \min\{1, e^{\varepsilon'} \Pr[\texttt{pLEM}^{k,\bar{k}}(\mathbf{h}', \mathbf{d}^\varepsilon) \in S^\varepsilon] + \delta'\}$$

which implies

$$\Pr[\texttt{pLEM}^{k,\bar{k}}(\mathbf{h}, \mathbf{d}^{\bar{k}}(\mathbf{h})) \in S] \leq \min\{1, e^{\varepsilon'} \Pr[\texttt{pLEM}^{k,\bar{k}}(\mathbf{h}', \mathbf{d}^\varepsilon) \in S^\varepsilon] + \delta'\} \cdot (1 - \delta_{\mathbf{h}}) + \delta_{\mathbf{h}}$$
$$= \min\{1 - \delta_{\mathbf{h}'}, e^{\varepsilon'} \Pr[\texttt{pLEM}^{k,\bar{k}}(\mathbf{h}', \mathbf{d}^\varepsilon) \in S^\varepsilon](1 - \delta_{\mathbf{h}'}) + \delta'(1 - \delta_{\mathbf{h}'})\} \frac{1 - \delta_{\mathbf{h}}}{1 - \delta_{\mathbf{h}'}} + \delta_{\mathbf{h}}$$
$$= \min\{1 - \delta_{\mathbf{h}'}, e^{\varepsilon'} \Pr[\texttt{pLEM}^{k,\bar{k}}(\mathbf{h}', \mathbf{d}^{\bar{k}}(\mathbf{h}')) \in S^\varepsilon] + \delta'(1 - \delta_{\mathbf{h}'})\} \frac{1 - \delta_{\mathbf{h}}}{1 - \delta_{\mathbf{h}'}} + \delta_{\mathbf{h}}$$

where the last step follows from Lemma 5.6. We then apply Claim 5.1 [5] to get

$$\Pr[\texttt{pLEM}^{k,\bar{k}}(\mathbf{h}, \mathbf{d}^{\bar{k}}(\mathbf{h})) \in S] \leq \min\{1 - \delta_{\mathbf{h}'}, e^{\varepsilon'} \Pr[\texttt{pLEM}^{k,\bar{k}}(\mathbf{h}', \mathbf{d}^{\bar{k}}(\mathbf{h}')) \in S^\varepsilon] + \delta'(1 - \delta_{\mathbf{h}'})\} + \max\{\delta_{\mathbf{h}}, \delta_{\mathbf{h}'}\}$$

We further use Lemma 5.5 to bound $\max\{\delta_{\mathbf{h}}, \delta_{\mathbf{h}'}\} \leq \delta$ and obtain

$$\Pr[\texttt{pLEM}^{k,\bar{k}}(\mathbf{h}, \mathbf{d}^{\bar{k}}(\mathbf{h})) \in S] \leq e^{\varepsilon'} \Pr[\texttt{pLEM}^{k,\bar{k}}(\mathbf{h}', \mathbf{d}^{\bar{k}}(\mathbf{h}')) \in S^\varepsilon] + \delta' + \delta$$

Finally, by definition $\Pr[\texttt{pLEM}^{k,\bar{k}}_\Delta(\mathbf{h}, \mathbf{d}^{\bar{k}}(\mathbf{h})) \in S^\delta] = 0$ which implies our desired bound. $\qquad\square$

*Proof of Theorem 1.* From Lemma 5.4 we know that Algorithm 1 is equivalent to the peeling exponential mechanism. Our privacy guarantees then follow immediately from Lemma 5.7. $\qquad\square$

# 6 Variants and Improvements of the Limited Domain Algorithm

In this section, we will discuss some variants of our main algorithm and some improvements. Specifically, we discuss a variant that adds Laplace noise to the counts, which is similar to the report noisy max algorithm in Dwork and Roth [10]. This variant of report noisy max allows us to only pay a $\Delta$ factor on the $\varepsilon$ parameter in the $\Delta$-restricted sensitivity setting. We then present a more practical version of our main algorithm only considers domain elements that are strictly greater than the $(\bar{k}+1)$-th value, so it has cardinality at most $\bar{k}$. We then present a way to optimize the threshold value $\bar{k}$ in a data dependent way.

## 6.1 Laplace Limited Domain Algorithm

We now restrict ourselves to the case in which neighboring histograms can vary in at most $\Delta$ positions so that $||\mathbf{h}-\mathbf{h}'||_\infty \leq 1$ and $||\mathbf{h}-\mathbf{h}'||_0 \leq \Delta$, i.e. $\mathbf{h}, \mathbf{h}'$ are $\Delta$-restricted sensitivity neighbors. We want to show that we can see substantial improvements in the privacy loss, where the loss can instead be written in terms of $\Delta$, rather than $k$ when $\Delta < k$. Our previous algorithm did achieve improvement for the *restricted sensitivity* setting, in that the additive term on the threshold could instead be written as $\ln(\Delta/\delta)/\varepsilon$, but our privacy loss for the $\varepsilon$ term was still in terms of $k$. For this section, we will then assume $\min\{\Delta, \bar{k}\} = \Delta$.

Note that the procedure $\texttt{LimitDom}^{k,\bar{k}}_{\texttt{Lap}}$ in Algorithm 4 is nearly equivalent to our main procedure $\texttt{LimitDom}^{k,\bar{k}}$ in Algorithm 1, with the critical difference that we use Laplace noise, rather than Gumbel noise. Our privacy analysis uses the Laplace mechanism [12] to return the top-$k$ over a limited domain set that is given as input, which we call $\texttt{LapMax}^{k,\bar{k}}$. We cannot achieve this similar privacy guarantee for Gumbel noise because unlike Laplace noise, releasing a count value with added Gumbel noise is not necessarily differentially private. Conversely, we cannot achieve the same privacy guarantees from this procedure as we do with Gumbel noise, particularly with respect to the application of advanced composition.

---

**Algorithm 4** $\texttt{LimitDom}^{k,\bar{k}}_{\texttt{Lap}}$; $\Delta$-Restricted Sensitivity Random Threshold with $k, \bar{k}$

---

**Input:** Histogram $\mathbf{h}$, cut off at $\bar{k} \geq k$, along with parameters $\varepsilon, \delta$.
**Output:** Ordered set of indices $S$.
Set $v_\perp = h_{(\bar{k}+1)} + 1 + \ln(\Delta/\varepsilon) + \texttt{Lap}(1/\varepsilon)$
**for** $i \leq \bar{k}$ **do**
    Set $v_i = h_{(i)} + \texttt{Lap}(1/\varepsilon)$
Sort $\{v_i\} \cup v_\perp$
Let $v_{i_{(1)}}, ...., v_{i_{(j)}}$ be the sorted list until $v_\perp$
Return $\{i_{(1)}, ..., i_{(j)}, \perp\}$ if $j < k$, otherwise return $\{i_{(1)}, ..., i_{(k)}\}$

---

**Lemma 6.1.** *Algorithm 4 is $(\Delta\varepsilon, (e^{\Delta\varepsilon}+1)\bar{\delta})$-DP where $\bar{\delta} = \frac{\delta}{4}(3 + \ln(\Delta/\delta))$*

As in Section 5.2, we will instead write this algorithm with respect to a more generalized version that considers restricting to an arbitrary subset of indices as opposed to just those with value in the true top-$\bar{k}$.

**Definition 6.1.** *[Limited Histogram Report Noisy top k] We define the limited histogram report noisy top k to be $\texttt{LapMax}^{k,\bar{k}} : \mathbb{N}^d \times 2^{[d]} \to [d] \cup \{\perp\}$ such that*

$$\texttt{LapMax}^{k,\bar{k}}(\mathbf{h}, \mathbf{d}) = \begin{cases} (i_{(1)}, ..., i_{(j)}, \perp) & \text{if } j < k \\ (i_{(1)}, ..., i_{(k)}) & \text{otherwise} \end{cases}$$

*where $(v_{(1)}, ..., v_{(j)}, v_\perp)$ is the sorted list until $b_\perp$ of $v_i = h_{(i)} + \texttt{Lap}(1/\varepsilon)$ and $v_\perp = h_\perp + \texttt{Lap}(1/\varepsilon)$, for each $i \in \mathbf{d}$ and*

$$h_\perp := h_{(\bar{k}+1)} + 1 + \ln(\Delta/\varepsilon) \tag{7}$$

We then have the following result that connects $\texttt{LapMax}^{k,\bar{k}}$ with $\texttt{LimitDom}_{\texttt{Lap}}^{k,\bar{k}}$.

**Corollary 6.1.** *For any histogram $\mathbf{h}$, we have that $\texttt{LapMax}^{k,\bar{k}}(\mathbf{h}, \mathbf{d}^{\bar{k}}(\mathbf{h}))$ and $\texttt{LimitDom}_{\texttt{Lap}}^{k,\bar{k}}(\mathbf{h})$ are equal in distribution.*

If we fix a domain $\mathbf{d}$ beforehand, then we have the following privacy statement. Note that we could allow $\texttt{LapMax}^{k,\bar{k}}(\mathbf{h}, \mathbf{d})$ to release the full noisy histogram, with counts, over the limited domain $\mathbf{d} \cup \perp$, since the privacy analysis follows from the Laplace mechanism [12] being $\Delta\varepsilon$-DP. We just need to ensure that $i_{(\bar{k}+1)} \notin \mathbf{d}$ because then if it was, then changing one index would change the count of both $h_{(\bar{k}+1)}$ and $h_\perp$.

**Lemma 6.2.** *For any fixed $\mathbf{d} \subseteq [d]$ and neighboring histograms $\mathbf{h}, \mathbf{h}'$ such that $i_{(\bar{k}+1)}, i'_{(\bar{k}+1)} \notin \mathbf{d}$, then we have that for any set of outcomes $S$*

$$\Pr[\texttt{LapMax}^{k,\bar{k}}(\mathbf{h}, \mathbf{d}) \in S] \le e^{\Delta\epsilon} \Pr[\texttt{LapMax}^{k,\bar{k}}(\mathbf{h}', \mathbf{d}) \in S]$$

As we did in Definition 5.2, we define the *good* and *bad* outcome sets.

**Definition 6.2.** *Given two neighboring histograms $\mathbf{h}, \mathbf{h}'$, we define $\mathcal{S}_{Lap}$ as the outcome set of $\texttt{LimitDom}_{\texttt{Lap}}^{k,\bar{k}}(\mathbf{h}, \mathbf{d}^{\bar{k}}(\mathbf{h}))$ and the outcome set of $\texttt{LimitDom}_{\texttt{Lap}}^{k,\bar{k}}(\mathbf{h}', \mathbf{d}^{\bar{k}}(\mathbf{h}'))$ as $\mathcal{S}'_{Lap}$.*

*We then define the* bad outcomes *as*

$$\mathcal{S}_{Lap}^\delta := \mathcal{S}_{Lap} \setminus \mathcal{S}'_{Lap} \qquad \text{and} \qquad \mathcal{S}_{Lap}'^\delta := \mathcal{S}'_{Lap} \setminus \mathcal{S}_{Lap}$$

As with the analysis in Section 5.2, we will need to bound the probability of outputting something from $\mathcal{S}_{Lap}^\delta$, and also show that we can achieve differential privacy for the remaining outputs that are possible for both $\mathbf{h}$ and $\mathbf{h}'$. For bounding the bad outcomes, it suffices to consider each index in $\mathbf{d}^{\bar{k}}(\mathbf{h}) \setminus \mathbf{d}^{\bar{k}}(\mathbf{h}')$ and bound the probability that its respective noisy value is above the noisy threshold. By construction, both will have Laplace noise added, which will cause the expression for this bound to be slightly messier because we cannot rewrite $\texttt{Lap}(1/\varepsilon) + \texttt{Lap}(1/\varepsilon) \equiv \texttt{Lap}(2/\varepsilon)$, and the proof will require a more technical analysis.

**Lemma 6.3.** *For any $\Delta$-restricted sensitivity neighboring histograms $\mathbf{h}, \mathbf{h}'$, we must have*

$$\Pr[\texttt{LapMax}^{k,\bar{k}}(\mathbf{h}, \mathbf{d}^{\bar{k}}(\mathbf{h})) \in \mathcal{S}_{Lap}^\delta] \le \frac{\delta}{4} \cdot (3 + \ln(\Delta/\delta)) =: \bar{\delta} \tag{8}$$

We defer the proof to Appendix D. We then also need to give differential privacy bounds on the *good* outcomes, but these bounds will be harder to achieve for Laplace noise because working with conditional probabilities in this setting is much more difficult. More specifically, if we again consider the rejection sampling scheme where we throw out any outcomes in $\mathcal{S}_{\mathtt{Lap}}^{\delta}$, when Laplace noise is added we cannot just consider this equivalent to never having considered any of the indices in $\mathbf{d}^{\bar{k}}(\mathbf{h}) \setminus \mathbf{d}^{\bar{k}}(\mathbf{h}')$. As a result, our bounds on the *good* outcomes will instead be approximate differential privacy guarantees, which is the reason for the $(e^{\Delta\varepsilon} + 1)$ factor on the $\delta$ in our final privacy guarantees.

**Lemma 6.4.** *For any neighboring histograms* $\mathbf{h}, \mathbf{h}'$ *and for any* $S \subseteq \mathcal{S}_{\mathtt{Lap}} \cap \mathcal{S}'_{\mathtt{Lap}}$, *we let* $\mathbf{d}^{\varepsilon} = \mathbf{d}^{\bar{k}}(\mathbf{h}) \cap \mathbf{d}^{\bar{k}}(\mathbf{h}')$ *and we must have the following for* $\bar{\delta}$ *given in* (8)

$$\Pr[\mathit{LapMax}^{k,\bar{k}}(\mathbf{h}, \mathbf{d}^{\bar{k}}(\mathbf{h})) \in S] \leq \Pr[\mathit{LapMax}^{k,\bar{k}}(\mathbf{h}, \mathbf{d}^{\varepsilon}) \in S] \leq \Pr[\mathit{LapMax}^{k,\bar{k}}(\mathbf{h}, \mathbf{d}^{\bar{k}}(\mathbf{h})) \in S] + \bar{\delta}$$

We defer the proof to Appendix D. Combining these two lemmas in a similar way to the previous section, will then give our main result of this subsection.

**Lemma 6.5.** *For any neighboring histograms* $\mathbf{h}, \mathbf{h}'$ *and any* $S \subseteq \mathcal{S}_{\mathtt{Lap}}$, *then for* $\bar{\delta}$ *given in* (8),

$$\Pr[\mathit{LapMax}^{k,\bar{k}}(\mathbf{h}, \mathbf{d}^{\bar{k}}(\mathbf{h})) \in S] \leq e^{\Delta\varepsilon} \Pr[\mathit{LapMax}^{k,\bar{k}}(\mathbf{h}', \mathbf{d}^{\bar{k}}(\mathbf{h}')) \in S] + (e^{\Delta\varepsilon} + 1)\bar{\delta}.$$

*Proof.* We will first separate $S$ such that $S^{\delta} = S \cap \mathcal{S}_{\mathtt{Lap}}^{\delta}$ and $S^{\varepsilon} = S \setminus S^{\delta}$. We use Lemma 6.3 to bound

$$\Pr[\mathtt{LapMax}^{k,\bar{k}}(\mathbf{h}, \mathbf{d}^{\bar{k}}(\mathbf{h})) \in S^{\delta}] \leq \bar{\delta}$$

Furthermore, by Lemma 6.4 we have

$$\Pr[\mathtt{LapMax}^{k,\bar{k}}(\mathbf{h}, \mathbf{d}^{\bar{k}}(\mathbf{h})) \in S^{\varepsilon}] \leq \Pr[\mathtt{LapMax}^{k,\bar{k}}(\mathbf{h}, \mathbf{d}^{\varepsilon}) \in S^{\varepsilon}]$$

and also

$$\Pr[\mathtt{LapMax}^{k,\bar{k}}(\mathbf{h}', \mathbf{d}^{\varepsilon}) \in S^{\varepsilon}] \leq \Pr[\mathtt{LapMax}^{k,\bar{k}}(\mathbf{h}', \mathbf{d}^{\bar{k}}(\mathbf{h}')) \in S^{\varepsilon}] + \bar{\delta}$$

Combining these inequalities with Lemma 6.2 we have

$$
\begin{aligned}
\Pr[\mathtt{LapMax}^{k,\bar{k}}(\mathbf{h}, \mathbf{d}^{\bar{k}}(\mathbf{h})) \in S^{\varepsilon}] &\leq \Pr[\mathtt{LapMax}^{k,\bar{k}}(\mathbf{h}, \mathbf{d}^{\varepsilon}) \in S^{\varepsilon}] \\
&\leq e^{\Delta\varepsilon} \Pr[\mathtt{LapMax}^{k,\bar{k}}(\mathbf{h}', \mathbf{d}^{\varepsilon}) \in S^{\varepsilon}] \\
&= e^{\Delta\varepsilon} \left( \Pr[\mathtt{LapMax}^{k,\bar{k}}(\mathbf{h}', \mathbf{d}^{\bar{k}}(\mathbf{h}')) \in S^{\varepsilon}] + \bar{\delta} \right)
\end{aligned}
$$

We use the fact that $\Pr[\mathtt{LapMax}^{k,\bar{k}}(\mathbf{h}', \mathbf{d}^{\bar{k}}(\mathbf{h}')) \in S^{\delta}] = 0$ by definition, so $\Pr[\mathtt{LapMax}^{k,\bar{k}}(\mathbf{h}', \mathbf{d}^{\bar{k}}(\mathbf{h}')) \in S^{\varepsilon}] = \Pr[\mathtt{LapMax}^{k,\bar{k}}(\mathbf{h}', \mathbf{d}^{\bar{k}}(\mathbf{h}')) \in S]$. Finally, this gives

$$
\begin{aligned}
\Pr[\mathtt{LapMax}^{k,\bar{k}}(\mathbf{h}, \mathbf{d}^{\bar{k}}(\mathbf{h})) \in S] &= \Pr[\mathtt{LapMax}^{k,\bar{k}}(\mathbf{h}, \mathbf{d}^{\bar{k}}(\mathbf{h})) \in S^{\delta}] + \Pr[\mathtt{LapMax}^{k,\bar{k}}(\mathbf{h}, \mathbf{d}^{\bar{k}}(\mathbf{h})) \in S^{\varepsilon}] \\
&\leq \bar{\delta} + e^{\Delta\varepsilon} \left( \Pr[\mathtt{LapMax}^{k,\bar{k}}(\mathbf{h}', \mathbf{d}^{\bar{k}}(\mathbf{h}')) \in S^{\varepsilon}] + \bar{\delta} \right) \\
&= e^{\Delta\varepsilon} \Pr[\mathtt{LapMax}^{k,\bar{k}}(\mathbf{h}', \mathbf{d}^{\bar{k}}(\mathbf{h}')) \in S] + (e^{\Delta\varepsilon} + 1)\bar{\delta}
\end{aligned}
$$

□

*Proof of Lemma 6.1.* Follows immediately from Corollary 6.1 and Lemma 6.5.  □

## 6.2  Strictly Limited Domain Algorithm

In this section, we give a slight variant of Algorithm 1 that achieves the same privacy guarantees in the *unrestricted sensitivity* setting, but will allow for an even more efficient implementation. Note that in Algorithm 1 we always assume access to the true top-$\bar{k}$, and this is necessary even if some of those values are zero. However, as you would expect, most online analytical processing algorithms for top-$\bar{k}$ queries only return non-zero values. One way to work around this is to maintain a fixed list of $\bar{k}$ indices, and anytime the true top-$\bar{k}$ has fewer than $\bar{k}$ non-zeros we auto-populate the histogram with indices from our fixed list (note that we allowed for a fixed arbitrary tie-breaking of indices). A more practical approach would be to instead only consider indices whose value is strictly greater than the $(\bar{k}+1)$th index. We will show that this approach still achieves the same privacy guarantees, but does not see the same improvement in the *restricted-sensitivity* setting.

---

**Algorithm 5** Top $k$ from the strictly limited domain of $\bar{k} \in \{k, \cdots, d-1\}$ of the histogram

---

**Input:** Histogram $\mathbf{h}$; cut off at $\bar{k} \geq k$, along with parameters $\varepsilon, \delta$.
**Output:** Ordered set of indices.
Set $h_\perp = h_{(\bar{k}+1)} + 1 + \ln(\bar{k}/\delta)/\varepsilon$
Set $v_\perp = h_\perp + \texttt{Gumbel}(1/\varepsilon)$
**for** $j \leq \bar{k}$ **do**
    **if** $h_{(j)} > h_{(\bar{k}+1)}$ **then**
        Set $v_j = h_{(j)} + \texttt{Gumbel}(1/\varepsilon)$
Sort $\{v_j\} \cup v_\perp$
Let $v_{i_{(1)}}, ...., v_{i_{(j)}}, v_\perp$ be the sorted list up until $v_\perp$
Return $\{i_{(1)}, ..., i_{(j)}, \perp\}$ if $j < k$, otherwise return $\{i_{(1)}, ..., i_{(k)}\}$

---

**Lemma 6.6.** *Algorithm 5 achieves the same privacy guarantees as Algorithm 1.*

Recall how we defined the limited domain $\mathbf{d}^k(\mathbf{h})$ in (5) for a given histogram $\mathbf{h}$. We will further restrict this domain to a smaller domain $\mathbf{d}^k_>(\mathbf{h})$ for each database and a given value $k$, where he only consider counts that are strictly larger than the $(k+1)$-th largest count,

$$\mathbf{d}^k_>(\mathbf{h}) := \{i \in [d] : h_i > h_{(k+1)}\} \subseteq \mathbf{d}^k(\mathbf{h}). \tag{9}$$

We now present a variant of Lemma 5.2 for the limited domain $\mathbf{d}^k_>(\mathbf{h})$

**Lemma 6.7.** *For any neighboring histograms $\mathbf{h}, \mathbf{h}'$, and some fixed $k < d$, then one of the following must hold:*

1. *$\mathbf{d}^k_>(\mathbf{h}') \subseteq \mathbf{d}^k_>(\mathbf{h})$ and for any $i \in \mathbf{d}^k_>(\mathbf{h}) \setminus \mathbf{d}^k_>(\mathbf{h}')$ we must have $h_i = h_{(k+1)} + 1$*

2. *$\mathbf{d}^k_>(\mathbf{h}) \subseteq \mathbf{d}^k_>(\mathbf{h}')$ and for any $i \in \mathbf{d}^k_>(\mathbf{h}') \setminus \mathbf{d}^k_>(\mathbf{h})$ we must have $h'_i = h'_{(k+1)} + 1$*

Note that either of these properties is possible even if $\mathbf{h}'$ is the histogram obtained from removing one person's data from $\mathbf{h}$, or vice versa.

*Proof.* To simplify notation, let $\mathbf{d} = \mathbf{d}^k_>(\mathbf{h})$ and $\mathbf{d}' = \mathbf{d}^k_>(\mathbf{h}')$. Consider the case where $\mathbf{h} \geq \mathbf{h}'$. Assume $\mathbf{d} \not\subseteq \mathbf{d}'$ so there must exist $j \in \mathbf{d} \setminus \mathbf{d}'$. Hence, $h_j > h_{(k+1)}$ but $h'_j \leq h'_{(k+1)}$. Since $h'_j \leq h_j \leq h'_j + 1$, we must have

$$h_{(k+1)} < h_j \leq h'_j + 1 \leq h'_{(k+1)} + 1.$$

By Lemma 5.1 we must have $h_{(k+1)} = h'_{(k+1)}$ and $h_j = h_{(k+1)} + 1$. We now show that it must be the case that $\mathbf{d}' \subseteq \mathbf{d}$, so we assume that $\exists j' \in \mathbf{d}' \setminus \mathbf{d}$. Thus, $h'_{(k+1)} < h'_{j'}$ but $h_{j'} \leq h_{(k+1)}$ and recall that $h_{(k+1)} = h'_{(k+1)}$ in this case, in which case $h_{(k+1)} < h'_{j'} \leq h_{j'} \leq h_{(k+1)}$, and hence a contradiction.

We now assume that $\mathbf{d}' \not\subseteq \mathbf{d}$ so there must exist $j' \in \mathbf{d}' \setminus \mathbf{d}$. We then have $h'_{j'} > h'_{(k+1)}$ but $h_{j'} \leq h_{(k+1)}$, which forms the following sequence of inequalities,

$$h'_{(k+1)} < h'_{j'} \leq h_{j'} \leq h_{(k+1)}.$$

Again, from Lemma 5.1 we must have $h_{(k+1)} = h'_{(k+1)} + 1$ and $h'_{j'} = h'_{(k+1)} + 1$. Now assume that $\exists j \in \mathbf{d} \setminus \mathbf{d}'$ and so $h_{(k+1)} < h_j$ and $h'_j \leq h'_{(k+1)}$. This gives us $h'_{(k+1)} + 1 < h'_j + 1 \leq h'_{(k+1)} + 1$, and hence a contradiction.

The case where $\mathbf{h}' \geq \mathbf{h}$ follows the same argument. □

*Proof of Lemma 6.6.* For all the proofs in Section 5.2 we can just replace all calls to $\mathbf{d}^{\bar{k}}(\mathbf{h})$ with our new definition of $\mathbf{d}^{\bar{k}}_>(\mathbf{h})$, as this variant is the same algorithm but simply considers a smaller set of indices to output. We wrote our definition of the peeling exponential mechanism to allow any subset of indices, so all the proofs in this regard will generalize to this new domain definition.

The only change we then need to consider is how this changes our bound of $\delta$. The two sufficient properties that were used for bounding $\delta$ were that for any $i \in \mathbf{d}^{\bar{k}}_>(\mathbf{h}) \setminus \mathbf{d}^{\bar{k}}_>(\mathbf{h}')$ we had $h_i \leq h_{(\bar{k}+1)} + 1$, i.e. Lemma 5.2, and that $|\mathbf{d}^{\bar{k}}_>(\mathbf{h}) \setminus \mathbf{d}^{\bar{k}}_>(\mathbf{h}')| \leq \min\{\Delta, \bar{k}\}$, i.e. Lemma 5.3. Note that Lemma 6.7 still gives us the first property for this new domain. Further note that the only other change to our algorithm is that we now have the value added for $h_\perp$ is now $\ln(\bar{k}/\delta)/\varepsilon$ as opposed to $\ln(\min\{\Delta, \bar{k}\}/\delta)/\varepsilon$. Accordingly, it then just suffices to bound $|\mathbf{d}^{\bar{k}}_>(\mathbf{h}) \setminus \mathbf{d}^{\bar{k}}_>(\mathbf{h}')| \leq \bar{k}$, which is true by construction.

□

## 6.3 Optimizing for Threshold Index

In this section, we will discuss the choice of $\bar{k}$ for determining our threshold when we assume that our sensitivity is unbounded, i.e. $\Delta = d$. This can viewed as an optimization problem that is dependent on the data, where we would like to maximize the probability of the noisy values being above our noisy threshold $h_\perp = h_{(\bar{k}+1)} + 1 + \ln(\bar{k}/\delta)/\varepsilon + \texttt{Gumbel}(1/\varepsilon)$ in Algorithm 1. Note that our threshold has one term, $h_{(\bar{k}+1)}$, that is decreasing in $\bar{k}$ and another term, $\ln(\bar{k}/\delta)/\varepsilon$, that is increasing in $\bar{k}$. The optimization problem will then be to minimize $h_{(\bar{k}+1)} + 1 + \ln(\bar{k}/\delta)/\varepsilon$. Intuitively, this will be a point in the histogram in which we see a sudden drop. However, making this data-dependent can violate privacy, so we will instead compute this minimal index using the

---
**Algorithm 6** Optimal Threshold
---
**Input:** Histogram $\mathbf{h}$, cut off at $\bar{d}$ with values $h_{(1)} \geq \cdots \geq h_{(\bar{d})} \geq \cdots \geq h_{(d)}$, along with $k, \varepsilon, \delta$.
**Output:** Ordered set of indices $S$.
**for** $i \in [k, \bar{d}]$ **do**
    Set $v_i = h_{(i)} + \ln(i/\delta)/\varepsilon + \texttt{Gumbel}(1/\varepsilon)$
Set $\bar{k} = \text{argmin}\{v_i\}$
Return $\texttt{LimitDom}^{k-1, \bar{k}}(\mathbf{h}; \varepsilon, \delta)$
---

exponential mechanism $\texttt{EM}_q$ with $q(\mathbf{h}, i) = -h_i - \ln(i/\delta)/\varepsilon$ to return an estimate for the minimum count and pay an additional $\varepsilon$ in the privacy to achieve a better threshold.

We can actually return the optimal $\bar{k}$ that was computed with no additional privacy cost.

**Lemma 6.8.** *For any $\delta' \geq 0$, Algorithm 6 is $(\varepsilon', \delta + \delta')$-DP for $\varepsilon'$ given in* (2).

*Proof.* From Lemma 5.1 we know that if we have neighboring histograms $\mathbf{h} \geq \mathbf{h}'$ then for any $i \in [d]$ we must have $h_{(i)} = h'_{(i)}$ or $h_{(i)} = h'_{(i)} + 1$, which is to say that it has sensitivity 1 and is monotonic. Furthermore, we know that $\ln(i/\delta)/\varepsilon$ is fixed and not data dependent, and so because adding Gumbel noise is equivalent to running the exponential mechanism, which is $\varepsilon$-range-bounded, our choice of $\bar{k}$ is $\varepsilon$-range-bounded. We define $\mathcal{M}(\mathbf{h})$ to be the mechanism that returns $\text{argmin}_{i \in [k+1, \bar{d}]}\{h_{(i)} + \ln(i/\delta)/\varepsilon + \texttt{Gumbel}(1/\varepsilon)\}$ and is hence $\varepsilon$-range-bounded.

Note that our analysis in Section 5.2 required introducing an adaptive sequence of *peeling* exponential mechanisms $\texttt{pLEM}^{k, \bar{k}}(\mathbf{h}, \mathbf{d}^{\bar{k}}(\mathbf{h}))$ and then showing that $\texttt{LimitDom}^{k, \bar{k}}(\mathbf{h})$ was equivalent in distribution to it. Here, we are starting with a different exponential mechanism to obtain $\bar{k}$, but once we have that we continue as usual, albeit with $k-1$ rather than $k$. It is then straightforward to replicate the analysis in Section 5.2, except with the alternative peeling exponential mechanism $\texttt{pLEM}^{k-1, \mathcal{M}(\mathbf{h})}\left(\mathbf{h}, \mathbf{d}^{\mathcal{M}(\mathbf{h})}(\mathbf{h})\right)$.

$\square$

# 7 Pay-what-you-get Composition

In this section, we will examine our multiple calls procedure $\texttt{multiLimitDom}^{k^\star, \ell^\star}$ given in Algorithm 2, and present the analysis to prove Theorem 2.[6] We will use the fact that the closeness of the probability distribution for our limited domain algorithm depends on the size of the output in order to reduce the privacy loss over multiple calls to the limited domain algorithm. We can view this as a combination of the peeling exponential mechanism and the sparse vector technique [10]. More specifically, we know that Algorithm 1 is equivalent to a variant of the peeling exponential mechanism, and the size of the output tells us the number of adaptive calls made to the exponential mechanism. Accordingly, we set a threshold that bounds the total number of adaptively selected limited domain exponential mechanism calls made over multiple rounds of Algorithm 1. This parallels the sparse-vector technique that bounds the number of calls to above-threshold, a mechanism that also has a stopping condition based upon noisy estimates falling above or below

a noisy threshold. However, if we want to show that the privacy degrades with the size of the output, then we cannot just take the privacy guarantees from Theorem 1 (which are in terms of $k$) and apply known adaptive composition theorems. Additionally, we want to apply advanced composition on the total calls to the limited exponential mechanism within calls to Algorithm 1, but we only want to pay for an additional $\delta$ each time we call Algorithm 1.

This will require a more meticulous analysis of our multiple calls procedure. As in our previous analysis, we want to fix neighboring histograms so that we can separate the respective outcome space into *good* and *bad* events. This will become notationally heavy for this procedure because the histograms are an adaptive sequence. It then becomes necessary to fix neighboring adaptive sequences of queries, which we will set up with the following notation.

Let $\mathbf{h}_i(o_{<i})$ be the adaptive histogram that results from seeing the previous outcomes $o_1, ..., o_{i-1}$. We want to fix the randomness in the adaptive sequence of neighboring histograms. We then define $\mathcal{H}^{(0)}$ to be the family of all possible adaptive histogram sequences, $\mathbf{h}_1^{(0)}, \mathbf{h}_2^{(0)}(o_1), \mathbf{h}_3^{(0)}(o_{<3})$, $\cdots, \mathbf{h}_\ell^{(0)}(o_{<\ell})$ where $\ell \leq \ell^\star$ and $o_i$ is a feasible outcome for $\texttt{LimitDom}^{k_i, \bar{k}_i}\left(\mathbf{h}_i^{(0)}(o_{<i})\right)$. We then write $\mathcal{H}^{(1)}$ to be a neighboring family of $\mathcal{H}^{(0)}$, which is to say that it consists of adaptive histogram sequences $\mathbf{h}_1^{(1)}, \mathbf{h}_2^{(1)}(o_1), \cdots$, where each $\mathbf{h}_i^{(1)}(o_{<i})$ is a neighbor of $\mathbf{h}_i^{(0)}(o_{<i})$. Note that there might be outcomes $o_i$ that are feasible in the adaptive sequence of histograms $h_1^{(b)}, \cdots h_i^{(b)}(o_{<i})$ when $b = 0$ but not when $b = 1$. Further, when $b = 1$, we are not even considering some feasible outcomes in $\mathcal{H}^{(0)}$ or $\mathcal{H}^{(1)}$. We want to make sure that we are only looking over feasible outcomes for both $b = 0$ and $b = 1$. Note that $\mathcal{H}^{(b)}$ for $b \in \{0, 1\}$ can be thought of as a tree showing each possible realized histogram in an adaptive sequence based on the previous outcomes, so that a sequence $\vec{\mathbf{h}} \in \mathcal{H}^{(b)}$ gives a possible path in $\mathcal{H}^{(b)}$. Furthermore, for any $\mathbf{h}_i^{(1)}(o_{<i})$ and $\mathbf{h}_i^{(0)}(o_{<i})$, we let $\mathbf{d}_i^{\bar{k}_i}(o_{<i}) = \mathbf{d}^{\bar{k}_i}(\mathbf{h}_i^{(0)}(o_{<i})) \cap \mathbf{d}^{\bar{k}_i}(\mathbf{h}_i^{(1)}(o_{<i}))$

The procedure $\texttt{pMultiLEM}_{\mathcal{H}^{(0)}, \mathcal{H}^{(1)}}^{k^\star, \ell^\star}$ in Algorithm 7 presents a variant of Algorithm 2 that calls the peeling exponential mechanism, but only has adaptive sequences in $\mathcal{H}^{(0)}$ or $\mathcal{H}^{(1)}$. Note that we have limited the outcomes at round $i$ to only be from $\texttt{pLEM}^{k_i}(\mathbf{h}_i^{(b)}(o_{<i}), \mathbf{d}_i^{\bar{k}_i}(o_{<i}))$, which restricts the outcomes to only be sequences in $\mathbf{d}_i^{\bar{k}_i}(o_{<i}) \cup \{\bot\}$, where the indices are common for either $b = 0$ or $b = 1$.

---

**Algorithm 7** $\texttt{pMultiLEM}_{\mathcal{H}^{(0)}, \mathcal{H}^{(1)}}^{k^\star, \ell^\star}$; Multiple queries to random threshold with limited domain

---

**Input:** Bit $b \in \{0, 1\}$ corresponding to $\mathcal{H}^{(b)}$ with privacy parameters $\varepsilon, \delta$.
**Output:** Sequence of outputs $(o_1, ..., o_\ell)$ for $\ell \leq \ell^\star$.
**while** $k^\star > 0$ and $\ell^\star > 0$ **do**
    From previous outcome $o_{<i}$, select $k_i$ and $\bar{k}_i$ and $\mathbf{h}_i^{(b)}(o_{<i})$ in $\mathcal{H}^{(b)}$
    **if** $k_i \leq k^\star$ **then**
        Let $o_i = \texttt{pLEM}^{k_i, \bar{k}_i}(\mathbf{h}_i^{(b)}(o_{<i}), \mathbf{d}_i^{\bar{k}_i}(o_{<i}))$ along with $\varepsilon$ and $\delta$
        $k^\star \leftarrow k^\star - |o_i|$ and $\ell^\star \leftarrow \ell^\star - 1$
  Return $o = (o_1, ..., o_\ell)$

---

**Lemma 7.1.** *For any $\delta' \geq 0$, Algorithm 7 is $(\varepsilon^\star, \delta')$-DP where $\varepsilon^\star$ is given in (3).*

*Proof.* Algorithm 7 is just an adaptive sequence of exponential mechanism calls (and at most $k^\star$),

and they must use the same subset of $[d]$ by construction so we know each is $\varepsilon$-range-bounded by Corollary 5.1. The privacy guarantees then follow from Lemma 4.4. $\qquad\square$

As in previous sections, we consider these fixed neighbors which are adaptive sequences here, and we separate our outcome space into *bad* and *good* outcome sets.

**Definition 7.1.** *Given neighboring families of adaptive histogram sequences $\mathcal{H}^{(0)}$ and $\mathcal{H}^{(1)}$, let $\mathcal{S}^{(b)}$ be the set of possible outputs from Algorithm 2 on $\mathcal{H}^{(b)}$. As before we let*

$$\mathcal{S}^{\delta} := \mathcal{S}^{(0)} \setminus \mathcal{S}^{(1)}$$

Note that $\mathcal{S}^{(0)} \cap \mathcal{S}^{(1)}$ is the common set of possible outputs when Algorithm 7 has input $b = 0$ or $b = 1$. We then bound the probability of an outcome in the set of *bad* outcomes by simply considering each adaptive call to Algorithm 1, and use our known bounds for this algorithm outputting a *bad* outcome.

**Lemma 7.2.** *For adaptive sequence of histograms $\mathcal{H}^{(0)}$, the procedure $\mathtt{multiLimitDom}^{k^{\star},\ell^{\star}}$ satisfies the following*

$$\Pr[\mathtt{multiLimitDom}^{k^{\star},\ell^{\star}}(\mathcal{H}^{(0)}) \in \mathcal{S}^{\delta}] \leq \ell^{\star}\delta$$

*Proof.* By construction, we know that $\mathtt{multiLimitDom}^{k^{\star},\ell^{\star}}(\mathcal{H}^{(0)})$ is just an adaptive sequence of calls to $\mathtt{LimitDom}^{k_i,\bar{k}_i}(\mathbf{h}_i^{(0)}(o_{<i}))$ that depends on the outcomes of the previous rounds. We know from Lemma 5.4 that conditional on $o_{<i}$, this probability distribution is equivalent to $\mathtt{pLEM}^{k_i,\bar{k}_i}(\mathbf{h}_i(o_{<i}), \mathbf{d}^{\bar{k}_i}(\mathbf{h}_i(o_{<i})))$. We will write $\mathcal{S}^{(b)}_{o_{<i}}$ to denote the possible outcomes for

$$\mathtt{pLEM}^{k_i,\bar{k}_i}(\mathbf{h}_i(o_{<i}), \mathbf{d}^{\bar{k}_i}(\mathbf{h}_i(o_{<i}))).$$

We then write $S^{\delta}_{o_{<i}} := S^{(0)}_{o_{<i}} \setminus S^{(1)}_{o_{<i}}$. From Lemma 5.5 we have

$$\Pr[\mathtt{pLEM}^{k_i,\bar{k}_i}(\mathbf{h}_i(o_{<i}), \mathbf{d}^{\bar{k}_i}(\mathbf{h}_i(o_{<i}))) \in \mathcal{S}^{\delta}_{o_{<i}} \mid o_{<i}] \leq \delta$$

We then have a bound for every call $\mathtt{LimitDom}^{k_i,\bar{k}_i}(\mathbf{h}_i(o_{<i}))$ conditioned on the previous outcomes, allowing us to union bound this event for each $i \leq \ell^{\star}$. $\qquad\square$

We further need to give differential privacy guarantees for the *good* outcomes as in Lemma 5.6, but we will not be able to achieve as nice of formulation composing these equalities because we are considering the martingale setting and these probabilities are dependent on previous outcomes. Accordingly, we will lose an additional factor of 2 on the $\delta$ for this multiple call setting because we must instead apply bounds on the probability of not outputting a *bad* outcome that hold regardless of the previous outcome, and we cannot apply the same trick from Claim 5.1.

**Lemma 7.3.** *Considering the neighboring adaptive sequences $\mathcal{H}^{(0)}$ and $\mathcal{H}^{(1)}$, for any $o \in \mathcal{S}^{(0)} \cap \mathcal{S}^{(1)}$ we have the following set of inequalities comparing $\mathtt{multiLimitDom}^{k^{\star},\ell^{\star}}(\mathcal{H}^{(0)})$ and $\mathtt{pMultiLEM}^{k^{\star},\ell^{\star}}_{\mathcal{H}^{(0)},\mathcal{H}^{(1)}}(0)$*

$$(1 - \ell^{\star}\delta)\Pr[\mathtt{pMultiLEM}^{k^{\star},\ell^{\star}}_{\mathcal{H}^{(0)},\mathcal{H}^{(1)}}(b) = o]$$
$$\leq \Pr[\mathtt{multiLimitDom}^{k^{\star},\ell^{\star}}(\mathcal{H}^{(b)}) = o]$$
$$\leq \Pr[\mathtt{pMultiLEM}^{k^{\star},\ell^{\star}}_{\mathcal{H}^{(0)},\mathcal{H}^{(1)}}(b) = o]$$

*Proof.* Let the length of outcome $o \in \mathcal{S}^{(0)} \cap \mathcal{S}^{(1)}$ be $\ell$. By construction, we have

$$\Pr[\texttt{multiLimitDom}^{k^\star, \ell^\star}(\mathcal{H}^{(b)}) = o] = \prod_{i=1}^{\ell} \Pr[\texttt{LimitDom}^{k_i, \bar{k}_i}(\mathbf{h}_i^{(b)}(o_{<i})) = o_i \mid o_{<i}]$$

We then apply Lemma 5.4 and Lemma 5.6 where we use $\mathcal{S}_{o_{<i}}^{\delta}$ as in the proof of Lemma 7.2

$$\Pr[\texttt{LimitDom}^{k_i, \bar{k}_i}(\mathbf{h}_i^{(b)}(o_{<i})) = o_i \mid o_{<i}] = \Pr[\texttt{pLEM}^{k_i, \bar{k}_i}(\mathbf{h}_i^{(b)}(o_{<i}), \mathbf{d}_i^{\bar{k}_i}(o_{<i})) = o_i \mid o_{<i}]$$
$$\cdot \Pr[\texttt{pLEM}^{k_i, \bar{k}_i}\left(\mathbf{h}_i^{(b)}(o_{<i}), \mathbf{d}^{\bar{k}_i}(\mathbf{h}_i^{(b)}(o_{<i}))\right) \notin \mathcal{S}_{o_{<i}}^{\delta} \mid o_{<i}].$$

We then apply Lemma 5.5 to obtain

$$(1 - \delta) \Pr[\texttt{pLEM}^{k_i, \bar{k}_i}(\mathbf{h}_i^{(b)}(o_{<i}), \mathbf{d}_i^{\bar{k}_i}(o_{<i})) = o_i \mid o_{<i}]$$
$$\leq \Pr[\texttt{LimitDom}^{k_i, \bar{k}_i}(\mathbf{h}_i^{(b)}(o_{<i})) = o_i \mid o_{<i}]$$
$$\leq \Pr[\texttt{pLEM}^{k_i, \bar{k}_i}(\mathbf{h}_i^{(b)}(o_{<i}), \mathbf{d}_i^{\bar{k}_i}(o_{<i})) = o_i \mid o_{<i}]$$

By construction, we also have

$$\Pr[\texttt{pMultiLEM}_{\mathcal{H}^{(0)}, \mathcal{H}^{(1)}}^{k^\star, \ell^\star}(b) = o] = \prod_{i=1}^{\ell} \Pr[\texttt{pLEM}^{k_i, \bar{k}_i}(\mathbf{h}_i^{(b)}(o_{<i}), \mathbf{d}_i^{\bar{k}_i}(o_{<i})) = o_i \mid o_{<i}].$$

We then use the fact that $\ell \leq \ell^\star$ and $(1 - \ell^\star \delta) \leq (1 - \delta)^{\ell^\star}$ to achieve our desired inequality. $\square$

As with our other privacy proofs, these bounds on the *bad* and *good.* outcomes are the main technical details for proving our main lemma.

**Lemma 7.4.** *For any $S \subseteq \mathcal{S}^{(0)}$, we have the following for $\varepsilon^\star$ given in (3).*

$$\Pr[\textit{multiLimitDom}^{k^\star, \ell^\star}(\mathcal{H}^{(0)}) \in S] \leq e^{\varepsilon^\star} \Pr[\textit{multiLimitDom}^{k^\star, \ell^\star}(\mathcal{H}^{(1)}) \in S] + 2\ell^\star \delta + \delta'$$

*where $\varepsilon^\star$ is given in (3)*

*Proof.* As in our previous analysis of Lemma 5.7, we will separate $S^\delta = S \cap \mathcal{S}^\delta$ and $S^\varepsilon = S \setminus S^\delta$. From Lemma 7.2 we can bound

$$\Pr[\texttt{multiLimitDom}^{k^\star, \ell^\star}(\mathcal{H}^{(0)}) \in S^\delta] \leq \ell^\star \delta$$

We then apply Lemma 7.3 and Lemma 7.1 to obtain

$$(1 - \ell^\star \delta) \Pr[\texttt{multiLimitDom}^{k^\star, \ell^\star}(\mathcal{H}^{(0)}) \in S^\varepsilon]$$
$$\leq (1 - \ell^\star \delta) \Pr[\texttt{pMultiLEM}_{\mathcal{H}^{(0)}, \mathcal{H}^{(1)}}^{k^\star, \ell^\star}(0) \in S^\varepsilon]$$
$$\leq (1 - \ell^\star \delta) \left(e^{\varepsilon^\star} \Pr[\texttt{pMultiLEM}_{\mathcal{H}^{(0)}, \mathcal{H}^{(1)}}^{k^\star, \ell^\star}(1) \in S^\varepsilon] + \delta'\right)$$
$$\leq e^{\varepsilon^\star} \Pr[\texttt{multiLimitDom}^{k^\star, \ell^\star}(\mathcal{H}^{(1)}) \in S^\varepsilon] + (1 - \ell^\star \delta) \delta'$$

Combining these properties with the fact that $\Pr[\texttt{multiLimitDom}^{k^\star,\ell^\star}(\mathcal{H}^{(1)}) \in S^\delta] = 0$ by construction, we achieve

$$
\begin{aligned}
\Pr[&\texttt{multiLimitDom}^{k^\star,\ell^\star}(\mathcal{H}^{(0)}) \in S] \\
&= \Pr[\texttt{multiLimitDom}^{k^\star,\ell^\star}(\mathcal{H}^{(0)}) \in S^\delta] + \Pr[\texttt{multiLimitDom}^{k^\star,\ell^\star}(\mathcal{H}^{(0)}) \in S^\varepsilon] \\
&\leq \ell^\star\delta + e^{\varepsilon^\star}\Pr[\texttt{multiLimitDom}^{k^\star,\ell^\star}(\mathcal{H}^{(1)}) \in S^\varepsilon] + \ell^\star\delta + \delta' \\
&\leq e^{\varepsilon^\star}\Pr[\texttt{multiLimitDom}^{k^\star,\ell^\star}(\mathcal{H}^{(1)}) \in S] + 2\ell^\star\delta + \delta'
\end{aligned}
$$

$\square$

*Proof of Theorem 2.* Follows immediately from Lemma 7.4

$\square$

# 8   Accuracy Analysis

Accuracy comparisons with the standard exponential mechanism approach need to be qualified by the fact that we allow for approximate differential privacy and do not require our algorithm to always output $k$ indices. We made these relaxations in order to achieve differential privacy guarantees while not having our DP algorithms to iterate over the entire dataset, but this also means that we do not face the same lower bounds [2].

For example, if we set $k = \bar{k} = 1$ in our Algorithm 1, then it would only either return the true top index or $\perp$. In general, by restricting ourselves to only looking at the true top-$\bar{k}$ values, the accuracy of our output indices will only improve, and in fact by setting $\bar{k} = k$ we guarantee that output indices must be in the top-$k$. However we can never guarantee that $k$ indices will be output, and the probability of outputting $k$ indices will only decrease the smaller we make $\bar{k}$. Furthermore, quantifying the probability that our algorithm will return $k$ indices is highly data-dependent. Consider the histogram in which all values are equal, then the true top-$\bar{k}$ could become a completely different set of indices in a neighboring database. Given that we only have access to this true top-$\bar{k}$ index set, our algorithm needs to ensure that for this histogram we will return $\perp$ with probability at least $1 - \delta$. However, we would not expect data distributions to be flat, but perhaps closer to a power law distribution, where there will be significant differences between the counts.

In general, our accuracy will be very similar to the standard exponential mechanism when there are reasonably large differences between the values in the histogram, but when values become much closer, our algorithm will return $\perp$ as opposed to a set of indices that is chosen close to uniformly at random. We see this as the primary advantage of our pay-what-you-see composition. If a histogram is queried with values that are very close, instead of providing a list of indices that are drawn close to uniformly at random, our algorithm will only output $\perp$, and the only privacy cost will be for that one output. The $\perp$ output is then also informative in itself.

For a more formal analysis, we consider a comparatively standard metric of accuracy for top-$k$ queries [2].

**Definition 8.1.** *Given histogram* $\mathbf{h}$ *along with non-negative integers* $k$ *and* $\alpha$, *we say that a subset of indices* $\mathbf{d} \subseteq [d] \cup \{\perp\}$ *is an* $(\alpha, k)$-*accurate if for any* $i \in \mathbf{d}$ *such that* $i \neq \perp$, *we have*

$$h_i \geq h_{(k)} - \alpha$$

For this definition, we can give asymptotically better accuracy guarantees than what the standard exponential mechanism achieves, which are known to be tight [2], but it is critically important to mention that our definition does not require $k$ indices to be output. Accordingly, we add a sufficient condition under which our algorithm will return $k$ indices with a given probability.

**Lemma 8.1.** *For any histogram* **h**, *with probability at least* $1 - \beta$ *the output from Algorithm 1 with parameters* $k, \bar{k}, \varepsilon, \delta$ *is* $(\alpha, k)$-*accurate where*

$$\alpha = \frac{\ln(k\bar{k}/\beta)}{\varepsilon}$$

*Additionally, we have that Algorithm 1 will return $k$ indices with probability at least* $1 - \beta$ *if*

$$h_{(k)} \geq h_{(\bar{k}+1)} + 1 + \frac{\ln(\min\{\Delta, \bar{k}\}/\delta)}{\varepsilon} + \frac{\ln(k/\beta)}{\varepsilon}$$

The first statement is essentially equivalent to Theorem 6 in [2] which would have $\alpha = \frac{\ln(kd/\beta)}{\varepsilon}$ in this setting because we incorporate advanced composition at the end of the analysis and we consider the absolute counts (not normalized by the total number of users). Accordingly, our $\alpha$ parameter swaps $d$ with $\bar{k}$ as expected, and will improve the accuracy statement for the output indices.

The utility statement in Bhaskar et al. [5] says that for some $\gamma \geq 0$, with probability at least $1 - \beta$ all the returned indices should have true count at least $h_{(k)} + \gamma$, i.e. *completeness*, and no returned indices should have true count less than $h_{(k)} - \gamma$, i.e. *soundness*.[7] The $\gamma$ in [5] gives

$$\gamma = O\left(\frac{\ln\binom{m}{\ell}}{\varepsilon} + \frac{\ln(k/\beta)}{\varepsilon}\right)$$

such that $m$ is the number of possible items and $\ell$ is the length of the itemset, and once again we remove the factor of $\frac{k}{n}$ for comparing to our setting because we apply composition at the end of our analysis and consider absolute counts instead of normalized counts. The second statement in Lemma 8.1 is similar to the soundness condition, where our algorithm ensures with probability 1 that no index with value below $h_{(\bar{k})}$ is output, and the probability statement is instead over whether we output $k$ indices (which occurs with probability 1 in [5]). The difference in our terms then becomes $\ln(\min\{\Delta, \bar{k}\}/\delta)$ as opposed to $\ln\binom{m}{\ell}$. For satisfying completeness, it is actually straightforward to show using the standard exponential mechanism analysis that we can achieve this for $\gamma = \ln(\bar{k}k/\beta)/\varepsilon$, which technically improves upon $\gamma = \ln(\binom{m}{\ell}k/\beta)/\varepsilon$ in [5] where we can consider $\binom{m}{\ell} = d$. However, this is only because their choice of $\bar{k}$ is the index that satisfies $h_{(k)} \geq h_{(\bar{k}+1)} + \gamma$ so it could be as large as the $d$th index, whereas we consider $\bar{k}$ fixed and satisfies this assumption, so these bounds are equivalent when we have to find $\bar{k}$ to satisfy $h_{(k)} \geq h_{(\bar{k}+1)} + \gamma$.

We now prove the lemma.

*Proof of Lemma 8.1.* We first set up some notation. Let $\mathbf{d}_\alpha := \{i \in \mathbf{d}^{\bar{k}}(\mathbf{h}) : h_i < h_{(k)} - \alpha\}$ be the set of indices in the top-$\bar{k}$ with true value below $h_{(k)} - \alpha$. Furthermore, let $\mathcal{S}_\alpha := \{o : o \cap \mathbf{d}_\alpha \neq \emptyset\}$ be the set of outcomes that includes some index in $\mathbf{d}_\alpha$. Formally, the first statement is equivalent to showing for any histogram $\mathbf{h}$ that

$$\Pr[\texttt{LimitDom}^{k,\bar{k}}(\mathbf{h}) \in \mathcal{S}_\alpha] \leq \beta.$$

From Lemma 5.4 this is equivalent to showing

$$\Pr[\texttt{pLEM}^{k,\bar{k}}(\mathbf{h}, \mathbf{d}^{\bar{k}}(\mathbf{h})) \in \mathcal{S}_\alpha] \leq \beta.$$

By construction, the peeling exponential mechanism makes at most $k$ calls to the limited exponential mechanism $\texttt{LEM}^{\bar{k}}(\mathbf{h}, \mathbf{d})$, and each of these calls must be using an input set $\mathbf{d}$ that contains some index in $\{i_{(1)}, ..., i_{(k)}\}$, which are all the indices in the top-$k$. It then suffices to show that

$$\Pr[\texttt{LEM}^{\bar{k}}(\mathbf{h}, \mathbf{d}^{\bar{k}}(\mathbf{h}) \setminus \{i_{(1)}, ..., i_{(k-1)}\}) \in \mathbf{d}_\alpha] \leq \frac{\beta}{k}$$

Applying our definition of the limited exponential mechanism, we can then obtain the bound

$$\Pr[\texttt{LEM}^{\bar{k}}(\mathbf{h}, \mathbf{d}^{\bar{k}}(\mathbf{h}) \setminus \{i_{(1)}, ..., i_{(k-1)}\}) \in \mathbf{d}_\alpha] \leq \frac{\sum_{i \in \mathbf{d}_\alpha} \exp(\varepsilon h_i)}{\exp(\varepsilon h_{(k)})} \leq \frac{\bar{k} \exp(\varepsilon(h_{(k)} - \alpha))}{\exp(\varepsilon h_{(k)})}$$

where the last step follows from the fact that $|\mathbf{d}_\alpha| \leq \bar{k}$ by construction and for each $j \in \mathbf{d}_\alpha$ we have $h_j < h_{(k)} - \alpha$ by assumption. Cancelling like terms and plugging in for $\alpha = \ln(k\bar{k}/\beta)/\varepsilon$ gives

$$\Pr[\texttt{LEM}^{\bar{k}}(\mathbf{h}, \mathbf{d}^{\bar{k}}(\mathbf{h}) \setminus \{i_{(1)}, ..., i_{(k-1)}\}) \in \mathbf{d}_\alpha] \leq \frac{\bar{k}}{\exp(\varepsilon \alpha)} = \frac{\beta}{k}$$

which proves our first claim.

For the second claim, we want to show that with probability at least $1 - \beta$ there are $k$ indices whose noisy estimate is above the noisy threshold. It then suffices to show that for any $i \leq k$ we have $\Pr[h_\perp + \texttt{Gumbel}(1/\varepsilon) > h_{(i)} + \texttt{Gumbel}(1/\varepsilon)] \leq \frac{\beta}{k}$ where $h_\perp = h_{(\bar{k}+1)} + 1 + \frac{\ln(\min\{\Delta, \bar{k}\}/\delta)}{\varepsilon}$. Setting $k = 1$ in Lemma 4.2, we have that

$$\Pr[h_\perp + \texttt{Gumbel}(1/\varepsilon) > h_{(i)} + \texttt{Gumbel}(1/\varepsilon)] = \frac{\exp(\varepsilon h_\perp)}{\exp(\varepsilon h_{(i)}) + \exp(\varepsilon h_\perp)}$$

Due to the fact that $h_{(i)} \geq h_{(k)}$, we then apply our assumption that $h_{(i)} \geq h_\perp + \ln(k/\beta)/\varepsilon$, and this reduces to

$$\Pr[h_\perp + \texttt{Gumbel}(1/\varepsilon) > h_{(i)} + \texttt{Gumbel}(1/\varepsilon)] \leq \frac{\exp(\varepsilon h_\perp)}{\frac{k}{\beta}\exp(\varepsilon h_\perp) + \exp(\varepsilon h_\perp)} = \frac{\frac{\beta}{k}}{1 + \frac{\beta}{k}} \leq \frac{\beta}{k}$$

$\square$

# 9    Conclusions and Future Directions

We have presented a way to efficiently report the top-$k$ elements in a dataset subject to differential privacy. Our approach does not require adjusting the input data to an existing system, nor does it require altering the non-private data analytics. Our algorithms can be seen as being an additional layer on top of existing systems so that we can leverage highly efficient, scalable data analytics platforms in our private systems. Our algorithms can balance utility in terms of both privacy with $\varepsilon$ as well as efficiency with $\bar{k}$. Further, we have improved on the general composition bounds in differential privacy that can be applied in our setting to extract more utility under the same privacy budget.

We believe that other mechanisms, such as report noisy max [10], could benefit from the tighter characterization of *range-bounded* in advanced composition. An interesting line of future work is developing an optimal composition theorem for further savings in this setting similar to [19, 26]. It would also be useful to show that we could replace Gumbel noise with another distribution and achieve similar or better guarantees, e.g. Laplace or Gaussian noise. In fact for Gaussian noise, one would hope to improve the privacy parameters in Lemma 6.1 from $\Delta\varepsilon$ to $\sqrt{\Delta}\varepsilon$ for the $\Delta$-restricted sensitivity setting.

It would also be interesting to explore other ways to choose $\bar{k}$ in a private, yet also in a data-dependent manner, other than what we presented in Algorithm 6. These directions will be more application dependent that are conditional on the desired tradeoffs between computational restrictions, accuracy, and maximizing the number of outputs. For instance, if we relax the computational restrictions, we could privately choose $\bar{k}$ that achieves a certain separation between $h_{(k)}$ and $h_{(\bar{k})}$ to maximize the probability of outputting $k$ indices. We also leave it as an open problem to construct instance specific lower bounds when the algorithms can return fewer than $k$-indices.

## Footnotes

[1] Note that if $\bar{k}$ becomes comparable to $d$, then we can also have $d - \bar{k}$ in the minimum statement, but we omit for simplicity. If $\bar{k} = d$, then we use write $h_{(d+1)} = 0$, in which case the algorithm becomes equivalent to the exponential mechanism with peeling. This emphasizes that $\bar{k}$ provides a tuning knob between efficiency and utility.

[2]Note that we could have alternatively written our algorithm in terms of iteratively applying exponential mechanism (and all of our analysis will be in this context), but instead adding `Gumbel` noise once is computationally more efficient.

[3]We could also equivalently define this in terms of output sets $S, S' \subseteq \mathcal{Y}$ because we are only considering pure ($\delta = 0$) differential privacy.

[4] http://www.math.wisc.edu/~roch/grad-prob/gradprob-notes20.pdf

[5] Note that Claim 5.1 doesn't apply if $\delta_{\mathbf{h}'} = 1$, but Lemma 5.5 then implies that $\delta \geq 1$ and our desired statement is trivially true.

[6]Note that if we use Algorithm 6 from Section 6.3 to optimize the threshold at each round, then we just need to instead update with $k^\star \leftarrow k^\star - (|o_i| + 1)$ in $\texttt{multiLimitDom}^{k^\star, \ell^\star}$ because we need to additionally pay for the optimization in each call.

[7]It also considers an accuracy statement on the values output after adding fresh Laplace noise to each index that was privately output as part of the top-$k$, which could easily extend to our setting if we wanted to give noisy estimates of the values for our output indices.

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

# A    Comparison between Bounded Range DP Composition and Optimal DP Composition

**Figure 1.** Comparison of bounded range DP composition from Lemma 4.4 and the optimal DP composition from [19]. A ratio larger than 1 means that the optimal composition bound in Lemma A.1 is larger.

Here we compare the composition bound given in Lemma 4.4 and show that it can actually improve on the optimal bound for generally DP, which we state here for the homogeneous (all privacy parameters are the same at each round) case.

**Lemma A.1** (Optimal DP Composition [19]). *For any $\varepsilon \geq 0$ and $\delta \in [0, 1]$, the composed mechanism of $k$ adaptively chosen $\varepsilon$-DP is $((k - 2i)\varepsilon, \delta_i)$-DP for all $i \in \{0, 1, \cdots, \lfloor k/2 \rfloor\}$ where*

$$\delta_i = \frac{\sum_{\ell=0}^{i-1} \binom{k}{\ell} \left(e^{(k-\ell)\varepsilon} - e^{(k-2i+\ell)\varepsilon}\right)}{(1 + e^\varepsilon)^k}$$

In Figure 1, we plot, for various $k$ and $\varepsilon$, the ratio between the composition bound for range bounded DP algorithms and the general DP optimal composition bound, where a ratio larger than 1 means that our bound is smaller. Due to the discrete formula for $\delta_i$ in Lemma A.1, we select the index $i$ that produces the smallest $(k - 2i)\varepsilon$ while $\delta_i \leq 10^{-6}$. Frequently, this $\delta_i$ that is selected is much smaller than the threshold $10^{-6}$, so we use this same $\delta_i$ when we compare our bounds to the optimal composition bound. Note that the jaggedness in the plot is because the optimal composition bound might be $((k - 2i)\varepsilon, \delta \ll 10^{-6})$-DP at round $k$ but $((k + 1 - 2(i + 1))\varepsilon, \delta \approx 10^{-6})$-DP at round $k + 1$. Hence, plotting only the first privacy parameter might be non-monotonic.

# B    Omitted Proofs from Section 4

## B.1    Proof of Lemma 4.2

*Proof.* We start with the peeling exponential mechanism.

$$\Pr[\mathtt{pEM}_q^k(\mathbf{h}) = (o_1, \cdots, o_k)]$$

$$= \frac{\exp(\frac{\varepsilon}{\Delta(q)} q(\mathbf{x}, o_{i_1}))}{\sum_{y \in \mathcal{Y}} \exp(\frac{\varepsilon}{\Delta(q)} q(\mathbf{x}, y))} \cdot \frac{\exp(\frac{\varepsilon}{\Delta(q)} q(\mathbf{x}, o_{i_2}))}{\sum_{y \neq o_1} \exp(\frac{\varepsilon}{\Delta(q)} q(\mathbf{x}, y))} \cdot \cdots \cdot \frac{\exp(\frac{\varepsilon}{\Delta(q)} q(\mathbf{x}, o_{i_k}))}{\sum_{y \notin \{o_1, \cdots o_{k-1}\}} \exp(\frac{\varepsilon}{\Delta(q)} q(\mathbf{x}, y))}$$

Now we consider the one-shot Gumbel noise mechanism. We will write $p_{\texttt{Gumbel}}$ as the density of a $\texttt{Gumbel}(\Delta(q)/\varepsilon)$ random variable, which is given in (1).

$$\Pr[\mathcal{M}^k_{\texttt{Gumbel}}(\mathbf{q}(\mathbf{x})) = (o_1, \cdots, o_k)]$$
$$= \int_{-\infty}^{\infty} p_{\texttt{Gumbel}}(u_1 - q(\mathbf{x}, o_1)) \int_{-\infty}^{u_1} p_{\texttt{Gumbel}}(u_2 - q(\mathbf{x}, o_2)) \cdots \int_{-\infty}^{u_{k-1}} p_{\texttt{Gumbel}}(u_k - q(\mathbf{x}, o_k))$$
$$\prod_{y \neq \{o_1, \cdots, o_k\}} \Pr[\texttt{Gumbel}(\Delta(q)/\varepsilon) < u_k - q(\mathbf{x}, y)] du_k \cdots du_1.$$

Note that we have

$$\Pr[\texttt{Gumbel}(1/\varepsilon) < y] = \exp\left(-\exp\left(-\varepsilon y\right)\right)$$

and

$$p_{\texttt{Gumbel}}(y) = \varepsilon \exp\left(-\varepsilon y - \exp(-\varepsilon y)\right).$$

We then integrate to get the following with the substitution $\varepsilon_q = \frac{\varepsilon}{\Delta(q)}$ and $q(\mathbf{x}, \cdot) = q(\cdot)$,

$$\int_{-\infty}^{u_{k-1}} p_{\texttt{Gumbel}}(u_k - q(o_k)) \prod_{y \notin \{o_1, \cdots, o_k\}} \Pr[\texttt{Gumbel}(1/\varepsilon_q) < u_k - q(y)] du_k$$

$$= \int_{-\infty}^{u_{k-1}} \varepsilon_q \cdot \exp\left(-\varepsilon_q(u_k - q(o_k)) - e^{-\varepsilon_q(u_k - q(o_k))}\right) \cdot \exp\left(-e^{\varepsilon_q u_k} \sum_{y \notin \{o_1, \cdots, o_k\}} e^{\varepsilon_q q(y)}\right) du_k$$

$$= \varepsilon_q e^{\varepsilon_q q(o_k)} \int_{-\infty}^{u_{i_{k-1}}} \exp\left(-\varepsilon_q u_k - e^{-\varepsilon_q u_k}\left(e^{\varepsilon_q q(o_k)} + \sum_{y \notin \{i_1, \cdots i_k\}} e^{\varepsilon_q q(y)}\right)\right) du_k$$

$$= \frac{e^{\varepsilon_q q(o_k)}}{\sum_{y \notin \{o_1, \cdots, o_{k-1}\}} e^{\varepsilon_q q(y)}} \cdot \exp\left(-e^{-\varepsilon_q u_{k-1}} \cdot \sum_{y \notin \{o_1, \cdots, o_{k-1}\}} e^{\varepsilon_q q(y)}\right).$$

By induction, we have

$$\Pr[M^k_{\texttt{Gumbel}}(\mathbf{q}(\mathbf{x})) = (o_1, \cdots, o_k)] = \frac{e^{\varepsilon_q q(o_1)}}{\sum_{y \in \mathcal{Y}}^d e^{\varepsilon_q q(y)}} \frac{e^{\varepsilon_q q(o_2)}}{\sum_{y \neq o_1} e^{\varepsilon_q q(y)}} \cdots \frac{e^{\varepsilon_q q(o_k)}}{\sum_{y \notin \{o_1, \cdots, o_{k-1}\}} e^{\varepsilon_q q(y)}}.$$

$\square$

## B.2   Proof of Lemma 4.4

*Proof.* We use the same argument as in [13]. Thus, we form the privacy loss random variable at round $i$ as $Z_i = Z_i(v_{\leq i})$ where $v_{\leq i} \sim \mathcal{M}_{\leq i}(\mathbf{x})$ and

$$Z_i(v_{\leq i}) = \ln\left(\frac{\Pr[\mathcal{M}_i(\mathbf{x}) = v_i \mid \mathcal{M}_{<i}(\mathbf{x}) = v_{<i}]}{\Pr[\mathcal{M}_i(\mathbf{x}') = v_i \mid \mathcal{M}_{<i}(\mathbf{x}') = v_{<i}]}\right)$$

We then define our martingale $X_t = \sum_{i=1}^t (Z_i - \mu_i)$ where $\mu_i(v_{<i}) = \mathbb{E}[Z_i | v_{<i}]$. To bound $\mu_i(v_{<i})$, we use results from [6], since each $\mathcal{M}_i$ is also $\varepsilon_i$-DP, which states $\mu_i(v_{<i}) \leq \frac{1}{2}\varepsilon_i^2$. Note that because each algorithm $\mathcal{M}_i$ is $\varepsilon_i$-bounded range DP, there is for some $\alpha_t \in [0, \varepsilon_t]$ such that

$$X_t - X_{t-1} = Z_t - \mu_t(v_{\leq t}) \in [-\alpha_t - \mu_t(v_{\leq t}), \varepsilon_t - \alpha_t - \mu_t(v_{\leq t})]$$

Using Theorem 4, we get the following result

$$\Pr\left[\sum_{i=1}^{t} Z_i \geq \sum_{i=1}^{k} \frac{1}{2}\varepsilon_i^2 + \beta\right] \leq \exp\left(\frac{-2\beta^2}{\sum_{i=1}^{k} \varepsilon_i^2}\right)$$

Hence setting $\beta = \sqrt{\frac{1}{2}\sum_{i=1}^{k} \varepsilon_i^2 \ln(1/\delta)}$ ensures that the total privacy loss is bounded with probability at least $\delta$. The function for $\varepsilon''(\cdot)$ is the minimum over three terms, the first and second terms being from Theorem 3 and the last term being what we just computed. This completes the proof. $\qquad\square$

## C   Omitted Proofs from Section 5.2

### C.1   Proof of Lemma 5.5

**Lemma C.1.** *Given neighboring histograms* $\mathbf{h}, \mathbf{h}'$. *Let* $\mathbf{d}_\delta = \mathbf{d}^{\bar{k}}(\mathbf{h}) \setminus \mathbf{d}^{\bar{k}}(\mathbf{h}')$, *then we have*

$$\Pr[\textit{LEM}^{\bar{k}}(\mathbf{h}, \mathbf{d}_\delta) \neq \perp] \leq \delta$$

*Proof.* For simplicity, we will write $m = \min\{\Delta, \bar{k}\}$. By definition, we can write

$$\Pr[\text{LEM}^{\bar{k}}(\mathbf{h}, \mathbf{d}_\delta) \neq \perp] = \frac{\sum_{i \in \mathbf{d}_\delta} \exp(\varepsilon h_i)}{\exp(\varepsilon h_\perp) + \sum_{i \in \mathbf{d}_\delta} \exp(\varepsilon h_i)}$$

where we know that $h_\perp = h_{(\bar{k}+1)} + 1 + \ln(m/\delta)/\varepsilon$. Furthermore, from Lemma 5.2 we know that $h_i \leq h_{(\bar{k}+1)} + 1$ for each $i \in \mathbf{d}_\delta$. Therefore if we let $x = h_{(\bar{k}+1)} + 1$, we can reduce this to

$$\Pr[\text{LEM}^{\bar{k}}(\mathbf{h}, \mathbf{d}_\delta) \neq \perp] \leq \frac{|\mathbf{d}_\delta| \exp(\varepsilon x)}{\exp(\varepsilon(x + \ln(m/\delta)/\varepsilon)) + |\mathbf{d}_\delta| \exp(\varepsilon x)}$$

Further factoring out all the $\exp(\varepsilon x)$ terms gives

$$\Pr[\text{LEM}^{\bar{k}}(\mathbf{h}, \mathbf{d}_\delta) \neq \perp] \leq \frac{|\mathbf{d}_\delta|}{(m/\delta) + |\mathbf{d}_\delta|} = \frac{\delta(|\mathbf{d}_\delta|/m)}{1 + (\delta/m)|\mathbf{d}_\delta|} \leq \delta(|\mathbf{d}_\delta|/m) \leq \delta$$

where the last inequality follows from the fact that $|\mathbf{d}_\delta| \leq m$ from Lemma 5.3.
$\qquad\square$

**Lemma C.2.** *Consider a subset* $T \subseteq [d]$, *and domain* $\mathbf{d}$ *such that* $T \subseteq \mathbf{d} \subseteq [d]$. *For histogram* $\mathbf{h}$, *we will write the outcome set of* $\textit{pLEM}^k(\mathbf{h}, \mathbf{d})$ *as* $\mathcal{O}$ *and define the set* $\mathcal{T} = \{o \in \mathcal{O} : o \cap T \neq \emptyset\}$. *We then have,*

$$\left(\Pr[\textit{LEM}^{\bar{k}}(\mathbf{h}, T) \neq \perp]\right)^{-1} \Pr[\textit{pLEM}^{k,\bar{k}}(\mathbf{h}, \mathbf{d}) \in \mathcal{T}] \leq 1$$

*Proof.* We will prove this inductively on the size of $k$ where our base case is $k = 1$. By definition

$$\Pr[\text{pLEM}^{k,\bar{k}}(\mathbf{h}, \mathbf{d}) \in \mathcal{T}] = \Pr[\text{LEM}^{\bar{k}}(\mathbf{h}, \mathbf{d}) \in T] = \frac{\sum_{i \in T} \exp(\varepsilon h_i)}{\exp(\varepsilon h_\perp) + \sum_{i \in \mathbf{d}} \exp(\varepsilon h_i)}$$

Therefore,

$$\left(\Pr[\mathtt{LEM}^{\bar{k}}(\mathbf{h}, T) \neq \bot]\right)^{-1} \Pr[\mathtt{pLEM}^{k,\bar{k}}(\mathbf{h}, \mathbf{d}) \in \mathcal{T}] = \frac{\exp(\varepsilon h_\bot) + \sum_{i \in T} \exp(\varepsilon h_i)}{\exp(\varepsilon h_\bot) + \sum_{i \in \mathbf{d}} \exp(\varepsilon h_i)} \leq 1$$

We now assume for $k-1$, and we use the fact that our peeling exponential mechanism iteratively applies the limited domain exponential mechanism, which allows us to rewrite our probability as

$$\Pr[\mathtt{pLEM}^{k,\bar{k}}(\mathbf{h}, \mathbf{d}) \in \mathcal{T}]$$
$$= \Pr[\mathtt{LEM}^{\bar{k}}(\mathbf{h}, \mathbf{d}) \in T] + \sum_{i \in \mathbf{d} \backslash T} \Pr[\mathtt{LEM}^{\bar{k}}(\mathbf{h}, \mathbf{d}) = i] \Pr[\mathtt{pLEM}^{k-1,\bar{k}}(\mathbf{h}, \mathbf{d} \backslash \{i\}) \cap T \neq \emptyset]$$

where the first term is the probability that the first index is in $T$ (and thus the outcome must be in $\mathcal{T}$), then we consider all non-$\bot$ possibilities for the first index, and take the probability of that event and multiply it by the probability that one of the remaining indices is in $T$ as the peeling process proceeds (and thus the outcome would be in $\mathcal{T}$). Multiplying through this summation by $\left(\Pr[\mathtt{LEM}^{\bar{k}}(\mathbf{h}, T) \neq \bot]\right)^{-1}$, we apply our inductive hypothesis to achieve

$$\left(\Pr[\mathtt{LEM}^{\bar{k}}(\mathbf{h}, T) \neq \bot]\right)^{-1} \Pr[\mathtt{pLEM}^{k-1,\bar{k}}(\mathbf{h}, \mathbf{d} \backslash \{i\}) \in \mathcal{T}] \leq 1$$

where our inductive hypothesis was on all $\mathbf{d}$ such that $T \subseteq \mathbf{d}$ and we must have $T \subseteq \mathbf{d} \backslash \{i\}$ because $i \in \mathbf{d} \backslash T$. Therefore, we can bound

$$\left(\Pr[\mathtt{LEM}^{\bar{k}}(\mathbf{h}, T) \neq \bot]\right)^{-1} \Pr[\mathtt{pLEM}^{k,\bar{k}}(\mathbf{h}, \mathbf{d}) \in \mathcal{T}]$$
$$\leq \left(\Pr[\mathtt{LEM}^{\bar{k}}(\mathbf{h}, T) \neq \bot]\right)^{-1} \Pr[\mathtt{LEM}^{\bar{k}}(\mathbf{h}, \mathbf{d}) \in T] + \sum_{i \in \mathbf{d} \backslash T} \Pr[\mathtt{LEM}^{\bar{k}}(\mathbf{h}, \mathbf{d}) = i]$$

Applying Definition 5.1, we can explicitly write both terms and obtain

$$\left(\Pr[\mathtt{LEM}^{\bar{k}}(\mathbf{h}, T) \neq \bot]\right)^{-1} \Pr[\mathtt{pLEM}^{k,\bar{k}}(\mathbf{h}, \mathbf{d}) \in \mathcal{T}]$$
$$\leq \frac{\exp(\varepsilon h_\bot) + \sum_{i \in T} \exp(\varepsilon h_i)}{\exp(\varepsilon h_\bot) + \sum_{i \in \mathbf{d}} \exp(\varepsilon h_i)} + \frac{\sum_{i \in \mathbf{d} \backslash T} \exp(\varepsilon h_i)}{\exp(\varepsilon h_\bot) + \sum_{i \in \mathbf{d}} \exp(\varepsilon h_i)} = 1$$

$\square$

*Proof of Lemma 5.5.* In the application of Lemma C.2, we set $T = \mathbf{d}_\delta$ as in Lemma C.1 and $\mathbf{d} = \mathbf{d}^{\bar{k}}(\mathbf{h})$, in which case $\mathcal{T} = \mathcal{S}^\delta$ from Definition 5.2. This gives

$$\Pr[\mathtt{pLEM}^{k,\bar{k}}(\mathbf{h}, \mathbf{d}^{\bar{k}}(\mathbf{h})) \in \mathcal{S}^\delta] \leq \Pr[\mathtt{LEM}^{\bar{k}}(\mathbf{h}, \mathbf{d}_\delta) \neq \bot]$$

and our bound follows from Lemma C.1 $\square$

## C.2   Proof of Lemma 5.6

**Lemma C.3.** *Consider a subset $T \subseteq [d]$, and domain $\mathbf{d} \subseteq [d]$. For histogram $\mathbf{h}$, we will write the outcome set of $pLEM^{k,\bar{k}}(\mathbf{h}, \mathbf{d})$ as $\mathcal{O}$ and define the set $\mathcal{T} = \{o \in \mathcal{O} : o \cap T \neq \emptyset\}$. For any $o = (i_1, ..., i_\ell) \notin \mathcal{T}$ we have*

$$\Pr[pLEM^{k,\bar{k}}(\mathbf{h}, \mathbf{d}) = o | pLEM^{k,\bar{k}}(\mathbf{h}, \mathbf{d}) \notin \mathcal{T}]$$
$$= \prod_{j=0}^{\ell-1} \Pr[LEM^{\bar{k}}(\mathbf{h}, \mathbf{d} \setminus \{i_1, ..., i_j\}) = i_{j+1} | LEM^{\bar{k}}(\mathbf{h}, \mathbf{d} \setminus \{i_1, ..., i_j\}) \notin T]$$

*Proof.* We can rewrite the event $\{pLEM^{k,\bar{k}}(\mathbf{h}, \mathbf{d}) = o\}$ as the intersection of independent events $\{LEM^{\bar{k}}(\mathbf{h}, \mathbf{d}) = i_1\} \cap \{pLEM^{k-1,\bar{k}}(\mathbf{h}, \mathbf{d} \setminus \{i_1\}) = (i_2, ..., i_\ell)\}$. Similarly, we can rewrite as independent events

$$\{pLEM^{k,\bar{k}}(\mathbf{h}, \mathbf{d}) \notin \mathcal{T}\} = \{LEM^{\bar{k}}(\mathbf{h}, \mathbf{d}) \notin T\} \cap \left\{ \bigcap_{i \in \mathbf{d} \setminus T} \{pLEM^{k-1,\bar{k}}(\mathbf{h}, \mathbf{d} \setminus \{i\}) \notin \mathcal{T}\} \right\}$$

Therefore, we can rewrite our probability statement as

$$\Pr[pLEM^{k,\bar{k}}(\mathbf{h}, \mathbf{d}) = o | pLEM^{k,\bar{k}}(\mathbf{h}, \mathbf{d}) \notin \mathcal{T}] = \Pr[A_1 \cap A_2 | B_1 \cap B_2]$$

where $A_1 = \{LEM^{\bar{k}}(\mathbf{h}, \mathbf{d}) = i_1\}, A_2 = \{pLEM^{k-1,\bar{k}}(\mathbf{h}, \mathbf{d} \setminus \{i_1\}) = (i_2, ..., i_\ell)\}$ and also $B_1 = \{LEM^{\bar{k}}(\mathbf{h}, \mathbf{d}) \notin T\}, B_2 = \{pLEM^{k-1,\bar{k}}(\mathbf{h}, \mathbf{d} \setminus \{i_1\}) \notin \mathcal{T}\}$. Accordingly, we only have that $A_1$ is dependent on $B_1$ and $A_2$ is dependent on $B_2$ with everything else being pairwise independent. It is straightforward to then show that

$$\Pr[A_1 \cap A_2 | B_1 \cap B_2] = \Pr[A_1 | B_1] \Pr[A_2 | B_2]$$

Substituting back in for our variables then gives

$$\Pr[pLEM^{k,\bar{k}}(\mathbf{h}, \mathbf{d}) = o | pLEM^{k,\bar{k}}(\mathbf{h}, \mathbf{d}) \notin \mathcal{T}]$$
$$= \Pr[LEM^{\bar{k}}(\mathbf{h}, \mathbf{d}) = i_1 | LEM^{\bar{k}}(\mathbf{h}, \mathbf{d}) \notin T]$$
$$\cdot \Pr[pLEM^{k-1,\bar{k}}(\mathbf{h}, \mathbf{d} \setminus \{i_1\}) = (i_2, ..., i_\ell) | pLEM^{k-1,\bar{k}}(\mathbf{h}, \mathbf{d} \setminus \{i_1\}) \notin \mathcal{T}]$$

Applying this argument inductively (where the base case of $k = 1$ is true by definition of our peeling exponential mechanism) then gives our desired claim. $\qquad \square$

**Lemma C.4.** *For any $\mathbf{d} \subseteq [d]$ and $T \subseteq [d]$, along with outcome $j \in \{[d] \cup \{\bot\}\} \setminus T$, we have for any $\mathbf{h}$*

$$\Pr[LEM^{\bar{k}}(\mathbf{h}, \mathbf{d}) = j | LEM^{\bar{k}}(\mathbf{h}, \mathbf{d}) \notin T] = \Pr[LEM^{\bar{k}}(\mathbf{h}, \mathbf{d} \setminus T) = j]$$

*Proof.* From the definition of conditional probabilities we have

$$\Pr[\text{LEM}^{\bar{k}}(\mathbf{h}, \mathbf{d}) = j | \text{LEM}^{\bar{k}}(\mathbf{h}, \mathbf{d}) \notin T] = \frac{\Pr[\{\text{LEM}^{\bar{k}}(\mathbf{h}, \mathbf{d}) = j\} \cap \{\text{LEM}^{\bar{k}}(\mathbf{h}, \mathbf{d}) \notin T\}]}{\Pr[\text{LEM}^{\bar{k}}(\mathbf{h}, \mathbf{d}) \notin T]}$$

By our assumption that $j \notin T$ we can reduce this to

$$\Pr[\text{LEM}^{\bar{k}}(\mathbf{h}, \mathbf{d}) = j | \text{LEM}^{\bar{k}}(\mathbf{h}, \mathbf{d}) \notin T] = \frac{\Pr[\text{LEM}^{\bar{k}}(\mathbf{h}, \mathbf{d}) = j]}{\Pr[\text{LEM}^{\bar{k}}(\mathbf{h}, \mathbf{d}) \notin T]}$$

Applying Definition 5.1, we can explicitly write both terms and obtain

$$\Pr[\text{LEM}^{\bar{k}}(\mathbf{h}, \mathbf{d}) = j | \text{LEM}^{\bar{k}}(\mathbf{h}, \mathbf{d}) \notin T] = \frac{\frac{\exp(\varepsilon h_j)}{\exp(\varepsilon h_\perp) + \sum_{i \in \mathbf{d}} \exp(\varepsilon h_i)}}{\frac{\exp(\varepsilon h_\perp) + \sum_{i \in \mathbf{d} \setminus T} \exp(\varepsilon h_i)}{\exp(\varepsilon h_\perp) + \sum_{i \in \mathbf{d}} \exp(\varepsilon h_i)}} = \frac{\exp(\varepsilon h_j)}{\exp(\varepsilon h_\perp) + \sum_{i \in \mathbf{d} \setminus T} \exp(\varepsilon h_i)}$$

where because $j \notin T$, this then reduces to $\Pr[\text{LEM}^{\bar{k}}(\mathbf{h}, \mathbf{d} \setminus T) = j]$ as desired.

$\square$

**Corollary C.1.** *Consider a subset $T \subseteq [d]$, and domain $\mathbf{d} \subseteq [d]$. For histogram $\mathbf{h}$, we will write the outcome set of $pLEM^{k,\bar{k}}(\mathbf{h}, \mathbf{d})$ as $\mathcal{O}$ and define the set $\mathcal{T} = \{o \in \mathcal{O} : o \cap T \neq \emptyset\}$. For any $o = (i_1, ..., i_\ell) \notin \mathcal{T}$ we have*

$$\Pr[pLEM^{k,\bar{k}}(\mathbf{h}, \mathbf{d}) = o] = \Pr[pLEM^{k,\bar{k}}(\mathbf{h}, \mathbf{d} \setminus T) = o] \Pr[pLEM^{k,\bar{k}}(\mathbf{h}, \mathbf{d}) \notin \mathcal{T}]$$

*Proof.* We first use the fact that $o \notin \mathcal{T}$ to rewrite our probability as

$$\Pr[\text{pLEM}^{k,\bar{k}}(\mathbf{h}, \mathbf{d}) = o] = \Pr[\{\text{pLEM}^{k,\bar{k}}(\mathbf{h}, \mathbf{d}) = o\} \cap \{\text{pLEM}^{k,\bar{k}}(\mathbf{h}, \mathbf{d}) \notin \mathcal{T}\}]$$
$$= \Pr[\text{pLEM}^{k,\bar{k}}(\mathbf{h}, \mathbf{d}) = o | \text{pLEM}^{k,\bar{k}}(\mathbf{h}, \mathbf{d}) \notin \mathcal{T}] \Pr[\text{pLEM}^{k,\bar{k}}(\mathbf{h}, \mathbf{d}) \notin \mathcal{T}]$$

We then apply Lemma C.3 and this gives

$$\Pr[\text{pLEM}^{k,\bar{k}}(\mathbf{h}, \mathbf{d}) = o]$$
$$= \prod_{j=0}^{\ell-1} \Pr[\text{LEM}^{\bar{k}}(\mathbf{h}, \mathbf{d} \setminus \{i_1, ..., i_j\}) = i_{j+1} | \text{LEM}^{\bar{k}}(\mathbf{h}, \mathbf{d} \setminus \{i_1, ..., i_j\}) \notin T] \cdot \Pr[\text{pLEM}^{k,\bar{k}}(\mathbf{h}, \mathbf{d}) \notin \mathcal{T}]$$

We then apply Lemma C.4 to achieve

$$\Pr[\text{pLEM}^{k,\bar{k}}(\mathbf{h}, \mathbf{d}) = o] = \prod_{j=0}^{\ell-1} \Pr[\text{LEM}^{\bar{k}}(\mathbf{h}, (\mathbf{d} \setminus T) \setminus \{i_1, ..., i_j\}) = i_{j+1}] \Pr[\text{pLEM}^{k,\bar{k}}(\mathbf{h}, \mathbf{d}) \notin \mathcal{T}]$$

Using our construction of the peeling mechanism then implies our desired equality.

$\square$

*Proof of Lemma 5.6.* We will set $\mathbf{d} = \mathbf{d}^{\bar{k}}(\mathbf{h})$ and $T = \mathbf{d}^{\bar{k}}(\mathbf{h}) \setminus \mathbf{d}^{\bar{k}}(\mathbf{h}')$ in our Corollary C.1 and our desired result is immediately implied.

$\square$

# D  Omitted Proofs from Section 6.1

## D.1  Proof of Lemma 6.3

**Lemma D.1.** *Given an histogram* $\mathbf{h}$ *and some* $\mathbf{d} \subseteq [d]$. *For any* $i \in \mathbf{d}$ *such that* $h_i \leq h_{(\bar{k}+1)} + 1$, *then*

$$\Pr[i \in \textit{LapMax}^{k,\bar{k}}(\mathbf{h}, \mathbf{d})] \leq \frac{3\delta}{4\Delta} + \frac{\ln(\Delta/\delta)\delta}{4\Delta}$$

*Proof.* For simplicity, we will set $T = \ln(\frac{\Delta}{\delta})/\varepsilon$, which implies $h_\perp = h_{(\bar{k}+1)} + 1 + T$ and plug back in at the end of the analysis. By construction of our mechanism, we know that the noisy estimate of $h_i$ must be greater than the noisy estimate of our threshold $h_\perp = h_{(\bar{k}+1)} + 1 + T$ to be a possible output, which implies

$$\Pr[i \in \texttt{LapMax}^{k,\bar{k}}(\mathbf{h}, \mathbf{d})] \leq \Pr[h_i + \texttt{Lap}(1/\varepsilon) > h_\perp + \texttt{Lap}(1/\varepsilon)]$$

By assumption, $h_i \leq h_{(\bar{k}+1)} + 1$, which gives us

$$\Pr[i \in \texttt{LapMax}^{k,\bar{k}}(\mathbf{h}, \mathbf{d})] \leq \Pr[\texttt{Lap}(1/\varepsilon) > T + \texttt{Lap}(1/\varepsilon)].$$

We can then rewrite this as the convolution of two Laplace random variables. We will denote the density of a $\texttt{Lap}(1/\varepsilon)$ random variable as $p(\cdot)$.

$$\int_{-\infty}^{\infty} p(x)\Pr[\texttt{Lap}(1/\varepsilon) < x - T]dx \tag{10}$$

$$= \int_{-\infty}^{0} p(x)\Pr[\texttt{Lap}(1/\varepsilon) < x - T]dx + \int_{0}^{T} p(x)\Pr[\texttt{Lap}(1/\varepsilon) < x - T]dx \tag{11}$$

$$+ \int_{T}^{\infty} p(x)\Pr[\texttt{Lap}(1/\varepsilon) < x - T]dx \tag{12}$$

We will bound each separately. First, note that $\Pr[\texttt{Lap}(1/\varepsilon) < -T] = \frac{\delta}{2\Delta}$, which implies that

$$\int_{-\infty}^{0} p(x)\Pr[\texttt{Lap}(1/\varepsilon) < x - T]dx \leq \frac{\delta}{2\Delta} \int_{-\infty}^{0} p(x) = \frac{\delta}{4\Delta}$$

where the last step follows from the symmetry of the Laplace distribution where $\int_{-\infty}^{0} p(x) = \frac{1}{2}$. By similar reasoning, we have

$$\int_{T}^{\infty} p(x)\Pr[\texttt{Lap}(1/\varepsilon) < x - T]dx \leq \int_{T}^{\infty} p(x)dx = \frac{\delta}{2\Delta}$$

The middle term in (12) will be a bit trickier to bound, and we will need to apply the explicit form of the Laplace distribution. Rewriting $\Pr[\texttt{Lap}(1/\varepsilon) < x - T] = \int_{-\infty}^{x-T} p(y)dy$, we then use the fact that $x - T \leq 0$ for $x \in [0, T]$, and it is straightforward to see that plugging in the Laplace pdf to this integral evaluates to the following for $x \in [0, T]$

$$\Pr[\texttt{Lap}(1/\varepsilon) < x - T] = \int_{-\infty}^{x-T} p(y)dy = \frac{1}{2}e^{(x-T)\varepsilon}.$$

We then plug this into the final term we want to bound and get

$$\int_0^T p(x)\Pr[\mathtt{Lap}(1/\varepsilon) < x - T]dx = \int_0^T p(x)\frac{1}{2}e^{(x-T)\varepsilon}dx$$

Furthermore, we plug in the PDF for the Laplace distribution with the absolute value eliminated because $x \in [0, T]$, which reduces to

$$\begin{aligned}\int_0^T p(x)\Pr[\mathtt{Lap}(1/\varepsilon) < x - T]dx &= \int_0^T \frac{\varepsilon}{2}e^{-x\varepsilon}\frac{1}{2}e^{(x-T)\varepsilon}dx \\ &= \int_0^T \frac{\varepsilon}{4}e^{-T\varepsilon}dx = \frac{\varepsilon T}{4}e^{-\varepsilon T} \\ &= \frac{\delta}{4\Delta}\ln(\tfrac{\Delta}{\delta})\end{aligned}$$

Combining these inequalities and plugging back in for $\delta$ easily gives the desired bound. $\qquad\square$

*Proof of Lemma 6.3.* This will follow from a simple union bound on each $i \in \mathbf{d}^{\bar{k}}(\mathbf{h}) \setminus \mathbf{d}^{\bar{k}}(\mathbf{h}')$ where we consider each subset of $\mathcal{S}_{\mathtt{Lap}}^\delta$ such that each outcome contains $i$, or more formally we define $\mathcal{S}_{\mathtt{Lap}}^\delta(i) := \{o \in \mathcal{S}_{\mathtt{Lap}}^\delta : i \in o\}$ This then implies that

$$\Pr[\mathtt{LapMax}^{k,\bar{k}}(\mathbf{h}, \mathbf{d}^{\bar{k}}(\mathbf{h})) \in \mathcal{S}_{\mathtt{Lap}}^\delta] \le \sum_{i \in \mathbf{d}^{\bar{k}}(\mathbf{h}')\setminus \mathbf{d}^{\bar{k}}(\mathbf{h})} \Pr[\mathtt{LapMax}^{k,\bar{k}}(\mathbf{h}, \mathbf{d}^{\bar{k}}(\mathbf{h})) \in \mathcal{S}_{\mathtt{Lap}}^\delta(i)]$$

because each outcome $o \in \mathcal{S}_{\mathtt{Lap}}^\delta$ must contain some $i \in \mathbf{d}^{\bar{k}}(\mathbf{h}) \setminus \mathbf{d}^{\bar{k}}(\mathbf{h}')$ by construction. Furthermore, by construction we also have

$$\Pr[\mathtt{LapMax}^{k,\bar{k}}(\mathbf{h}, \mathbf{d}^{\bar{k}}(\mathbf{h})) \in \mathcal{S}_{\mathtt{Lap}}^\delta(i)] = \Pr[i \in \mathtt{LapMax}^{k,\bar{k}}(\mathbf{h}, \mathbf{d}^{\bar{k}}(\mathbf{h}))]$$

Our claim then immediately follows from Lemma D.1 and the fact that the size of $\mathbf{d}^{\bar{k}}(\mathbf{h})\setminus\mathbf{d}^{\bar{k}}(\mathbf{h}')$ is at most $\Delta$ by Lemma 5.3 and our assumption that $\min\{\Delta, \bar{k}\} = \Delta$. $\qquad\square$

## D.2  Proof of Lemma 6.4

*Proof of Lemma 6.4.* For any $o = (i_1, ..., i_\ell) \in S$ we know that by definition of $\mathcal{S}_{\mathtt{Lap}} \cap \mathcal{S}'_{\mathtt{Lap}}$ we must have each $i_j \in \mathbf{d}^{\bar{k}}(\mathbf{h}) \cup \{\bot\}$ and also $i_j \in \mathbf{d}^\varepsilon \cup \{\bot\}$. Furthermore, letting $p(\cdot)$ be the PDF for a $\mathtt{Lap}(1/\varepsilon)$ random variable, we know

$$\Pr[\mathtt{LapMax}^{k,\bar{k}}(\mathbf{h}, \mathbf{d}^{\bar{k}}(\mathbf{h})) = o] =$$
$$\int_{-\infty}^{\infty} p(x_1 - h_{i_1})\int_{-\infty}^{x_1} p(x_2 - h_{i_2})\cdots\int_{-\infty}^{x_{\ell-1}} p(x_\ell - h_{i_\ell})\prod_{j \in \{\mathbf{d}^{\bar{k}}(\mathbf{h})\cup\{\bot\}\}\setminus\{o\}} \Pr[\mathtt{Lap}(1/\varepsilon) < x_\ell - h_{i_\ell}]dx_\ell\cdots dx_1$$

We then note that $\{\mathbf{d}^\varepsilon \cup \{\bot\}\} \setminus \{o\}$ is a subset of $\{\mathbf{d}^{\bar{k}}(\mathbf{h}) \cup \{\bot\}\} \setminus \{o\}$ so the only difference is that the product contains more probabilities for $\Pr[\mathtt{LapMax}^{k,\bar{k}}(\mathbf{h}, \mathbf{d}^{\bar{k}}(\mathbf{h})) = o]$, which implies

$$\Pr[\mathtt{LapMax}^{k,\bar{k}}(\mathbf{h}, \mathbf{d}^{\bar{k}}(\mathbf{h})) = o] \leq \Pr[\mathtt{LapMax}^{k,\bar{k}}(\mathbf{h}, \mathbf{d}^{\varepsilon}) = o]$$

and hence our first inequality.

For the second inequality, we will prove by contradiction and suppose that there is some $S \subseteq \mathcal{S}_{\mathtt{Lap}} \cap \mathcal{S}'_{\mathtt{Lap}}$ such that

$$\Pr[\mathtt{LapMax}^{k,\bar{k}}(\mathbf{h}, \mathbf{d}^{\varepsilon}) \in S] > \Pr[\mathtt{LapMax}^{k,\bar{k}}(\mathbf{h}, \mathbf{d}^{\bar{k}}(\mathbf{h})) \in S] + \bar{\delta}$$

From our first inequality, we know that if we let $\bar{S} = \{\mathcal{S}_{\mathtt{Lap}} \cap \mathcal{S}'_{\mathtt{Lap}}\} \setminus S$ then

$$\Pr[\mathtt{LapMax}^{k,\bar{k}}(\mathbf{h}, \mathbf{d}^{\varepsilon}) \in \bar{S}] \geq \Pr[\mathtt{LapMax}^{k,\bar{k}}(\mathbf{h}, \mathbf{d}^{\bar{k}}(\mathbf{h})) \in \bar{S}]$$

and by summing together the two inequalities implies

$$\Pr[\mathtt{LapMax}^{k,\bar{k}}(\mathbf{h}, \mathbf{d}^{\varepsilon}) \in \mathcal{S}_{\mathtt{Lap}} \cap \mathcal{S}'_{\mathtt{Lap}}] > \Pr[\mathtt{LapMax}^{k,\bar{k}}(\mathbf{h}, \mathbf{d}^{\bar{k}}(\mathbf{h})) \in \mathcal{S}_{\mathtt{Lap}} \cap \mathcal{S}'_{\mathtt{Lap}}] + \bar{\delta}$$

From Lemma 6.3 we then conclude

$$\Pr[\mathtt{LapMax}^{k,\bar{k}}(\mathbf{h}, \mathbf{d}^{\varepsilon}) \in \mathcal{S}_{\mathtt{Lap}} \cap \mathcal{S}'_{\mathtt{Lap}}] >$$
$$\Pr[\mathtt{LapMax}^{k,\bar{k}}(\mathbf{h}, \mathbf{d}^{\bar{k}}(\mathbf{h})) \in \mathcal{S}_{\mathtt{Lap}} \cap \mathcal{S}'_{\mathtt{Lap}}] + \Pr[\mathtt{LapMax}^{k,\bar{k}}(\mathbf{h}, \mathbf{d}^{\bar{k}}(\mathbf{h})) \in \mathcal{S}^{\delta}_{\mathtt{Lap}}]$$

and this gives our contradiction because

$$\Pr[\mathtt{LapMax}^{k,\bar{k}}(\mathbf{h}, \mathbf{d}^{\bar{k}}(\mathbf{h})) \in \mathcal{S}_{\mathtt{Lap}} \cap \mathcal{S}'_{\mathtt{Lap}}] + \Pr[\mathtt{LapMax}^{k,\bar{k}}(\mathbf{h}, \mathbf{d}^{\bar{k}}(\mathbf{h})) \in \mathcal{S}^{\delta}_{\mathtt{Lap}}] = 1$$

$\square$

# E  Further Accuracy Guarantees

We will give a few additional accuracy guarantees, complementing results in Section 8 regarding correctly outputting the true top index first. More specifically, we will look at the conditions under which our algorithm returns the true top index rather than the traditional exponential mechanism, which has access to the full histogram. Additionally, we show that the probability of incorrectly outputting an index other than the true top index or $\perp$ will only be smaller in our algorithm versus the exponential mechanism. Throughout this section we will write $\mathtt{EM}$ be the exponential mechanism with quality score $q(\mathbf{h}, i) = h_i$.

**Lemma E.1.** *Given histogram $\mathbf{h}$ where $i_{(1)}$ is the index of $h_{(1)}$, then*

$$\Pr[\mathit{LEM}^{\bar{k}}(\mathbf{h}, \mathbf{d}^{\bar{k}}(\mathbf{h})) = i_{(1)}] \geq \Pr[\mathit{EM}(\mathbf{h}) = i_{(1)}]$$

*iff we have*

$$\frac{\min\{\Delta, \bar{k}\} e^{\varepsilon}}{\delta} \cdot \exp(\varepsilon h_{(\bar{k}+1)}) \leq \sum_{j > \bar{k}} \exp(\varepsilon h_{(j)})$$

*Proof.* This follows immediately from the fact that with $h_\perp = h_{(\bar{k}+1)} + 1 + \frac{\ln(\min\{\Delta,\bar{k}\}/\delta)}{\varepsilon}$, we have

$$\Pr[\text{LEM}^{\bar{k}}(\mathbf{h}, \mathbf{d}^{\bar{k}}(\mathbf{h})) = i_{(1)}] = \frac{\exp(\varepsilon h_{(1)})}{\exp(\varepsilon h_\perp) + \sum_{j \leq \bar{k}} \exp(\varepsilon h_{(j)})}$$

and

$$\Pr[\text{EM}(\mathbf{h}) = i_{(1)}] = \frac{\exp(\varepsilon h_{(1)})}{\sum_{j \leq d} \exp(\varepsilon h_{(j)})}$$

$\square$

In contrast, if we consider $\perp$ to be a null event, then the probability that our variant will output an incorrect index will always be smaller than for the peeling mechanism.

**Lemma E.2.** *Given histogram* $\mathbf{h}$ *where* $i_{(1)}$ *is the index of* $h_{(1)}$, *then*

$$\Pr[\text{LEM}^{\bar{k}}(\mathbf{h}, \mathbf{d}^{\bar{k}}(\mathbf{h})) \notin \{i_{(1)}, \perp\}] < \Pr[\text{EM}(\mathbf{h}) \neq i_{(1)}]$$

*Proof.* Writing the explicit form of each we have

$$\Pr[\text{LEM}^{\bar{k}}(\mathbf{h}, \mathbf{d}^{\bar{k}}(\mathbf{h})) \notin \{i_{(1)}, \perp\}] = \frac{\sum_{1 < j \leq \bar{k}} \exp(\varepsilon h_{(j)})}{\exp(\varepsilon h_\perp) + \sum_{j \leq \bar{k}} \exp(\varepsilon h_{(j)})}$$

and

$$\Pr[\text{EM}(\mathbf{h}) \neq i_{(1)}] = \frac{\sum_{1 < j \leq d} \exp(\varepsilon h_{(j)})}{\sum_{j \leq d} \exp(\varepsilon h_{(j)})}$$

Multiply each side by the denominator and cancelling like terms, we get that the inequality is equivalent to

$$\left( \sum_{1 < j \leq \bar{k}} \exp(\varepsilon h_{(j)}) \right) \exp(\varepsilon h_{(1)}) < \left( \sum_{1 < j \leq d} \exp(\varepsilon h_{(j)}) \right) \left( \exp(\varepsilon h_{(1)}) + \exp(\varepsilon h_\perp) \right)$$

which holds.

$\square$

# F    Fixed Threshold Mechanism

We also consider a mechanism that keeps the threshold fixed, rather than adding noise to it, which may be of independent interest since it requires a slightly different analysis than our main, randomized threshold algorithm. Furthermore, it requires a smaller additive factor to the threshold (an additive savings of $\ln(2)/\varepsilon$), which along with the fact that the threshold is fixed, increases the probability that the noisy values of considered indices are above this threshold. However, it requires us to set $\bar{k} = k$, cannot return an ordering on the indices, and also does not achieve the same *range-bounded* composition or *pay-what-you-get* composition. As such, the primary application of this algorithm would only be in the setting in which $k$ is small and the user only wanted to know indices within the top-$k$, and would prefer being returned $\perp$ as opposed to an index not in the top-$k$.

We start with a basic property of returning a noisy count that is above a data dependent threshold.

**Lemma F.1.** *For any neighboring databases* $\mathbf{h}, \mathbf{h}'$*, any fixed* $k < d$*, and any* $T \in \mathbb{R}$*, we have the following for any* $i \in [d]$

$$\Pr[h_i + \textbf{\textit{Lap}}(1/\varepsilon) > h_{(k+1)} + T] \leq e^\varepsilon \Pr[h_i' + \textbf{\textit{Lap}}(1/\varepsilon) > h_{(k+1)}' + T]$$

This result is not entirely trivial because the threshold being considered is data-dependent and not fixed across the mechanism.

*Proof.* Follows immediately from the fact that $|(h_i - h_{(k+1)}) - (h_i' - h_{(k+1)}')| \leq 1$ and using known properties of the Laplace distribution. $\square$

We will connect our fixed threshold mechanism to a simple randomized response for each $i \in [d]$, but at most $k$ will have non-zero probabilities.

**Definition F.1.** *For index* $i \in [d]$*, and some fixed value* $k < d$ *and given* $\varepsilon, \delta > 0$*, we define the truncated randomized response mechanism* $\textbf{\textit{tRR}}_i^k : \mathbb{N}^d \to \{i, \perp\}$*, such that*

$$\Pr[\textbf{\textit{tRR}}_i^k(\mathbf{h}) = i] = \begin{cases} \Pr\left[h_i + \textbf{\textit{Lap}}(1/\varepsilon) > h_{(k+1)} + 1 + \frac{\ln(\frac{1}{2\delta})}{\varepsilon}\right] & \text{if } h_i > h_{(k+1)} \\ 0 & \text{otherwise} \end{cases}$$

We then have the following properties of the truncated randomized response. Recall that we defined the strictly limited domain $\mathbf{d}_>^k(\mathbf{h})$ in (9).

**Lemma F.2.** *For a fixed* $i \in [d]$ *and fixed* $k < d$*, along with given* $\varepsilon, \delta > 0$*, then for any neighboring histograms* $\mathbf{h}, \mathbf{h}'$

1. $\Pr[\textbf{\textit{tRR}}_i^k(\mathbf{h}) = i] = \Pr[\textbf{\textit{tRR}}_i^k(\mathbf{h}') = i] = 0$ *if* $i \notin \mathbf{d}_>^k(\mathbf{h}) \cup \mathbf{d}_>^k(\mathbf{h}')$

2. $\Pr[\textbf{\textit{tRR}}_i^k(\mathbf{h}) = i] = \delta$ *if* $i \in \mathbf{d}_>^k(\mathbf{h}) \setminus \mathbf{d}_>^k(\mathbf{h}')$ *or* $\Pr[\textbf{\textit{tRR}}_i^k(\mathbf{h}') = i] = \delta$ *if* $i \in \mathbf{d}_>^k(\mathbf{h}') \setminus \mathbf{d}_>^k(\mathbf{h})$.

3. $\Pr[\textbf{\textit{tRR}}_i^k(\mathbf{h}) = i] \leq e^\varepsilon \Pr[\textbf{\textit{tRR}}_i^k(\mathbf{h}') = i]$ *if* $i \in \mathbf{d}_>^k(\mathbf{h}) \cap \mathbf{d}_>^k(\mathbf{h}')$

*Proof.* Item 1. follows from the definition of $\textbf{\textit{tRR}}_i^k$ when $i \notin \mathbf{d}_>^k(\mathbf{h}) \cup \mathbf{d}_>^k(\mathbf{h}')$ and item 3. follows from Lemma F.1.

We now focus on item 2. Without loss of generality assume that $\mathbf{h} \geq \mathbf{h}'$ and let $i \in \mathbf{d}_>^k(\mathbf{h}) \setminus \mathbf{d}_>^k(\mathbf{h}')$. In this case, we must have $h_i' \leq h_{(k+1)}'$ yet $h_i > h_{(k+1)}$. If $h_i = h_i'$, then $h_{(k+1)} < h_{(k+1)}'$, which cannot happen. Thus, $h_i = h_i' + 1$, in which case $h_{(k+1)} < h_{(k+1)}' + 1$. Since $h_{(k+1)} = h_{(k+1)}'$ or $h_{(k+1)} = h_{(k+1)} + 1$ from Lemma 5.1, we must be in the $h_{(k+1)} = h_{(k+1)}'$ case. Stringing the inequalities, we have

$$h_{(k+1)} < h_i = h_i' + 1 \leq h_{(k+1)}' + 1 = h_{(k+1)} + 1$$

Since $h_i$ must be integral, we have $h_i = h_{(k+1)} + 1$. We then have the following

$$\Pr[\textbf{\textit{tRR}}_i^k(\mathbf{h}) = i] = \Pr\left[h_i + \textbf{\textit{Lap}}(1/\varepsilon) > h_{(k+1)} + 1 + \frac{\ln(\frac{1}{2\delta})}{\varepsilon}\right]$$

$$= \Pr\left[\textbf{\textit{Lap}}(1/\varepsilon) > \frac{\ln(\frac{1}{2\delta})}{\varepsilon}\right]$$

$$= \delta$$

$\square$

**Corollary F.1.** *For every $i \in [d]$ and fixed $k$, along with given $\varepsilon, \delta > 0$, the algorithm $\mathtt{tRR}_i^k$ is $(\varepsilon, \delta)$-DP*

Recall, that we will write our input to our mechanism as a histogram $\mathbf{h} \in \mathbb{N}^d$. Accordingly, we note that our randomized response mechanism $\mathtt{tRR}_i^k$ will only return $i$ with non-zero probability if $i \in \mathbf{d}_>^k(\mathbf{h})$, where by definition $|\mathbf{d}_>^k(\mathbf{h})| \leq k$, so we will only need to draw randomness for these indices and are not required to consider the entire histogram.

---

**Algorithm 8** Fixed Threshold at level $k$, $\mathtt{fT}^k$

---

**Input:** Histogram $\mathbf{h} \in \mathbb{N}^d$, and parameters $k, \varepsilon, \delta$.
**Output:** Set of indices $D$.
Set $h_\perp = h_{(k+1)} + 1 + \ln(1/2\delta)/\varepsilon$
Set $D = \emptyset$
**for** $i \leq k$ **do**
    **if** $h_{(i)} > h_\perp$ **then**
        Draw $r_i \sim Lap(1/\varepsilon)$
        **if** $h_{(i)} + r_i > h_\perp$ **then**
            $D \leftarrow D \cup i$
Output $D$

---

We first more formally define this mechanism $\mathtt{fT}^k$ in Algorithm 8. Note that we can connect $\mathtt{fT}^k$ with the randomized response algorithm $\mathtt{tRR}_i^k$ for any integer $k < d$ in the following way for any input histogram $\mathbf{h}$ and any outcome $D \subseteq [d]$,

$$\Pr[\mathtt{fT}^k(\mathbf{h}) = D] = \prod_{i \in D} \Pr[\mathtt{tRR}_i^k(\mathbf{h}) = i] \prod_{i \in [d] \setminus D} \Pr[\mathtt{tRR}_i^k = \perp].$$

**Definition F.2.** *We will restrict the domain and range of our fixed threshold mechanism to a subset of $[d]$. We fix $k < d$. Consider some histogram $\widehat{\mathbf{h}}$ and it's corresponding domain $\mathbf{d}_>^k(\widehat{\mathbf{h}}) \subseteq [d]$.*

$$\mathcal{H}_k(\widehat{\mathbf{h}}) := \left\{ \mathbf{h} \in \mathbb{N}^d : \mathbf{d}_>^k(\mathbf{h}) \subseteq \mathbf{d}_>^k(\widehat{\mathbf{h}}) \right\}.$$

*Let $\pi_{\widehat{\mathbf{h}}} : \mathbf{d}_>^k(\widehat{\mathbf{h}}) \to [|\mathbf{d}_>^k(\widehat{\mathbf{h}})|]$ be an invertible mapping. We define the fixed threshold mechanism limited to domain for some fixed histogram $\widehat{\mathbf{h}}$ to be $\mathtt{fT}^k|_{\mathbf{d}_>^k(\widehat{\mathbf{h}})} : \mathcal{H}_k(\widehat{\mathbf{h}}) \to \{\perp, 1\}^{|\mathbf{d}_>^k(\widehat{\mathbf{h}})|}$ for any integer $k \leq d$, such that for some input histogram $\mathbf{h}$ and any $\mathbf{y} \in \{\perp, 1\}^{|\mathbf{d}_>^k(\widehat{\mathbf{h}})|}$,*

$$\Pr[\mathtt{fT}^k|_{\mathbf{d}_>^k(\widehat{\mathbf{h}})}(\mathbf{h}) = \mathbf{y}] = \prod_{i : y_i = 1} \Pr[\mathtt{tRR}_{\pi_{\widehat{\mathbf{h}}}(i)}^k(\mathbf{h}) = \pi_{\widehat{\mathbf{h}}}(i)] \prod_{i : y_i = \perp} \Pr[\mathtt{tRR}_{\pi_{\widehat{\mathbf{h}}}(i)}^k(\mathbf{h}) = \perp]$$

**Lemma F.3.** *Fix a histogram $\widehat{\mathbf{h}}$ and $k < d$. The fixed threshold mechanism limited to a domain $\mathtt{fT}^k|_{\mathbf{d}^k(\widehat{\mathbf{h}})}$ can be written in terms of $\mathtt{tRR}_i^k$ for each $i \in \mathbf{d}_>^k(\widehat{\mathbf{h}})$ in the following way for any invertible $\pi_{\widehat{\mathbf{h}}} : \mathbf{d}_>^k(\widehat{\mathbf{h}}) \to [|\mathbf{d}_>^k(\widehat{\mathbf{h}})|]$*

$$\mathtt{fT}^k|_{\mathbf{d}^k(\widehat{\mathbf{h}})}(\mathbf{h}) = \left( y_{\pi_{\widehat{\mathbf{h}}}(i)} = \mathtt{tRR}_i^k(\mathbf{h}) : i \in \mathbf{d}_>^k(\widehat{\mathbf{h}}) \right).$$

*Thus, given a histogram $\widehat{\mathbf{h}}$ the limited mapping $fT^k|_{\mathbf{d}^k_>(\widehat{\mathbf{h}})}$ is $(\varepsilon'(\delta'), k\delta + \delta')$-DP for any $\delta' \geq 0$ where*

$$\varepsilon'(\delta') = \min\left\{k\varepsilon, k\varepsilon \cdot \left(\frac{e^\varepsilon - 1}{e^\varepsilon + 1}\right) + \varepsilon\sqrt{2k\ln(1/\delta')}\right\}.$$

*Proof.* This follows from Corollary F.1 as well as basic and advanced composition given in Theorem 3. $\qquad\square$

Note that if we are given a set $D \subseteq \mathbf{d}^k_>(\widehat{\mathbf{h}})$, then we can equivalently write it as a vector $\mathbf{y}$ where each coordinate $y_i = 1$ if $\pi_{\widehat{\mathbf{h}}}(i) \in D$ and $y_i = \perp$ otherwise.

**Corollary F.2.** *Given some collection of subsets $\mathcal{S} \subseteq 2^{[d]}$, subset $T \subseteq [d]$, and histogram $\mathbf{h}$, we denote $\mathcal{S}|_T = \{D \in \mathcal{S} : D \subseteq T\}$. Then we must have*

$$\Pr[fT^k(\mathbf{h}) \in \mathcal{S}] = \Pr[fT^k(\mathbf{h}) \in \mathcal{S}|_{\mathbf{d}^k_>(\mathbf{h})}]$$

**Lemma F.4.** *For any neighboring histograms $\mathbf{h}, \mathbf{h}'$ and any $\mathcal{S} \subseteq 2^{[d]}$, along with $k < d$ and parameters $\varepsilon, \delta > 0$ and $\delta' \geq 0$, then*

$$\Pr[fT^k(\mathbf{h}) \in \mathcal{S}] \leq e^{\varepsilon'(\delta')}\Pr[fT^k(\mathbf{h}') \in \mathcal{S}] + k\delta + \delta'$$

*where*

$$\varepsilon'(\delta') = \min\left\{\varepsilon k, \varepsilon k\left(\frac{e^\varepsilon + 1}{e^\varepsilon - 1}\right) + \varepsilon\sqrt{2k\ln(1/\delta')}\right\}.$$

*Proof.* We first apply Corollary F.2 to instead consider the set $\mathcal{S}|_{\mathbf{d}^k_>(\mathbf{h})}$. We will fix two neighboring histograms $\mathbf{h}, \mathbf{h}'$ and by Lemma 6.7 we need to only consider two cases.

First, if $\mathbf{d}^k_>(\mathbf{h}) \subseteq \mathbf{d}^k_>(\mathbf{h}')$, then we know $\mathbf{h} \in \mathcal{H}_k(\mathbf{h}')$ and $\mathcal{S}|_{\mathbf{d}^k_>(\mathbf{h})} \subseteq 2^{\mathbf{d}^k_>(\mathbf{h}')}$. Let $\pi_{\mathbf{h}'} : \mathbf{d}^k(\mathbf{h}') \to [|\mathbf{d}^k_>(\mathbf{h}')|]$ be an invertible mapping such that for every $S \subseteq \mathbf{d}^k_>(\mathbf{h}')$ there is a $\mathbf{y}^{(S)} \in \{1, \perp\}^{|\mathbf{d}^k_>(\mathbf{h}')|}$ such that $y^{(S)}_{\pi_{\mathbf{h}'}(i)} = 1$ if $i \in S$. Then we have

$$\Pr[\mathtt{fT}^k(\mathbf{h}) = S] = \Pr[\mathtt{fT}^k|_{\mathbf{d}^k_>(\mathbf{h}')}(\mathbf{h}) = \mathbf{y}_S]$$

and

$$\Pr[\mathtt{fT}^k(\mathbf{h}') = S] = \Pr[\mathtt{fT}^k|_{\mathbf{d}^k_>(\mathbf{h}')}(\mathbf{h}') = \mathbf{y}_S]$$

Hence, we also have equality in the probability statements when $\mathtt{fT}^k(\mathbf{h}) \in \mathcal{S}\,|_{\mathbf{d}^k_>(\mathbf{h}))}$ and $\mathtt{fT}^k(\mathbf{h}') \in \mathcal{S}\,|_{\mathbf{d}^k_>(\mathbf{h}))}$. We then apply Lemma F.3 to get the result.

The second case follows symmetrically.

$\qquad\square$

Summarizing the above results we have the following theorem.

**Theorem 5.** *Algorithm 8 is $(k\varepsilon, k\delta)$-DP, and also $(\varepsilon\sqrt{2k\ln(1/\delta')} + \varepsilon\left(\frac{e^\varepsilon - 1}{e^\varepsilon + 1}\right)k, k\delta + \delta')$-DP for $\delta' > 0$.*

We point out that in either the unrestricted sensitivity or the $\Delta$-restricted sensitivity setting, Algorithm 8 will still be differentially private with the same privacy parameters as in the above theorem, but we cannot improve the factor of $k\varepsilon$ to $\Delta\varepsilon$ because the count of a single element that a user contributed to can modify the threshold and change the count of all the at most $k$ elements above this threshold.