[Reviews · NeurIPS 2019]

Reviewer 1



Content: The authors’ explore the problem of DP top-k selection. They relax the requirements of the problem to allow the release of less than k of the top-k elements. This allows the authors’ to use a “noisy threshold” technique that allows them to produce a DP algorithm that does not need to know the entire domain. They further show that if the database contains k elements whose counts are significantly higher than the remaining elements then all k will be released with high probability. The algorithm also has an option of releasing a special symbol if at some point the remaining elements are all approximately equal. Originality: This work contains some original ideas. The major step of a noisy threshold seems somewhat familiar but it’s application in this setting is, to my knowledge, new. The step of reducing the effective domain size that is worked over is interesting. In practical applications, gaining constants in the privacy parameters is important. I have not seen an advanced composition theorem of this form before and since the authors’ prove it is applicable for the exponential mechanism with monotonic quality function, it seems broadly applicable. Quality: The paper seems technically sound. It contains full proofs and the results are clearly presented. I did not read the proofs in detail but the intuition is clear and a cursory look at the proofs showed no holes. Clarity: The paper is well-written. It is organized and easy to follow. The supplementary contains a larger version (presumably the arXiv version) that contains proofs. In the shortened version the authors’ do a good of providing the intuition of the proof. In my opinion, after reading the short version, an interested reader would have a good idea of how and why the results are true (although they may not be able to prove them without the supplementary). Significance: The authors’ consider an important problem in a more realistic context than has previously been considered. Unlike previous work, the algorithm presented in this paper does not require knowledge of the entire domain. This is important since in real-world systems, the algorithm rarely has access to even a description of the entire domain. It is difficult to achieve for a private algorithm because the unknown domain elements can appear in a neighboring database. The authors’ side-step this issue by only releasing elements that are clearly in the top-k, i.e. elements that definitely also appear in any neighboring database. The algorithm is also built to run on top of current systems that are already in place, rather than requiring specialized infrastructure. This is an important feature for speedy, large-scale deployment by non-experts. The paper also contains several results of independent interest. Perhaps the most interesting is the composition theorem for “eps-range bounded” mechanism. This allows them to shave constants off the advanced composition theorem for mechanisms that satisfy a slightly stronger condition than DP. They show that the exponential mechanism with a monotonic quality function satisfies this stronger condition. This allows them to prove a “pay what you get privacy” result (this result allows them to make iterative calls to their private top-k mechanism and essentially only pay for the number of elements output, rather than the number of calls made to the algorithm). They also present a proof of the fact that rather than using the exponential algorithm to iteratively “peel-off” the top-k results, one can simply add Gumbel noise to the quality scores and output the top-k noisy scores. This result has appeared in the literature before (e.g. “Private False Discovery Rate Control” [Dwork, Su, Zhang, arXiv ’15]) but is still a reasonably unknown result. Although it is not an original result, it’s nice to see it presently succinctly and clearly. Minor comments: - Typo, line 29. “In this work,…”

Reviewer 2



+ The problem is well-studied in the DP literature, but the paper still manages to provide a contribution that is novel and has practical relevance. The fact that the algorithm does not need to know the domain is great. + The algorithmic ideas are fairly natural and has Propose-test-release style flavor. Q: Can we include sparse-vector to characterize the privacy cost at a more fine-grained level? - It would be interesting to see a practical/empirical evaluation of the algorithms mentioend, while highlighting the savings in privacy cost, and compare it to existing results in the literature.

Reviewer 3



The paper contains many results, which is, unfortunately, counts as a weakness due to its poor organization. For example, there's an intermission on "bounded range composition", which introduces a strengthening of pure DP. Epsilon-range-boundedness ends up being a factor-2 approximation of pure-DP, and it is then shown to admit a slightly better (not even by a factor of 2) advanced composition theorem for a special kind of exponential mechanism. By itself, this is a useful observation (which appears already as a remark in the original McSherry & Talwar work) but does it really strengthen the paper? Overall the paper has several interesting ideas. First, it is the relaxation of the top-k objective. Second, it is the use of the Gumbel distribution whose connection to the exponential mechanism has not be fully explored. Third, it is the pay-what-you-get accounting mechanism. Everything else adds additional and unnecessary layers of complexity. It would interesting to frame the paper's approach using the language of privacy odometers and filters (https://arxiv.org/abs/1605.08294). The "Odometers" paper seemingly warns against the type of composition that the submission pursues; it'd be worthwhile to identify reasons why the submission avoids the problems presented by odometers. Another possible connection with previous work is "Scalable Private Learning with PATE" (https://arxiv.org/abs/1802.08908). The key primitive of that paper is top-1 query, and its main insight is that there are significant savings in the privacy budget can be extracted if the querying mechanism does not answer (returns a bottom) when there's no clear favorite.

[Author Response · NeurIPS 2019]

We thank the reviewers for their thorough reviews of our paper. We will fix the typos for the next submission. We want to clarify for all the reviewers that our improved composition result for range-bounded mechanisms applies to all instantiations of the exponential mechanism, and not just those with monotonic quality score. To help clarify, Lemmas 4.1 and 4.3 in our submission states that without any knowledge of the quality score, exponential mechanism is $\epsilon$-DP as well as $\epsilon$-range bounded. If it is known that the quality score is monotonic, then the same exponential mechanism is $\epsilon/2$-DP and $\epsilon/2$-range bounded. In both cases our composition bound will improve on advanced composition and in most cases the optimal DP composition bound. We address the questions and concerns for each reviewer below.

**Reviewer 1:** As the reviewer pointed out, we were careful to cite the 2015 version rather than the 2018 version of Dwork, Su, Zhang since the one-shot top-$k$ result was removed from the more recent version. However, our result is a strict improvement on the result in the 2015 version. Specifically, by adding Gumbel noise and taking the top-$k$ in one shot, we allow for the output to be a ranked list of elements, as would be the case with a peeling exponential mechanism approach. In DSZ'15, they use Laplace noise to return a set of $k$ elements in one shot and crucially cannot give the ranking. We state this in the introduction of our submission, although we use the incorrect author name "Qiao, et al" despite it being the correct citation. We will fix this in the next submission. We were not aware of a formal citation for the connection between exponential mechanism and Gumbel noise since it might be folklore in the community. However, the connection between peeling-exponential mechanism with a one-shot Gumbel, to our knowledge, has not appeared elsewhere, although we do not view this as one of the primary contributions of this work.

**Reviewer 2:** We interpret the question regarding sparse-vector as whether Gumbel noise can instead be added to this known algorithm, and then apply our improved composition bound. We think this is an interesting direction of future work, but in our initial attempts, we do not see a natural way for this variant to be range bounded.

Due to space limitations and without a natural comparison with existing top-$k$ DP algorithms (others must return $k$ elements and use a known domain), we did not include experiments. However, we did compare our range bounded composition bound with the existing optimal DP composition bound to show that we can get a significant improvement.

**Reviewer 3:** We see that the range bounded characterization of our algorithms is a significant contribution of this work. This was crucial in showing that we can improve on, not just advanced composition, but the optimal DP composition. If an algorithm is $\epsilon$-range bounded, one can only conclude it is also $\epsilon$-DP, but one can save almost a factor of two in composition knowing the sequence is also $\epsilon$-range bounded. We were unable to find such a remark in McSherry and Talwar that makes a similar observation to range-boundedness. Perhaps the remark the reviewer is referring to is the distinction of monotonic vs general quality scores, where there is a savings of 2 in the algorithm's DP parameter. On the other hand, range boundedness provides an additional savings in composition, which to our knowledge was not known before. We certainly understand and agree with the reviewer that, from a pure theory perspective, a 50% savings is not substantial. However, if we fix the total epsilon of a privacy system, this savings can translate to a significant increase in the total number of queries, which translates to the usefulness and viability of a product. Given that the exponential mechanism is widely used and fundamental in the DP literature, we felt that this was the appropriate venue for a combination of such a theoretical and practical contribution.

We have also explored the connection of privacy odometers [Rogers et al 2016] to this setting (as mentioned in the Related Works section). However, there is a crucial distinction between their setting and ours. In the odometer setting, the privacy parameter is determined based on the previous outcomes. In the context of random walks, the size of the step at round $t$ is completely determined by the previous steps at round $1, \ldots, t - 1$. In our current setting, the privacy loss for a call to the top-$k$ algorithm depends on the size of the outcome at the current round $t$. Hence, we cannot determine the privacy loss if we condition just on the previous outcomes, since it depends on the current round's randomness. This distinction makes privacy odometers not applicable in our setting. However, as we show, our algorithms can be analyzed as a sequence of range bounded algorithms despite each top-$k$ algorithm returning possibly fewer results. Hence, composition follows from analyzing the full sequence of range bounded mechanisms.

We have not considered the application of this approach to PATE, which is a great direction for future work. One difficulty with directly applying our results is that the privacy loss for the top-1 would be the same whether a $\perp$ or an actual value were returned.

To address the questions raised:

[full paper, page 7] Yes, $v_\perp$ is the threshold, we will fix this in the next submission. The rest of the sentence "any index in the top-$k$ ... but not in top-$\bar{k}$ for a neighboring histogram . . ." is correct as is with $\bar{k}$.

[full paper, page 29] We used the terms "soundness" and "completeness" since we wanted to directly compare to the utility results in Bhaskar et al. who also use those terms.

[Meta-Review · NeurIPS 2019]

The paper introduces several interesting, new ideas leading to novel results for top-k selection under differential privacy. The techniques devised in this work, including the composition accounting technique, are expected to be useful in future research in differential privacy.